# Bilateral interactions of optic-flow sensitive neurons coordinate course control in flies

Victoria O. Pokusaeva[1,2,3], Roshan Satapathy [1,3], Olga Symonova [1] & Maximilian Joesch [1] ✉

Animals rely on compensatory actions to maintain stability and navigate their environment efficiently. These actions depend on global visual motion cues known as optic-flow. While the optomotor response has been the traditional focus for studying optic-flow compensation in insects, its simplicity has been insufficient to determine the role of the intricate optic-flow processing network involved in visual course control. Here, we reveal a series of course control behaviours in *Drosophila* and link them to specific neural circuits. We show that bilateral electrical coupling of optic-flow-sensitive neurons in the fly's lobula plate are required for a proper course control. This electrical interaction works alongside chemical synapses within the HS-H2 network to control the dynamics and direction of turning behaviours. Our findings reveal how insects use bilateral motion cues for navigation, assigning a new functional significance to the HS-H2 network and suggesting a previously unknown role for gap junctions in non-linear operations.

During visual course control, animals rely on optic-flow as an important source of information about their self-motion and the structure of the environment. Optic-flow patterns change stereotypically with each of the animal's movements. Thus, by extracting global motion patterns, an animal can generate a faithful inference of its self-motion and elicit stabilising responses. The classic example of a visual course stabilising response is the optomotor reflex[1], a counteractive compensatory reaction observed in many animals[2] that minimises retinal slip during inadvertent movements. However, the accurate interpretation of visual motion cues can be equivocal when only local information is available. Distinct movements can result in similar local optic-flow patterns, especially when the visual structure of the environment is nonuniform, as it usually is in natural environments[3,4]. This challenges the ability of the visual system to instruct appropriate corrective motor commands since the same optic-flow could be interpreted in different ways. Here, we use the fruit fly as a model to understand how ambiguous motion cues, i.e., unilateral and bilateral optic-flow stimuli, are binocularly integrated across the visual field in order to instruct appropriate motor sequences.

In the fly's brain, the circuit thought to be primarily involved in steering stabilising responses is located in the posterior part of the lobula complex, called the lobula plate[5]. This circuit comprises a group of ~60 direction-selective neurons in each hemisphere called lobula plate tangential cells (LPTCs)[6]. Different classes of LPTCs encode optic-flow generated by self-motion about either translation, yaw, roll, or pitch axes[5–7]. One such class includes three yaw motion-sensitive horizontal system (HS) cells that respond to front-to-back (FtB) motion by changing their graded potential[8] and two spiking neurons, H1 and H2 cells, that increase their firing rate to back-to-front (BtF) motion[9]. Optogenetic and chemogenetic activation of HS cells in *Drosophila* evoked directed head movement and flight turns[10–12]. Accordingly, silencing of HS neurons reduced head optomotor response, albeit without substantially affecting body turns[13], suggesting that they are involved in gaze and course stabilisation.

The sensitivity to a specific pattern of optic-flow in the LPTCs arises primarily from the spatial organisation of the direction-selective input onto these neurons[3,7,14–17]. Nevertheless, this specificity is further enhanced by the synaptic interactions between individual LPTCs within one or both hemispheres of the fly brain. Synaptic connections between LPTCs are mostly electrical[8,18–21] and comprise ShakB gap junction channels[18,19,22,23]. These lateral electrical synapses mediate a direct flow of electrical current, allowing individual neurons to

[1]Institute of Science and Technology Austria (ISTA), Klosterneuburg, Austria. [2]Present address: Department of Biological Sciences, University of Toronto Scarborough, Toronto, ON, Canada. [3]These authors contributed equally: Victoria O. Pokusaeva, Roshan Satapathy. ✉e-mail: maxjosch@ist.ac.at

integrate motion information from visual areas outside their retinotopic dendritic inputs[18,19,24]. Indeed, experimental and modelling studies suggest that gap junctions between ipsilateral LPTCs enable robust encoding of flow-field parameters by refining the structure of their spatial receptive fields[25]. Interestingly, these electrical connections have been shown to mediate binocular interactions too[19,20]. The behavioural role of this contralateral connectivity remains speculative, being suggested to be required to disambiguate inadvertent translational and rotational movements in the horizontal plane, as indicated for neck motor neurons[26]. However, despite numerous predictions[19,23,27], there is little direct evidence establishing the role of electrical connections in the steering responses generated by LPTCs.

In this study, we provide experimental evidence for the crucial role of binocular interactions in the fly's course control system. Using a combination of behavioural, electrophysiological, and genetic approaches, we show that bilateral electrical connection plays a fundamental role in interpreting ego-motion-induced optic-flow patterns, expanding our understanding of the computational roles of electrical synapses in sensorimotor transformations and behavioural control.

## Results

### Antagonistic behavioural responses to unilateral optic-flow patterns

To explore the role of binocular optic-flow integration in an unconstrained setting, we built a circular arena where flies walked freely, their position was tracked, and the body axis was used for closed-loop presentation (<48 ms delay) of visual stimuli onto the arena's roof (Fig. 1a, Supplementary Fig. 1, and see methods)[28,29]. This virtual environment covered a substantial part of the animal's dorsal visual field starting ~5° above the horizon, extending ~170° in azimuth and elevation (Fig. 1b, Supplementary Fig. 2a). This paradigm enables the control of the fly's visual experience by locking arbitrary optic-flow patterns to each eye independently while allowing for natural proprioceptive feedback during locomotion. Using this system, we first reproduced the classic optomotor behaviour, a strong and robust turning reaction in the direction of the stimulus (Fig. 1c), when presenting wild-type *CantonS* flies with a radial periodic grating (pinwheel). This optomotor behaviour was composed of smooth turns and saccades (Fig. 1d). Interestingly, we also consistently observed saccades opposite to the direction of motion (anti-saccades) (Fig. 1d)[13,30]. These anti-saccades have been recently reported in walking flies[31,32], but their origin and functions remain unclear. Next, we decomposed the rotational stimulus into back-to-front (BtF) and front-to-back (FtB) motion in either half of the visual field. In this modified pinwheel stimulus, only half of the pinwheel rotated while the other half remained static, centred on the fly's major body axis, allowing us to present either BtF or FtB motion to one of the two halves of the visual field. While BtF motion elicited a weak and transient turning response in the direction of motion, FtB motion unexpectedly resulted in strong opposite turns (Fig. 1eii–iii, fii–iii, Supplementary Fig. 2, Supplementary Movie 1) – note that the behavioural responses are not absolute but presented relative to the stimulus direction. Crucially, unlike a recent report of anti-directional turning behaviour during optomotor response[32], the anti-optomotor response to FtB motion that we observe (i) is initiated immediately after the motion onset, (ii) is sustained over the whole duration of the trial and (iii) shows little to no turning in the direction of rotation.

Full-field rotation of the pinwheel can be decomposed into FtB and BtF motion, thus, we expected that the fly's response to the full-field rotation would be a simple summation of its response to FtB and BtF motion. However, the observed full-field response is drastically different from our linear prediction (Fig. 1g, h), contrary to a previous observation during flight using similar stimuli[33]. During flight, the behavioural response to a hemifield stimulus appears to produce saturated responses that show a small change in their linear prediction

in the direction of motion. During walking, the linear prediction would indicate that the flies would rotate against the classical optomotor response. This suggests that the fly's heading control system integrates visual information from the two halves of the visual field in a non-linear way. To determine the source of this non-linear behavioural response, we characterised the difference in behavioural properties across the three stimuli by separating the smooth and saccadic turning responses for each trial (see methods). While the classic optomotor response to full-field rotation is primarily composed of smooth turns (Fig. 1i left), the turning response to BtF motion has an equal contribution of both smooth and saccadic turning (Fig. 1i middle). Strikingly, the anti-optomotor turning in response to FtB motion is predominantly saccadic (Fig. 1i right, Supplementary Fig. 2d, h–j), eliciting faster changes in the heading direction (Supplementary Fig. 2b). Accordingly, in response to FtB motion, the locomotion path is straighter (Supplementary Fig. 2c). Thus, flies alter the direction as well as the nature of their turning response for different types of motion, which accounts for the significant prediction errors for both smooth and saccadic turning (Fig. 1h, j). The relative contribution of smooth and saccadic turning is evident across flies for each stimulus (Fig. 1k, m). Interestingly, although the number of anti-saccades per trial is higher (Supplementary Fig. 2d), syn- and anti-saccades contributed equally to the full-field optomotor responses (Fig. 1l) due to larger syn-saccadic turns (Supplementary Fig. 2e, f).

Although responses to FtB and BtF motion could be interpreted as escaping or turning away from the location of unilateral motion irrespective of its direction as shown during tethered flight[34], the kinetics and relative contribution of smooth and saccadic responses for FtB and BtF motion differ drastically (Fig. 1i, m). This can be observed in the different relative number of syn- and anti-saccades and their respective amplitudes and velocities (Supplementary Fig. 2). This indicates that FtB and BtF motion drive different modes of action, as evidenced by the difference in the straightness of the trajectories (Supplementary Fig. 2c, d). The stimulus dependence of the FtB anti-saccadic response becomes apparent as the stimulus strength is reduced. At low contrast, animals begin to reverse their turning direction for FtB motion, from avoidance to stabilisation (Supplementary Fig. 3a–c). We observed a similar dependence when presenting a different type of optic-flow stimulus, the 'star field' stimulus[35], that induced a weaker optomotor response. Consistent with our low contrast results, we observed that half of the flies showed a robust anti-response while the other half did not (Supplementary Fig. 3e–g). These results suggest that the robustness of the anti-saccadic response depends on the strength of the local motion stimuli and demonstrate the existence of stereotyped behavioural adaptations to nuanced changes in stimulus properties.

Given that flies have a binocular field that spans 40° of visual angle[36], our unilateral split-screen stimuli also stimulates the contralateral eyes. To test whether anti-saccadic behaviour requires binocular field of view overlap, we repeated the experiment while masking the binocular field. Although flies show weaker responses to FtB optic-flow, possibly due to the smaller optic-flow stimulus, masking the binocular FOV did not qualitatively change the anti-saccadic responses (Supplementary Fig. 3i, j).

In summary, wild-type flies show opposite and stimulus-dependent responses to monocular and unilateral FtB and BtF motion. Furthermore, these results reveal a non-linear binocular interaction that generates the classical optomotor response to full-field rotation and extend the repertoire of course control behaviours in walking *Drosophila*.

### LPTCs are required for binocular control of walking

To define the circuitry instructing non-linear binocular behaviours, we focused on the lobula plate tangential cells (LPTC), a network of neurons sensitive to wide-field motion and thought to influence

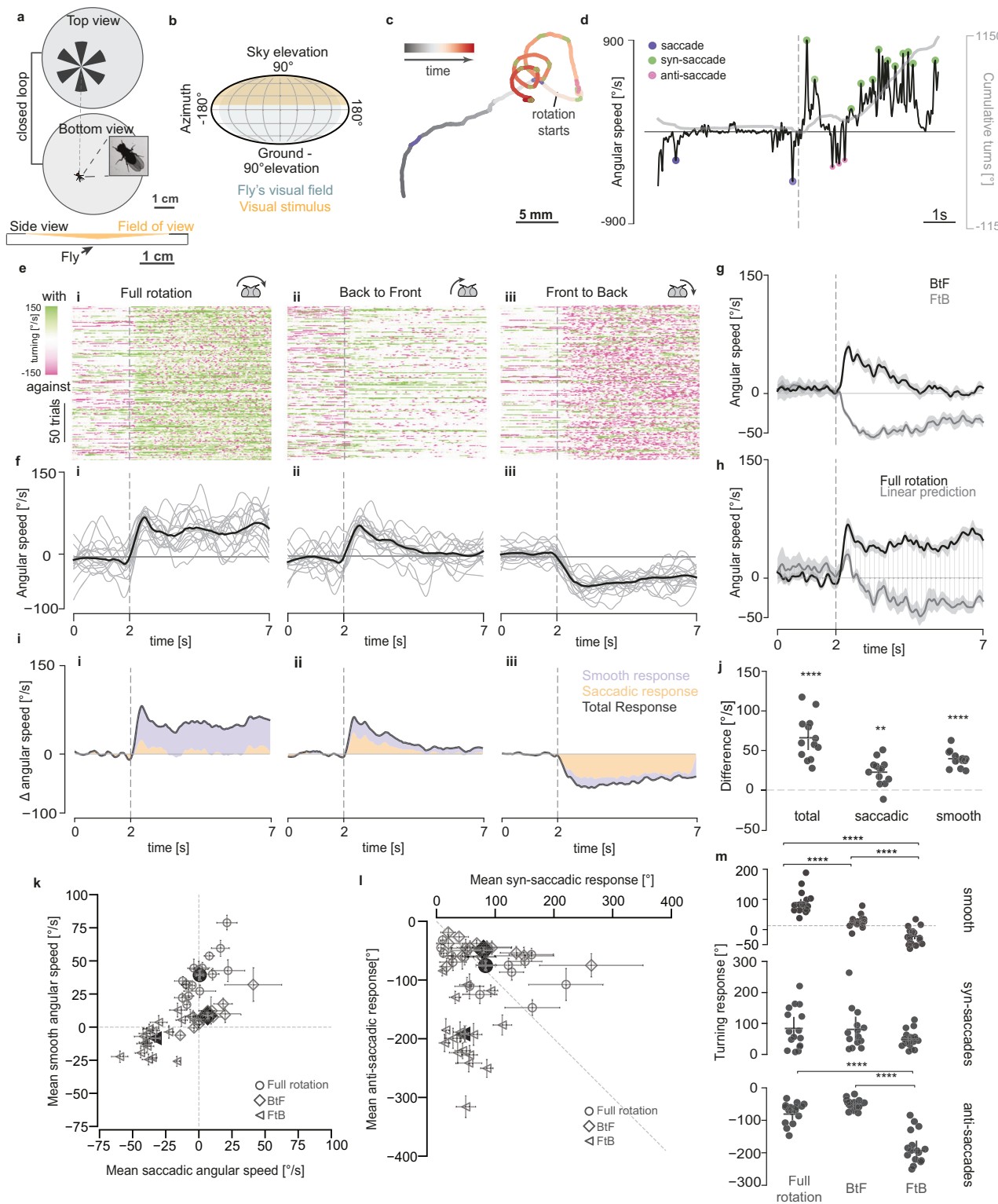

optomotor response in flies[19,37]. We silenced horizontal system (HS) and vertical system (VS) neurons by expressing the inwardly rectifying potassium channel Kir2.1. By combining the *VT058487-GAL4* driver line with the GAL4 repressor expressed in the ventral nerve cord, we achieved high levels of specificity (Fig. 2a). Surprisingly, silencing all HS and VS neurons did not abolish the classic optomotor response, in spite of a decrease in the overall strength. However, the response to FtB motion was markedly different, with flies showing an early transient response against the direction of motion, followed by a late sustained response in the direction of motion (Fig. 2b, c,

Supplementary Fig. 4a). More importantly, in flies with HS and VS neurons silenced, the response to full-field rotation matches the linear prediction (Fig. 2c), as demonstrated by the significantly decreased prediction error compared to *UAS-Kir2.1* control flies (Fig. 2d–g, Supplementary Fig. 4b).

Changes in the turning dynamics of the FtB response cause the linearization of optomotor response in flies with silenced HS and VS neurons. While the FtB response in control or wild-type flies is dominated by anti-saccades (Figs. 1l, m, 2i, j Supplementary Fig. 2d), silencing of HS and VS neurons results in a decrease in anti-

**Fig. 1 | Differential binocular control of walking behaviour in wild-type flies.**
**a** Schematic of the closed-loop behavioural setup. **b** Extent of the stimulus on the fly's visual field of view (binocular overlap, dark grey; from[81]), presented as a Mollweide 2D projection. **c** Example trajectory of a *CantonS* fly during a typical trial of a clockwise rotating stimulus. **d** Angular speed (black) and heading (grey) of the fly corresponding to the trial in (**c**) with syn- and anti-saccades highlighted. Dashed line indicates the beginning of the rotation. **e** Turning response of *CantonS* flies to (i) full-field, (ii) unilateral Back-to Front (BtF) and (iii) unilateral Front-to Back (FtB) rotation stimulus. Angular speed raster showing 200 randomly selected trials. Each row corresponds to one trial, each column to one frame (-16 ms). **f** Mean angular speed per fly (grey) and across all flies (black). **g** Angular speed of *CantonS* flies in response to FtB and BtF rotation (mean ± SEM). **h** Comparison of predicted and actual mean angular speed in response to full-field rotation for *CantonS* flies (mean ± SEM). Shaded region shows the difference between predicted and actual

response. **i** Stacked plot showing mean angular speed and the contribution of smooth and saccadic turning. **j** Mean prediction error across the trial period per fly. Bars: mean ± SEM. **k** Scatter plot of the mean of smooth and saccadic angular speed across experiments. Empty and filled markers show mean ± SEM per fly and per stimulus condition, respectively. **l** Cumulative angular displacement for syn-saccadic and anti-saccadic turns in one trial. Empty and filled markers show mean ± SEM per fly and per stimulus condition, respectively. **m** Mean of smooth (top), syn-saccadic (middle) and anti-saccadic (bottom) cumulative angular displacement per trial (mean ± SEM) (**j**) A one-sample two-sided *t*-test was applied to check if the error differed significantly from zero. **m** Two-sided Mann–Whitney *U*-test, \*p < 0.05, \*\*p < 0.01, \*\*\*p < 0.001, \*\*\*\*p < 0.0001. No asterisk: not significant. Number of flies: n = 17. Exact p-values for each experiment are listed in Supplementary Data 1. Schematic drawings credited to Laura Burnett.

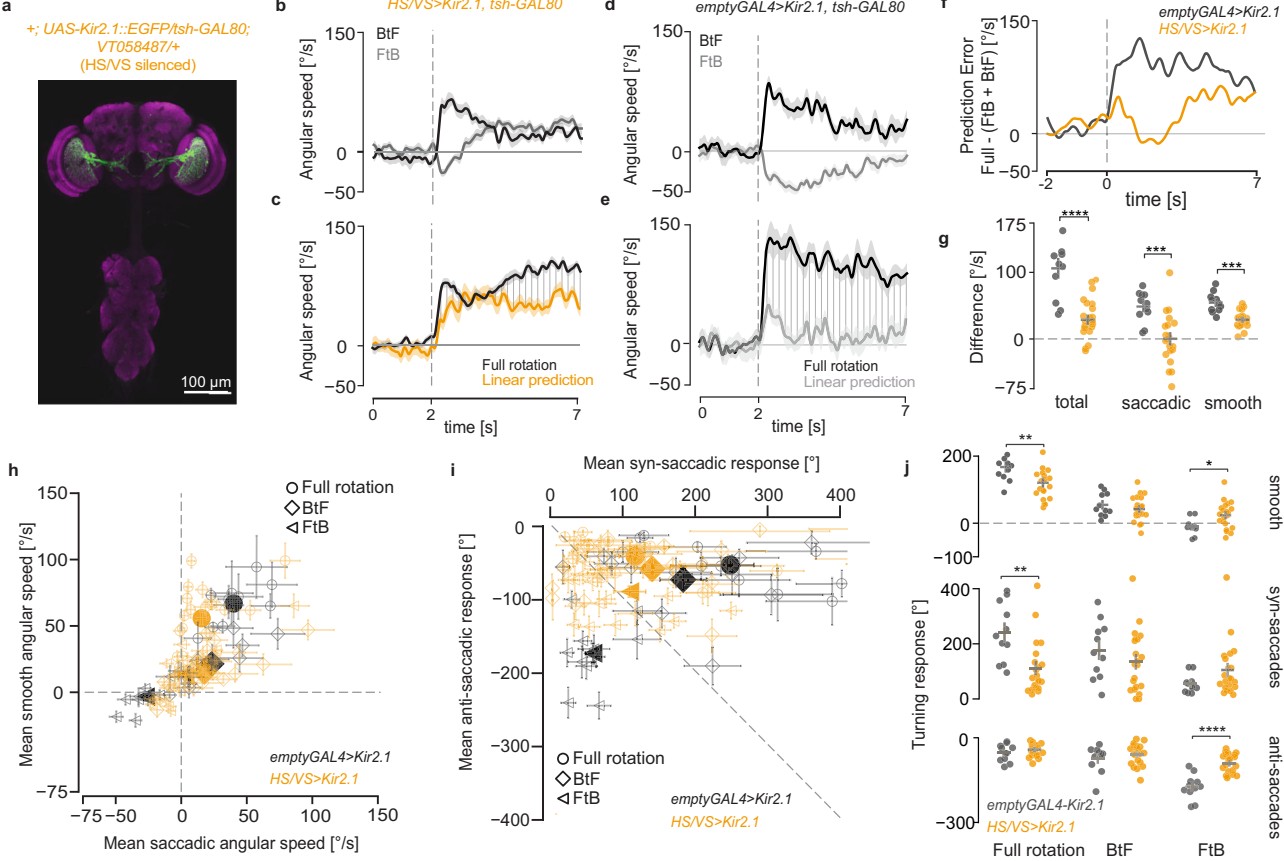

**Fig. 2 | LPTCs are required for binocular control of walking. a** Maximum z-projection of Kir2.1::EGFP expression by the *VTO58487-GAL4* line used to silence HS and VS cells. The flies also carried a tsh-GAL80 transgene to remove GAL4 expression in the ventral nerve cord. **b** Angular speed of HS & VS-silenced flies in response to FtB and BtF rotation (mean ± SEM). **c** Comparison of the predicted and the actual mean angular speed in response to full-field rotation for HS & VS-silenced flies (mean ± SEM). Shaded region shows the difference between the predicted and actual response. **d** Same as in (**b**) but for *UAS-Kir2.1* control flies. **e** Same as in (**c**) but for *UAS-Kir2.1* control flies. **f** Time series of mean prediction error. **g** Mean prediction error across the trial period per fly. Bars: mean ± SEM. **h** Mean of smooth and saccadic angular speed across a trial. Empty and filled markers show the mean

per fly and mean ± SEM for each genotype, respectively. **i** Cumulative angular displacement for syn-saccadic and anti-saccadic turns in one trial. Empty and filled markers show the mean per fly and mean ± SEM for each genotype, respectively. **j** Mean of smooth (top), syn-saccadic (middle) and anti-saccadic (bottom) cumulative angular displacement per trial. Bars: mean ± SEM. **g, j** Two-sided Mann–Whitney *U*-test, \*p < 0.05, \*\*p < 0.01, \*\*\*p < 0.001, \*\*\*\*p < 0.0001. No asterisk: not significant. Number of flies: *HS,VS>Kir2.1,tsh-Gal80* = 21, *emptyGal4>Kir2.1,tsh-Gal80* = 11. Exact p-values for each experiment are listed in Supplementary Data 1. Source data are provided as a Source Data file. In all panels, black = empty-Gal4>Kir2.1, tsh-Gal80 and orange = *HS,VS>Kir2.1, tsh-Gal80*.

saccadic response and an increase in syn-saccadic response (Fig. 2h–j, Supplementary Fig. 4d, Supplementary Movie 2). This causes the overall response to become smooth (Supplementary Fig. 4aiii), further demonstrated by a decrease in path straightness (Supplementary Fig. 4c). Consequently, the prediction error for saccadic turning is nearly zero in flies with silenced HS and VS neurons (Fig. 2g).

While our experiments suggest that HS neurons are responsible for generating anti-saccades in response to FtB motion, this conclusion is challenged by previous findings. HS cells are known to be excited by FtB motion[8], and their unilateral activation has been shown to drive behavioural responses in the direction of motion[10,11]. Yet, wild-type flies turn against the direction of motion in response to FtB motion, strongly suggesting the involvement of other wide-field direction-

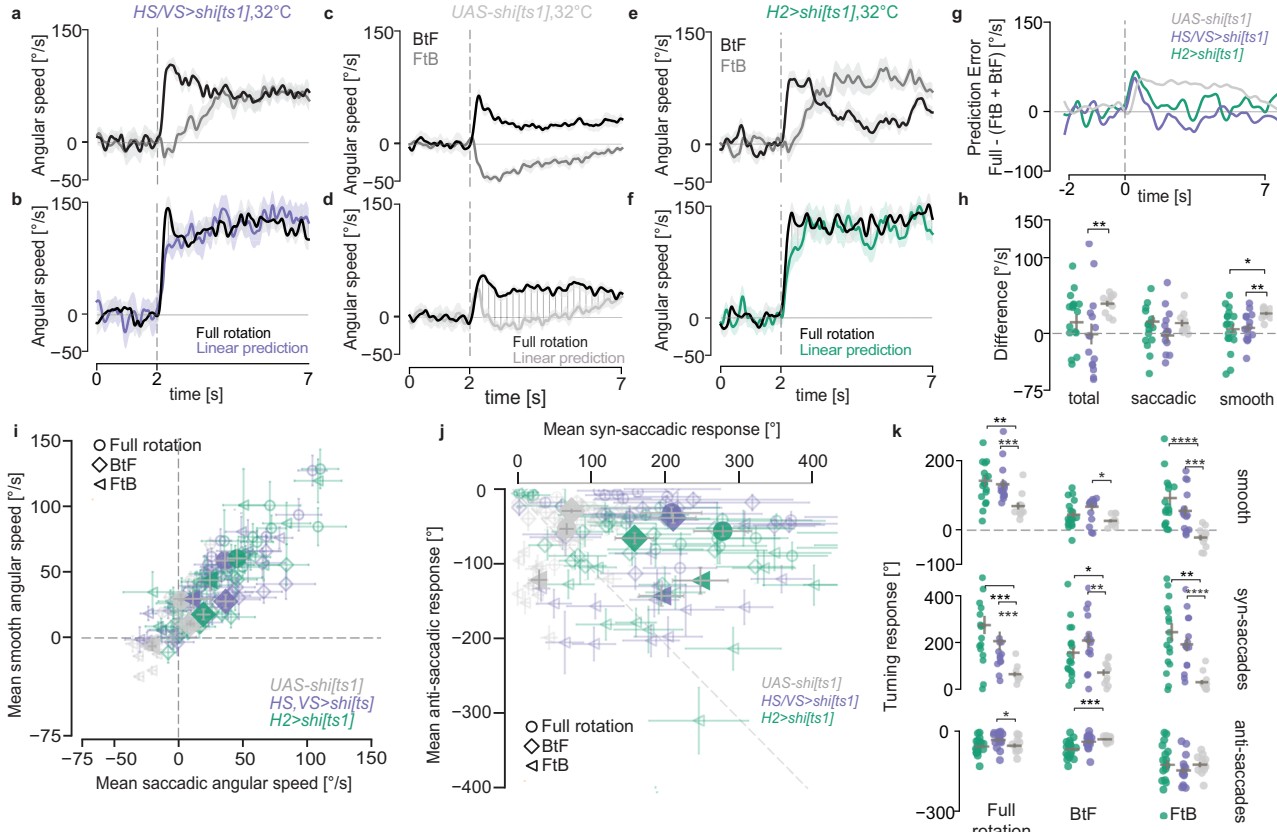

**Fig. 3 | The HS-H2 network orchestrates bilateral visuomotor processing.**
**a** Angular speed of *HS,VS>shi[ts1]* silenced flies at restrictive temperature (32 °C) in response to FtB and BtF rotation (mean ± SEM). **b** Comparison of the predicted and the actual mean angular speed in response to full-field rotation for *H2>shi[ts1]* silenced flies (mean ± SEM). Shaded region shows the difference between the predicted and actual response. **c** same as (**a**) for *UAS-shi[ts1]* control flies. **d** same as (**b**) for *UAS-shi[ts1]* control flies. **e** same as (**a**) for *H2>shi[ts1]* flies. **f** Same as (**b**) for *H2>shi[ts1]* flies. **g** Time series of mean prediction error. **h** Mean prediction error across the trial period per fly. Bars: mean ± SEM. **i**. Mean of smooth and saccadic angular speed across a trial. Empty and filled markers show the mean per fly and

mean ± SEM for each genotype, respectively. **j** Cumulative angular displacement for syn-saccadic and anti-saccadic turns in one trial. Empty and filled markers show the mean per fly and mean ± SEM for each genotype, respectively. **k** Mean of smooth (top), syn-saccadic (middle) and anti-saccadic (bottom) cumulative angular displacement per trial. Bars: mean ± SEM. **h**, **k** Two-sided Mann–Whitney *U*-test, *p < 0.05, **p < 0.01, ***p < 0.001, ****p < 0.0001. No asterisk: not significant. Number of flies: *HS,VS>shi[ts1]* = 16, *H2>shi[ts1]* = 18, *UAS-shi[ts1]* = 13. Exact *p*-values for each experiment are listed in Supplementary Data 1. Source data are provided as a Source Data file. In all panels, grey = *UAS-shi[ts1]*, purple = *HS,VS>shi[ts1]*, and turquoise = *H2>shi[ts1]*.

selective inputs. H2, a spiking LPTC that responds to horizontal BtF motion, is a possible candidate. Previous work in blow flies has shown that these neurons are electrically connected to the HSE cells[19], one of the three known HS cells where E stands for equatorial (a reference to their receptive field position centred on the equator), in the contralateral hemisphere and are their primary source of contralateral input[19]. Unfortunately, we could not properly inhibit H2 neurons using Kir2.1 due to the very weak expression in their axons and dendrites (Supplementary Fig. 4e). Consequently, we did not observe any behavioural difference (Supplementary Fig. 4e).

Recent connectomic analysis of the HS-H2 network revealed complex bilateral interactions based on chemical connectivity, involving HS, H2, LPTC recurrent neurons (LPTCrn) and inferior posterior slope neurons (bIPS) forming a competitive disinhibitory network[15]. As Kir2.1 manipulation disrupts both chemical and electrical synaptic output, we decided to test the behavioural contribution of these two distinct types of synapses separately. First we directly disrupted chemical synapses using *shibire[ts38]*; a dynamin orthologue that blocks chemical synaptic transmission at restrictive temperatures. We expressed *shibire[ts]* using the *VT058487-GAL4* line in HS and VS cells, heated the flies to the restrictive temperature prior to the experiment and tested their responses to FtB and BtF motion. Flies after activation of *shibire[ts]* show a reversal of the anti-saccadic responses to FtB motion (Fig. 3a, b, g–k, Supplementary Fig. 5c, d–g), which was not the case for

*UAS-shi[ts1]* control flies (Fig. 3c, d, g–k Supplementary Fig. 5a). Next, assuming that the use of a dominant-negative disruption might overcome the limitations of low expression levels in the H2 driver line, we performed the activation of *shibire[ts]* to disrupt the synaptic transmission in H2 neurons. We observed similar changes in fly responses (Fig. 3e–k, Supplementary Fig. 5b) to that observed when blocking HS & VS synaptic output. Taken together, these results show that the larger HS-H2 network[15] is required for course control.

**Validation of a new inducible *shakB* mutant**
Given the interhemispheric electrical connectivity between HS and H2 cells[19], we reasoned that electrical communication within the HS-H2 network would also be instrumental for proper binocular course control. However, genetic manipulation of gap junctions, the molecular substrates of electrical synapses, remains challenging. Widely used ethane methylsulfonate (EMS)-induced mutant lines, such as *shakB[2]*, offer the advantage of robust gene inactivation but frequently carry background mutations. Moreover, attempts at cell-specific inactivation of gap junctions in the fly visual system have been met with limited success[22]. Consequently, pinpointing the contribution of electrical synapses within visual neural circuits has remained elusive.

To attempt a detailed description of the role of electrical synapses in LPTCs while avoiding the pitfalls mentioned above, we used the

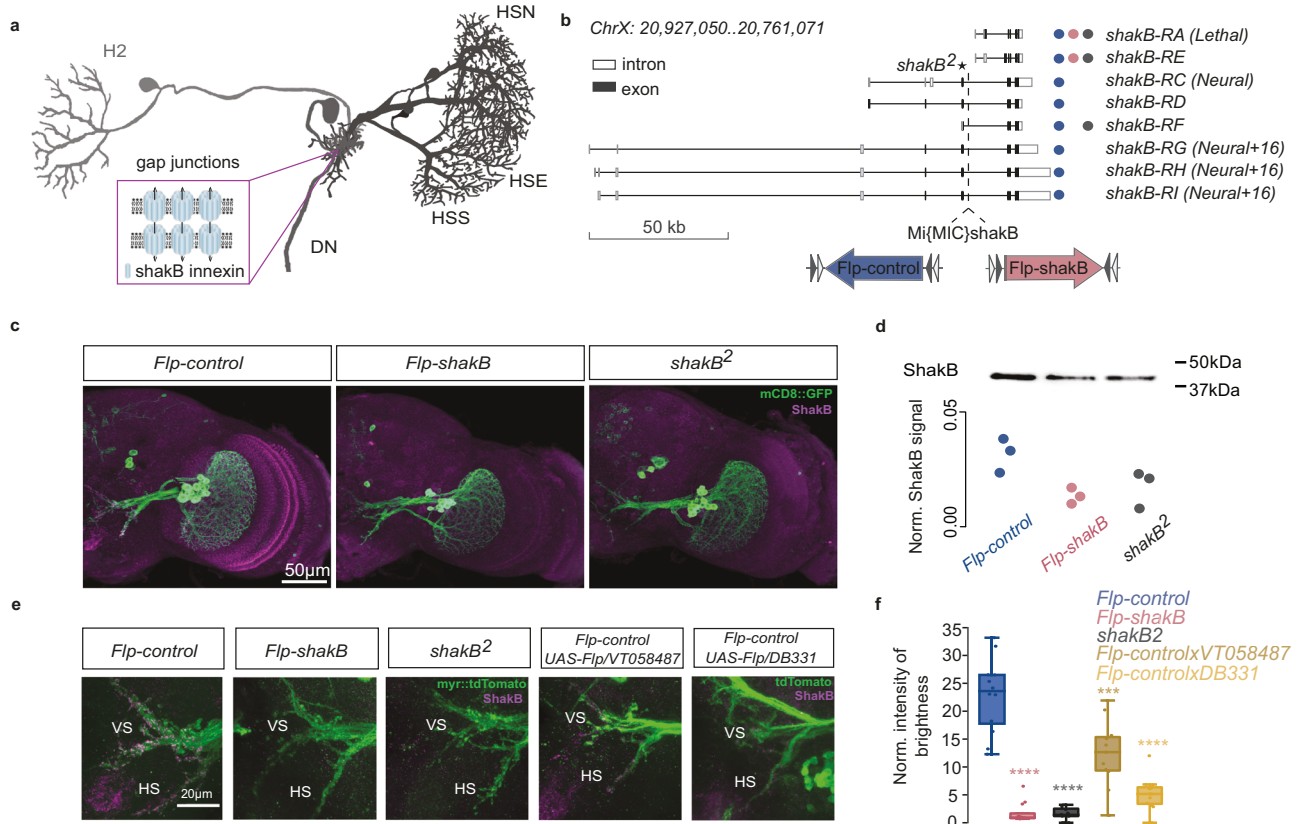

**Fig. 4 | FlpStop *shakB* interventions reveal ShakB protein stability. a** Schematic of known axonal gap junctions in horizontal system LPTCs. Pairs of ShakB innexins form transmembrane channels that enable the bidirectional flow of electrical current between two adjacent neurons. **b** Gene map of *shakB* isoforms. Nonsense mutation carried by *shakB[2]* flies is indicated with a star. Dashed line indicates the position of the intronic MiMIC cassette used to integrate the FlpStop cassette in disruptive (*Flp-shakB*, magenta) and non-disruptive (*Flp-control*, blue) orientations. The circles indicate isoforms that are intact in *Flp-control* (blue), *Flp-shakB* (magenta) and *shakB[2]* (black) flies. **c** Immunostainings for the gap junction protein ShakB in the optic lobe of wild-type line *Flp-control* and two mutant lines *Flp-shakB* and *shakB[2]*. **d** The western blot analysis of protein extracts from fly brain tissue using antibodies against ShakB. ShakB protein signal was normalised to the total amount of the protein sample per line; *n* = 3 for all cases. **e** LPTCs axonal

staining of ShakB protein in wild-type (*Flp-control*), full mutant (*Flp-shakB* and *shakB[2]*) and induced mutant (*VTO58487-Gal4* and *DB331-Gal4*) flies. Axons of two different LPTC classes, HS and VS cells, are indicated. **f** Quantification of the ShakB signal in LPTC axons (*n* = 6 brains per genotype, each hemisphere was quantified independently). Upper/lower limit and inner horizontal lines of the box plots represent upper/lower quartile and median, respectively; whiskers indicate 1.5 interquartile range from upper/lower quartiles. brown = *Flp-controlxVTO58487* and orange = *Flp-controlxDB331*. All the mutant genotypes were compared to *Flp-control* using two-sided Mann–Whitney *U*-test, ***p < 0.001, ****p < 0.0001. The number of recorded cells and exact *p*-values for each experiment are listed in Supplementary Data 1. Number of brains analysed: Flp_control = 6, Flp_shakB = 6, *shakB[2]* = 7, *Flp-controlxVTO58487* = 6, *Flp-controlxDB331* = 6. Source data are provided as a Source Data file.

FlpStop technique[39]. Specifically, we generated transgenic flies that carry the FlpStop cassette inside *shakB*[22], a member of the innexin family that is responsible for encoding gap junction proteins in LPTCs (Fig. 4a, b, Supplementary Fig. 6a). This approach enables us to generate both full and cell-specific mutants while maintaining an isogenic background, greatly facilitating the interpretation of behavioural experiments. shakB[FlpStop-ND] (non-disruptive orientation – Flp-control) and shakB[FlpStop-D] (disruptive orientation – *Flp-shakB*) flies were created by integrating FlpStop cassette into an intronic MiMIC insertion between exons 5 and 6. This insertion allows the inactivation of 6 out of 8 isoforms of the ShakB protein, leaving isoforms *shakB-RA* and *shakB-RE* intact (Fig. 4b). Notably, the widely used *shakB[2]* mutant[40] carries a null mutation in 5 isoforms, leaving *shakB-RF* undisrupted in addition to *shakB-RA* and *shakB-RE*. Thus, the *Flp-shakB* mutant should, similarly to *shakB[2]* disrupt electrical communication in a cell-specific manner in the nervous system.

We observed a significant reduction in the total amount of ShakB protein in the brain of *Flp-shakB* flies compared to *Flp-control* flies (Fig. 4c, d), a similar reduction as seen in the widely used *shakB[2]* mutant. Recent studies suggest that cell-specific inactivation of ShakB protein using driver lines selective to LPTCs is ineffective[22]. We

hypothesised that it might result from early expression onset and a slow turnover rate of innexins in these neurons. The potential influence of developmental timing on the efficiency of cell-specific inactivation of *shakB* prompted us to trigger the inversion of FlpStop cassette using two LPTC-specific driver lines – *DB331-Gal4* and *VTO58487-Gal4*. *DB331-Gal4* line initiated Flp-mediated cassette inversion in LPTCs at around pupal stage P9 while *VTO58487-Gal4* at around P12 (Supplementary Fig. 6c). ShakB immunolabelling in LPTC axons showed that *DB331-Gal4* induced a stronger knock-down of *shakB* than *VTO58487-Gal4* driver line (Fig. 4e, f), indicating that the timing of gene inactivation is a critical determinant of the phenotype. Given our inability to find a specific driver line for HS cells with early expression onset, we were unable to disrupt *shakB* in a cell-type specific way, opting to use *Flp-shakB*, a complete ShakB mutant, for the rest of our study,

Loss of gap junction may severely affect the development of brain tissue[41]. Therefore, it is crucial to rule out developmental phenotypes to avoid misinterpreting physiological and behavioural experiments in the newly generated mutant flies. To account for the potential effects of *shakB* disruption on the expression of other proteins in the fly brain, we performed the proteomic analysis of brain tissue in *Flp-control*,

*Flp-shakB*, and *shakB[2]* flies. Out of 5755 proteins identified, only 7 were upregulated in *Flp-shakB* flies, while 15 were up- and 26 were downregulated in *shakB[2]* flies (Supplementary Fig. 7a, b). The higher number of affected proteins in *shakB[2]* flies is probably due to its distinct genetic background to *Flp-control* flies but could also hint at additional unspecific extraneous mutations that arise through EMS-mutagenesis. Proteins Acbp2, Primo-1, and CG31345 were significantly upregulated in both *Flp-shakB* and *shakB[2]* flies, indicating a functional connection with ShakB protein that remains to be investigated.

Gap junctions were previously shown to be involved in refining neuronal morphology[41] and controlling the formation of chemical synapses[42]. To identify potential differences in the morphology of HS cells in wild-type and mutant flies, we fluorescently labelled individual HS cells using the SPARC technique[43]. The confocal image stacks were used for 3D reconstructions of neurons and then for morphology analysis. We observed no significant differences in dendritic branching and volume between HS cells in wild-type and mutant flies (Supplementary Fig. 8), indicating that no gross morphological changes are present. To label postsynaptic partners of HS cells in mutant and wild-type flies, we used the trans-Tango technique[44]. The postsynaptic partners of HS cells were detected in the optic lobe, in the posterior slope, and in the ventral nerve cord (Supplementary Fig. 9). All the postsynaptic partners of HS cells observed in the wild-type flies were also observed in *Flp-shakB* flies (Supplementary Fig. 9). Meanwhile, the transsynaptic tracing in *shakB[2]* flies revealed fewer synaptic partners of HS cells, with higher labelling variability suggesting that chemical synaptic connectivity might be altered in *shakB[2]* mutants.

Overall, we show that the newly developed FlpStop-based system allows efficient inactivation of ShakB protein with little to-no influence on the proteome as well as the morphology and formation of synaptic connections of HS cells. Furthermore, cell-specific inactivation of ShakB protein is driver line-dependent and can be efficiently achieved only by the inversion of the FlpStop cassette at early pupal stages.

## Loss of gap junctions does not abolish direction-selective responses in HS cells

To characterise the passive membrane properties and visual responses of HS cells lacking gap junctions (*Flp-shakB* flies), we used in vivo whole-cell patch-clamp recordings in restrained flies (Fig. 5a)[21]. We first confirmed that integrating the FlpStop cassette in non-disruptive orientation does not affect the direction-selective responses of HS cells and can be used as a wild-type control (Fig. 5b, Supplementary Fig. 10a, b).

On average, the resting membrane potential of HS cells in the *Flp-shakB* was higher and more variable than *Flp-control*, consistent with observations made for the *shakB[2]* mutant (Fig. 5c–e). The membrane potential of HS cells in *Flp-shakB* flies displayed spontaneous fluctuations (Fig. 5c, f, g), similar to those described in *shakB[2]* mutant flies[22]. Interestingly, while we detected fast β-oscillations of a frequency band similar to *shakB[2]*, we did not observe strongly hyperpolarizing ultraslow waves in *Flp-shakB* flies (Fig. 5c, g). Importantly, we observed membrane fluctuations in flies with cell-specific inactivation of *shakB* in LPTCs only in the rare events where dye coupling was fully abolished (Supplementary Fig. 10c), suggesting that only complete inactivation, and not a reduction of gap junction protein, can induce the membrane oscillations described before[22].

When presented with full-field flashes, ON-transient responses of HS cells in all animals had similar amplitudes, while OFF-transient responses were reduced in *Flp-shakB* flies (Fig. 5h, Supplementary Fig. 10d, e). However, the reduction in OFF-transients was significantly smaller than what was previously reported for *shakB[2]* flies[22]. In addition, we observed strong direction-selective responses in both mutant and *Flp-control* flies in response to bright ON and dark OFF edges travelling at a velocity of 14°/s (Fig. 5i, j, Supplementary Fig. 10f, g). These results suggest that the loss of gap junctions in *Flp-*

*shakB* flies partially affects the visual processing of light decrement, but not increment signals.

Moreover, HS cells in *Flp-shakB* flies did not show the reduced amplitude of direction-selective responses previously described for *shakB[2]* flies[22]. On the contrary, we observed enhanced hyperpolarizing responses to gratings moving in null direction (ND) in *Flp-shakB* flies but not in cell-specific *Flp-controlxDB331* flies (Fig. 5c, k, l, m). Interestingly, while ON-edge moving in ND elicited stronger hyperpolarizing responses in Flp-shakB flies, the amplitude of responses to OFF-edge was similar across genotypes (Fig. 5i, j). It suggests that enhanced hyperpolarization is not a cell-intrinsic phenomenon. Finally, direction (Fig. 5k), contrast (Fig. 5l, Supplementary Fig. 10h), and frequency tuning (Fig. 5m, Supplementary Fig. 10i) show that enhanced hyperpolarizations do not affect the overall tuning properties of HS cells in *Flp-shakB* flies.

Altogether, the analysis of electrophysiological properties shows that the membrane potential of HS cells in *Flp-shakB* flies exhibits fluctuations that do not affect the tuning of direction-selective responses in these cells. Nevertheless, the loss of gap junctions increases hyperpolarization amplitude in response to ND motion.

## Electrical synapses shape HS cell receptive fields

To investigate the impact of gap junctions on the response properties mediated by the LPTC network, we conducted a thorough analysis of the receptive fields (RF) of HS cells in both *Flp-control* and *Flp-shakB* flies. As done previously[8,45,46], we used a local moving spot that scanned the visual field in four cardinal directions to determine their local motion sensitivity (LMS) across large parts of the fly's bilateral visual field (140° in azimuth and 80° in elevation), substantially overlapping with the virtual environment used for the behavioural experiments (Supplementary Fig. 11a). The screen's size and its dorsally displaced positioning did not allow us to resolve the receptive fields of HSS cells, the HS cells with the most ventrally directed RF (S stands for south). Therefore, only HSN (N stands for north) and HSE vector fields were considered for detailed analysis (Fig. 6a, b), where the arrow's length and direction represent the response's relative local strength and preferred direction, respectively.

As shown previously in fruit and blow-flies[8,47], and in correspondence with the dendritic arborizations, RFs of HS cells in Flp-control flies are aligned horizontally, with their RF centres differing in elevation – HSN being more dorsal than HSE. Both types of HS cells showed sensitivity to local back-to-front motion in the contralateral visual field, comprising 25.3% of the total sensitivity (the amplitude of the sum of all local motion vectors) of HSN cells and 36.6% of HSE cells (Fig. 6a, b, Supplementary Fig. 11f, g). LMS of HSE cells was detected along the entire span of measured azimuth (−70° to 70°). This complex structure of LMS on the contralateral side is inherited from contralateral elements connected to HS cells[4,19].

In *Flp-shakB* flies the size of the spatial receptive fields of both HSN and HSE cells was largely reduced (Fig. 6c–f Supplementary Fig. 11b–i). While strong responses to ipsilateral horizontal motion along the equator were preserved in both HSN and HSE cells, HSN cells showed reduced responses in the fronto-dorsal and ventral areas, and HSE cells lost sensitivity to the motion in the contralateral field almost entirely (Fig. 6g–j). These changes suggest that LMS in these areas is a result of lateral interactions of HS with other tangential cells: likely horizontal-motion-sensitive ipsilateral neurons for HSN cells, e.g., CH cells[48], and contralateral horizontal-motion-sensitive neurons for HSE cells, e.g., H2, as shown in blow-flies[19]. Interestingly, direction-selective responses of HSE cells in mutant flies were enhanced in the fronto-dorsal area, indicating a potential role of gap junction-mediated inhibitory inputs in shaping the RF of these cells[4,48,49]. The described differences in RFs arise mainly through changes of their response to motion in their preferred direction, with a complete abolition of contralateral responses in HSE neurons (Supplementary Fig. 11b–e).

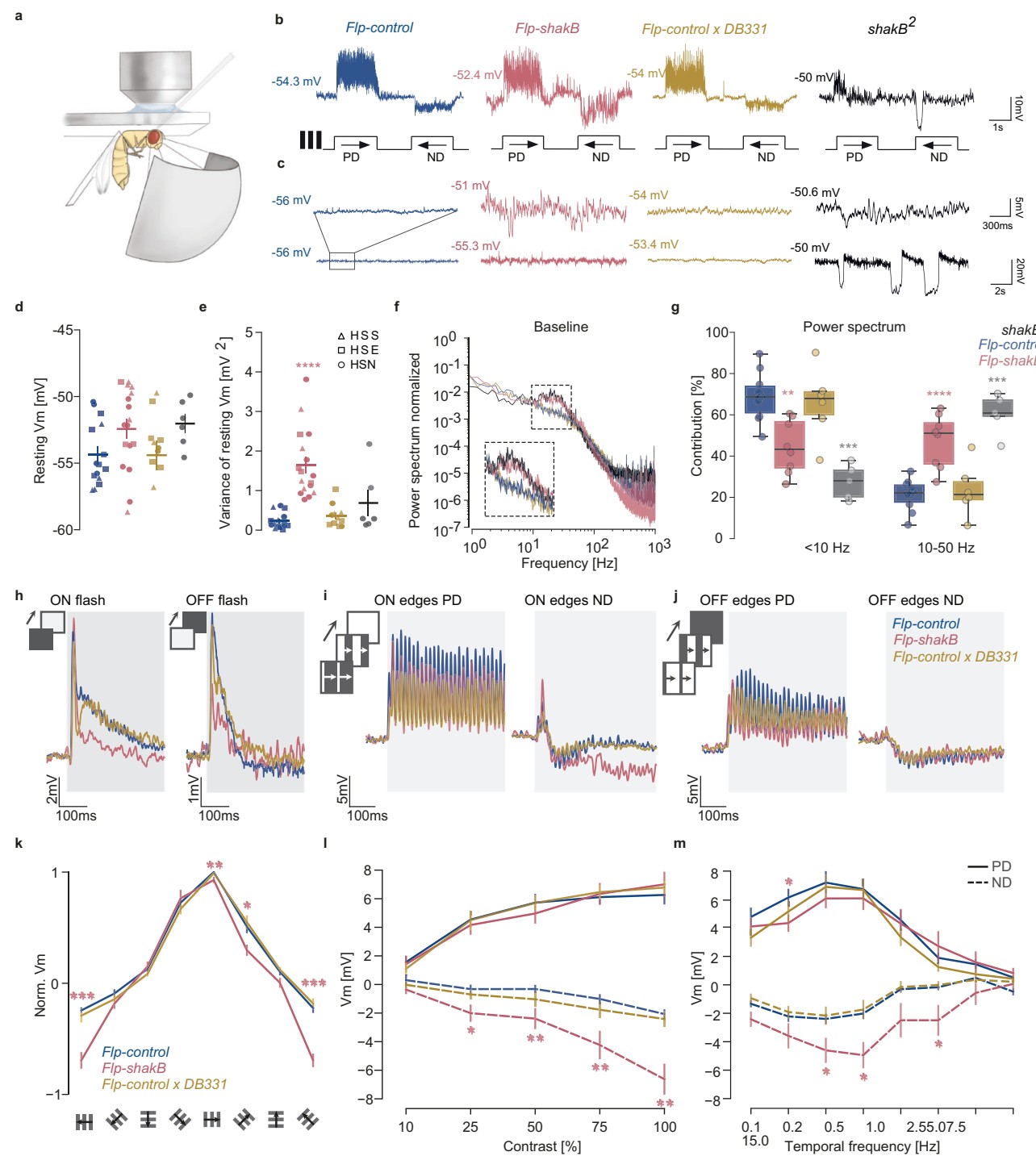

Altogether, our results suggest that the shape of HS cell RF is strongly modulated by lateral interactions via gap junctions. In particular, HSE neurons acquire back-to-front motion sensitivity in the contralateral area through a direct electrical coupling with contralateral horizontal-motion-sensitive neurons, enhancing responses to yaw-rotation tuning as suggested previously in blow flies[9,19,47].

## Dye coupling reveals the electrical connectivity network of HS cells

To provide further evidence for electrical coupling, we filled individual HS cells by injecting neurobiotin, a molecule that can diffuse through gap junctions. Subsequent visualisation of neurobiotin using fluorescently tagged streptavidin revealed that HS cells in *Flp-control* flies are electrically coupled to each other as well as to postsynaptic interneurons, motoneurons, and descending neurons (Fig. 7a, b), as shown in several previous studies[8,22,50–53]. HSN cells are strongly connected with two descending neurons, including DNp15 (DNHS1). Similar to blow flies, we also observed gap junctions between HS cells and neck motoneurons. Specifically, HSN and HSE form electrical connections with VCNM-like neurons (Ventral Cervical Nerve Motor Neuron), and HSS cells form electrical synapses with CNM-like neurons (Cervical Nerve Motor Neuron). Unlike wild-type flies, *Flp-shakB* flies showed little to no dye coupling between ipsilateral HS cells,

**Fig. 5 | Visual responses and passive membrane properties of HS cells lacking gap junctions. a** Setup for in vivo whole-cell patch-clamp recordings. **b** Example traces of membrane potential of HS cells during direction-selective responses and (**c**) while at rest. **d** Resting membrane potential of individual HS cells (mean ± SEM). Shapes depict distinct HS types: squares – HSN, triangles – HSE, circles – HSS (cell types were not identified for *shakB[2]*). **e** Variance of resting membrane potential (mean ± SEM). **f** Normalized power spectrum of baseline membrane potential in HS cells. **g** Contributions of low-range (<10 Hz) and mid-range (10–50 Hz) frequencies to the total power. Upper/lower limit and inner horizontal lines of the box plots represent the upper/lower quartile and median, respectively; whiskers indicate 1.5 interquartile range from upper/lower quartiles. **h** Average response traces of HS cells during the first 0.5 s after the onset of the ON/OFF flash. **i** Average response traces of HS cells to drifting ON edges moving in PD and ND. **j** Same as (**h**), but for OFF edges. **k** Normalized average voltage changes during 2 s presentation of square-wave gratings moving in 8 different directions (mean ± SEM). **l** Average

responses of HS cells during 2 s presentation of gratings with different contrast moving in PD and ND (1 Hz temporal frequency, mean ± SEM). **m** Average responses of HS cells during 2 s of presenting gratings moving with different temporal frequencies (mean ± SEM). **d**, **e**, **g**, **k**–**m** all the mutant genotypes were compared to *Flp-control* using two-sided Mann–Whitney *U*-test, *\*p* < 0.05, *\*\*p* < 0.01, *\*\*\*p* < 0.001. No asterisk: not significant. Number of cells recorded: **d**, **e** *Flp-control* = 15, *Flp-shakB* = 17, *shakB[2]* = 6, *Flp-controlxDB331* = 11; **f**, **g** *Flp-control* = 10, *Flp-shakB* = 8, *shakB[2]* = 5, *Flp-controlxDB331* = 6; **h** *Flp-control* = 10, *Flp-shakB* = 12, *Flp-controlxDB331* = 10; **i**, **j** Flp-control = 8, *Flp-shakB* = 7, *Flp-controlxDB331* = 5; **k** *Flp-control* = 10, *Flp-shakB* = 12, *Flp-controlxDB331* = 12; **l** *Flp-control* = 12, *Flp-shakB* = 9, *Flp-controlxDB331* = 9; **m** Flp-control = 12, Flp-shakB = 12, *Flp-controlxDB331* = 8. Exact *p*-values for each experiment are listed in Supplementary Data 1. In all panels, blue = *Flp-control*, magenta = *Flp-shakB*, orange = *Flp-controlxDB331*, and black = *shakB[2]*. Schematic drawings credited to Laura Burnett.

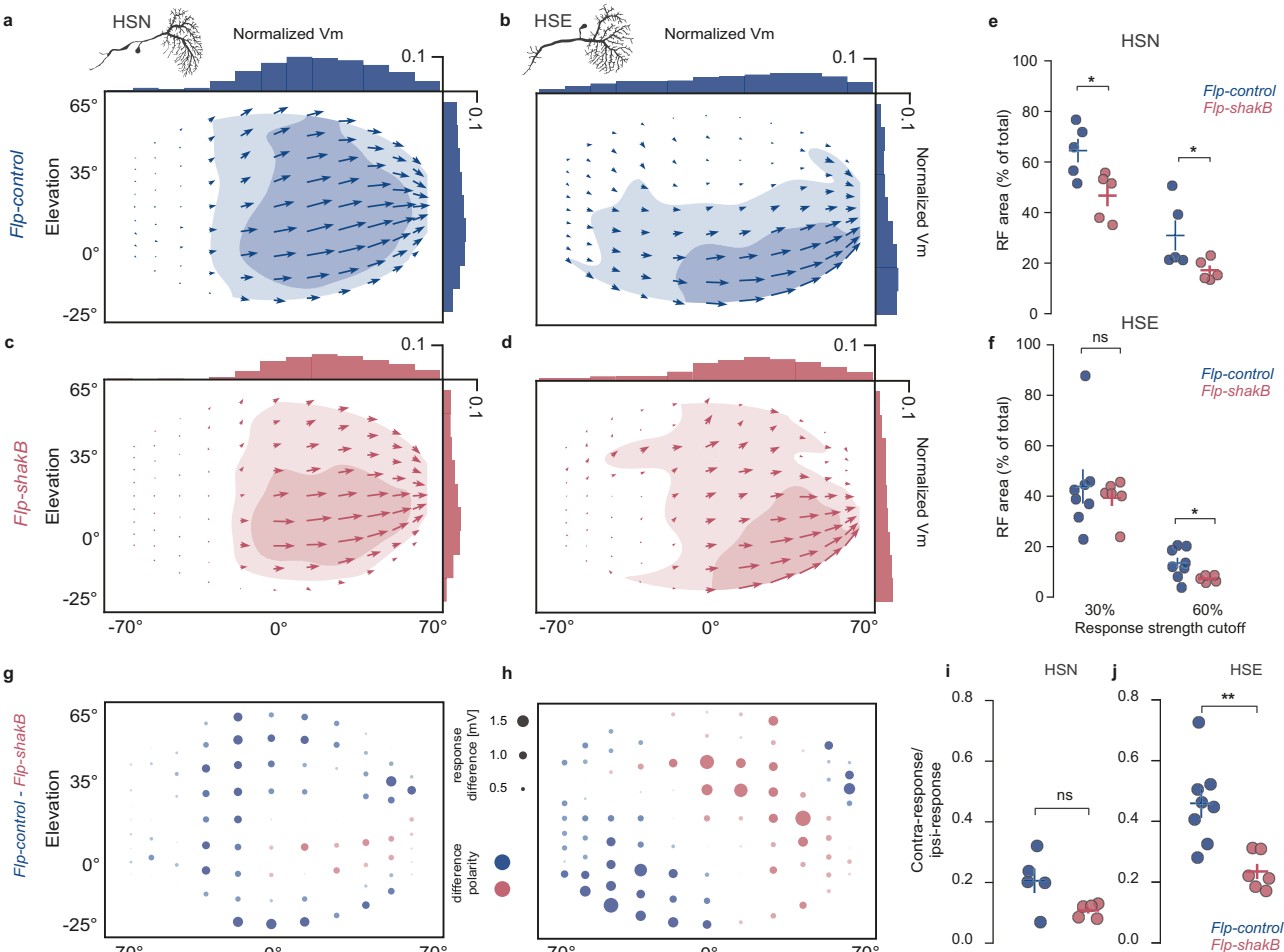

**Fig. 6 | Gap junctions shape receptive field structure. a–d** Spatial receptive fields reconstructed from the responses to local motion stimulus (see "Methods" section) for HSN (**a**, **c**) and HSE (**b**, **d**) cells in *Flp-control* and *Flp-shakB* flies. Light- and dark-shaded areas represent 30% and 60% of the maximal strength of the response for each cell type. **e** Size of the receptive field, corresponding to the shaded regions in (**a**, **c**) for individual HSN cells. Bars: mean ± SEM. **f** Same as in (**e**) for HSE cells. **g**, **h** Differences in the amplitude of absolute local motion sensitivity (LMS) in HSN and HSE cells between *Flp-control* and *Flp-shakB* flies. The area of the circle depicts the magnitude of the difference, and the colour represents polarity. Blue, LMS(*Flp-*

*control*) > LMS(Flp-shakB) and pink, LMS(Flp-shakB) > LMS(*Flp-control*) **i** Relative strength between the contralateral and ipsilateral visual fields of individual HSN cells. Bars: mean ± SEM. **j** Same as in (**i**) for HSE cells. **e**, **f** one-sided Mann–Whitney *U*-test, **i**, **j** two-sided Mann–Whitney *U*-test. *\*p* < 0.05, *\*\*p* < 0.01. Number of cells recorded: **a**, **c**, **e**, **g**, **i**: *Flp-control* = 5, *Flp-shakB* = 5; **b**, **d**, **f**, **h**, **j**: *Flp-control* = 8, *Flp-shakB* = 6. Exact *p*-values for each experiment are listed in Supplementary Data 1. Source data are provided as a Source Data file. In all panels, blue = *Flp-control*, magenta = *Flp-shakB*.

indicating that axo-axonal gap junctions between HS neurons are absent or severely perturbed (Fig. 7c, d). Additionally, coupling between HS cells and another class of ipsilateral LPTCs, presumably CH cells, was largely abolished in *Flp-shakB* flies.

While connections between LPTCs in *Flp-shakB* flies were abolished, electrical coupling between HS cells and postsynaptic descending neurons was largely preserved. This suggests that gap junctions formed between HS cells and postsynaptic neurons differ

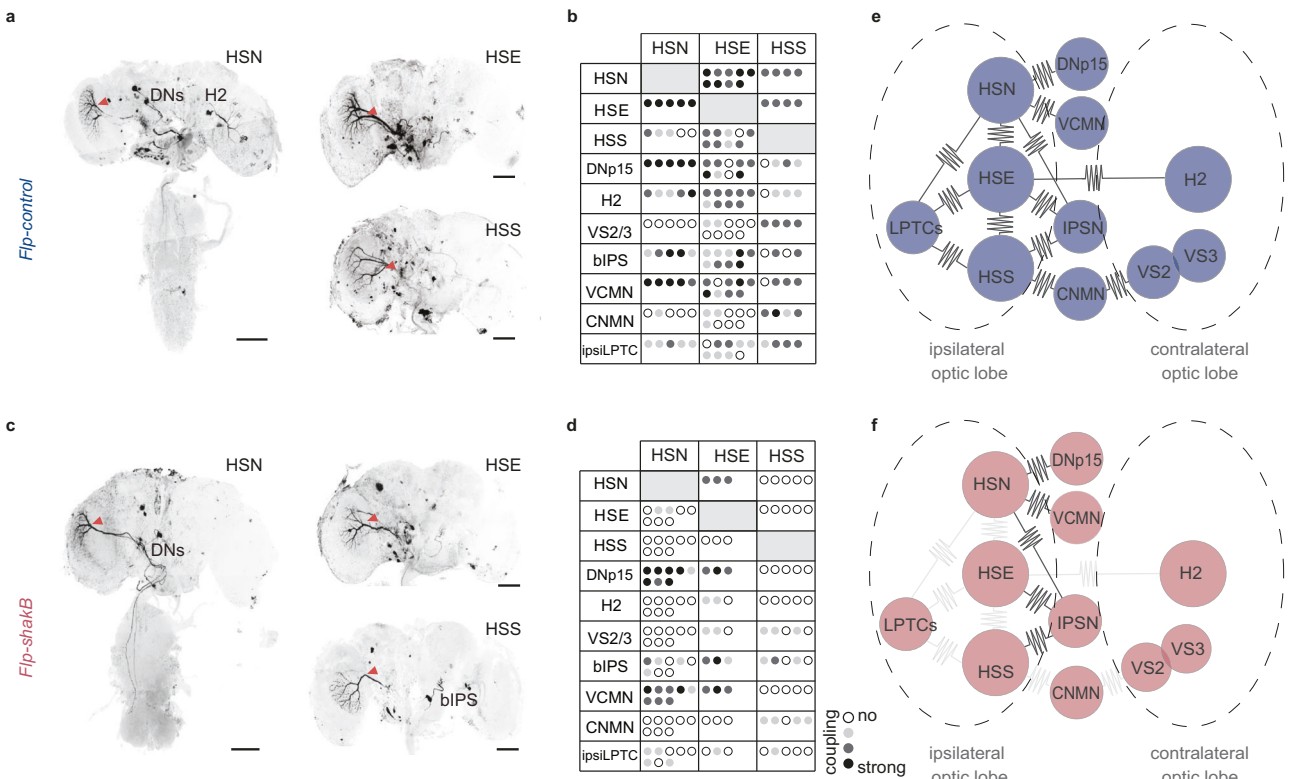

**Fig. 7 | Sensory-to-sensory, not sensory-to-motor electrical coupling is disrupted in *Flp-shakB*. a** Examples of neurobiotin injections into individual HS cells in *Flp-control* flies. Red arrow indicates an injected cell. **b** Quantification of dye coupling from individual HS cell dye fillings. The table summarises the most prominent dye coupling. Each dot represents an individual experiment. Dot opaqueness represents coupling strength inferred from the staining intensity. **c** Same as in (**a**), but for *Flp-shakB* flies. **d** Same as in (**b**), but for *Flp-shakB* flies. **e** Diagram summarising dye-coupling results of HS cells in *Flp-control*, a proxy of electrical coupling strength. **f** Same as in (**e**), but for *Flp-shakB* flies. **a**, **d** Scale bars: 50 µm. Number of brains analysed: **a**, **b** Flp_control = 9; **c**, **d** Flp_shakB = 8. Source data are provided as a Source Data file. In all panels, blue = *Flp-control*, magenta = *Flp-shakB*.

molecularly from those formed between LPTCs, and are likely composed of either undisrupted *shakB* isoforms or innexins other than *shakB*. Interestingly, unlike *Flp-shakB*, HS cells in *shakB[2]* mutant flies do not show any dye coupling[22] despite expressing an additional isoform (*shakB-RF*) (Fig. 4b).

Apart from ipsilateral electrical coupling, HS cells also formed extensive connections with neurons in the contralateral hemisphere. We identified that all three HS neurons were coupled to bIPS[15] that form a bridge between the tangential cells of the two hemispheres. This coupling is maintained in *Flp-shakB* flies. We also observed dye coupling between HSS and contralateral VS2 and VS3 neurons mediated via CNMNs[51]. HSE cells form electrical synapses with contralateral H2 cells[19]. The H2-HSE connection is substantially weakened in *Flp-shakB* flies and, therefore, can explain the difference in the structure of HSE receptive fields observed in *Flp-control* and *Flp-shakB* flies.

Receptive fields of HSE cells in *Flp-controlxDB331* flies did not show differences from wild-type flies (Supplementary Fig. 11j, k, m, o). This result is in line with the pattern of dye coupling (Supplementary Fig. 11l, n), showing that the gap junctions in HS cells are largely unaffected after cell-specific inactivation of *shakB*, despite a reduction of protein amount in LPTC terminals (Fig. 4f). However, as mentioned before, cell-specific inactivation worked completely on rare occasions in VS cells (Supplementary Fig. 10c), indicating that *shakB* can be disrupted in a cell-specific manner with GAL4 lines that express during early development.

Overall, the neurobiotin injections into HS cells revealed a significant reduction in the strength of their electrical coupling in *Flp-shakB* flies (Fig. 7e, f). However, this reduction was limited to connections between LPTCs, as shown by the reduction in contralateral

dye coupling with H2. Connectivity between HS cells and postsynaptic interneurons, descending neurons and motoneurons remained largely intact. Crucially, the reduction in contralateral electrical connectivity allows us to test the involvement of the LPTC-mediated behavioural role of gap junctions.

## Gap junctions coordinate binocular behavioural instructions

To study the behavioural role of gap junction-mediated binocular interactions, we compared the turning response of *Flp-shakB* and *Flp-control* flies to full-field and unilateral FtB and BtF rotation. Similar to wild-type flies, *Flp-control* flies turned in the direction of unilateral BtF and full-field rotation, and against the direction of unilateral FtB motion, demonstrating that the insertion of the genetic cassette does not affect the turning responses of the fly (Figs. 1eiii, 8aiii). However, the overall responses were stronger for *Flp-control* flies in comparison to *CantonS* flies, which can be attributed to the differences in the genetic backgrounds of these two lines. The relative contribution of smooth and saccadic turning in *Flp-control* was similar to that observed for wild-type flies (Figs. 1i left, 8ai *bottom*, Supplementary Movie 3). Similarly, *Flp-shakB* flies exhibited a strong turning response in the direction of full-field rotation (Fig. 8bi), showing remarkably robust optomotor responses despite modifications in the LPTC network, in line with afore-described observations in flies with silenced HS and VS neurons (Fig. 2c). Likewise, *Flp-shakB* flies turned with the direction of BtF motion, i.e., against the side of stimulation (Fig. 8bii, Supplementary Movie 4). However, in contrast to *Flp-control* flies that saccadically turned against the direction of FtB motion, *Flp-shakB* flies reversed their response direction and mode of action, turning on average smoothly with the direction of the stimulus, i.e., towards the

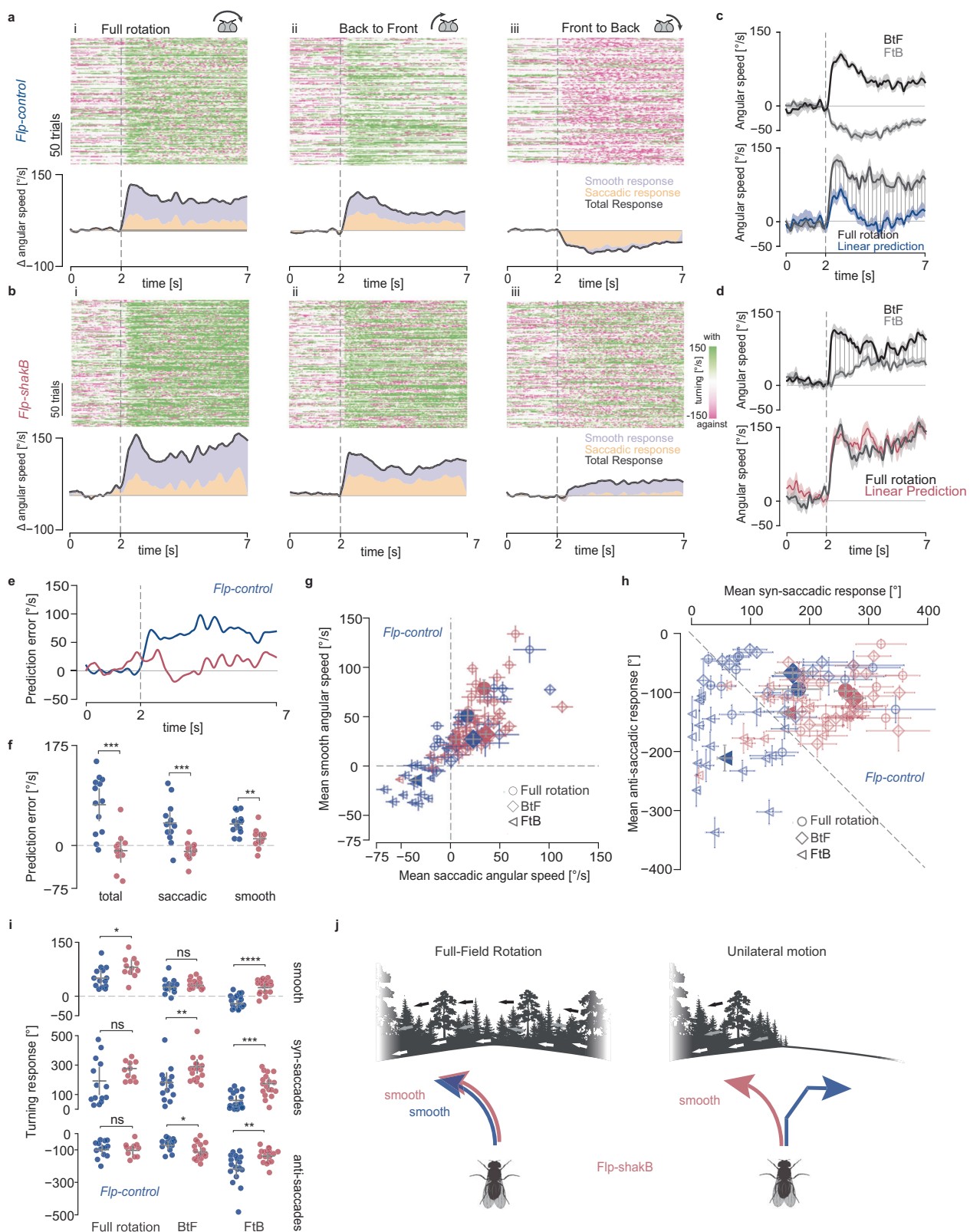

stimulated hemisphere (Fig. 8biii, Supplementary Fig. 12, Supplementary Movie 4). These behavioural differences were sex independent (Supplementary Fig. 13g–j), present without stimulating the binocular field (Supplementary Fig. 14a–d), and were independent of head movement feedback. Although head fixation increased the variability of responses, we observed quantitatively similar results (Supplementary Fig. 14e–h). Interestingly, anti-saccadic responses were dependent

on rearing temperature, and became more variable if animals were raised, but not kept at 25 °C, consistent with previous reports[32] (Supplementary Fig. 13a–f).

Consistent with wild-type flies, the optomotor response to full-field rotation exhibited significant differences from the linear prediction in Flp-control flies (Fig. 8c, e, f). However, in *Flp-shakB* flies, the optomotor response closely matched the linear prediction (Fig. 8d–f),

**Fig. 8 | Binocular control of walking behaviour requires gap junction connectivity. a** Turning response of *Flp-control* and *Flp-shakB* flies to (i) full-field, (ii) unilateral Back-to Front (BtF) and (iii) unilateral Front-to Back (FtB) rotation. *Top.* Angular speed raster plots showing 200 randomly selected trials. Each row corresponds to one trial, each column to one frame (-16 ms). *Bottom.* Stacked plot showing mean angular speed and the contribution of smooth and saccadic turning. **b** Same as in (**a**) but for *Flp-shakB* flies. **c** *Top.* Angular speed of *Flp-control* flies in response to FtB and BtF rotation (mean ± SEM). *Bottom.* Comparison of the predicted and the actual mean angular speed in response to full-field rotation for *Flp-control* flies. Shaded region shows the difference between predicted and actual response. **d** Same as (**b**) but for *Flp-shakB* flies. **e** Time series of mean prediction error. **f** Mean prediction error across the trial period per fly. Bars: mean ± SEM. **g** Summary plot of the mean of smooth and saccadic angular speed across experiments. Empty and filled markers show the mean per fly and mean ± SEM for each genotype, respectively. **h** Cumulative angular displacement for syn-saccadic and anti-saccadic turns in one trial. Empty and filled markers show the mean per fly and mean ± SEM for each genotype, respectively. **i** Mean of smooth (top), syn-saccadic (middle) and anti-saccadic (bottom) cumulative angular displacement per trial. Bars: mean ± SEM. **j** Schematic summary. Contralateral HSE-to-H2 electrical connectivity mediates the change between optomotor and saccadic counter-response. Disrupting the contralateral connectivity disrupts this finely tuned bilateral course control, disrupting the counter-saccadic response. **f, i** two-sided Mann–Whitney *U*-test was applied, *$p < 0.05$, **$p < 0.01$, ***$p < 0.001$, ****$p < 0.0001$. No asterisk: not significant. Number of flies: *Flp-control* = 13, *Flp-shakB* = 11. Exact *p*-values for each experiment are listed in Supplementary Data 1. Source data are provided as a Source Data file. In all panels, blue = *Flp-control*, magenta = *Flp-shakB*. Schematic drawings credited to Laura Burnett.

indicating that binocular interactions facilitated by gap junctions are involved in the non-linear summation of motion cues from both halves of the visual field. Importantly, the linear combination of ipsi- and contralateral inputs is primarily attributed to changes in smooth and saccadic responses, as evidenced by the reduction in prediction error for both types of movements (Fig. 8f). Notably, in *Flp-shakB* flies subjected to front-to-back (FtB) motion, smooth turning in the direction of the stimulus was enhanced, while saccadic counter turns were diminished (Fig. 8g–i, Supplementary Fig. 12b), suggesting that flies cannot properly discriminate between translation and rotation, which requires the integration of bilateral optic-flow motion signals. Inspired by a recent study describing a circuit mechanism that enhances the discriminability of bilateral interactions[15], i.e., translation and rotation, we reproduced the model to ask whether bilateral gap junctions between HS and H2 cells affect their selectivity to translational or rotation stimuli (Supplementary Fig. 15a). Our modelling results show that gap junctions increase the selectivity of neurons to rotational optic-flow in all cells of the circuit, similar to specific perturbations of their chemical output (Supplementary Fig. 15b, c).

In summary, the disruption of electrical binocular coupling has a profound impact on the observed behaviour. It enhances smooth and saccadic syn-directional turning while significantly reducing saccadic counter turns. As a result, the fly's binocular integration undergoes a shift in complexity. It transitions from non-linearly integrating monocular information for guiding course corrections to linearly summing binocular cues. The result is a reduced range of locomotor behaviour, with the fly showing a similar turning response to all three types of movement (Fig. 8j).

## Discussion
The intricate wiring of the nervous system relies on a variety of cellular mechanisms to facilitate communication between neurons. One such mechanism is electrical coupling by gap junctions, specialised channels between adjacent cells. Their computational roles have been proposed to contribute to the synchronisation or desynchronization of network activity[54–56], improve sensory information processing[25], act as computational switches[57], and be relevant for cell-intrinsic stability of their membrane potential[22]. However, the implications of such computations for behaviour remain largely unknown. Here we present evidence that specific gap junction connectivity plays a decisive role in sensorimotor transformations. First, we uncovered the richness and precision of optic-flow-based navigation by expanding the classical optomotor paradigm (Fig. 1a–e). We show that flies change their locomotion depending on the stimulus properties, i.e., unilateral and bilateral wide-field motion, changing the nature and direction of their turning responses (Fig. 1e, f). These changes underlie a non-linear visuomotor transformation since the summation of each unilateral stimulus can't account for the binocular reaction (Fig. 1g, h, j). This process is mediated partly by a subset of LPTCs (Figs. 2, 3), namely HS cells, known to be connected via gap junctions contralaterally to H2[19].

The *shakB* gene has been identified as a key player in forming these gap junction channels[22] (Fig. 4), and several studies have implicated electrical connectivity in the LPTC network with improved efficiency and accuracy of optic-flow estimations[19,24,25,27,58]. Yet, these studies could not link the intricate computations experimentally to behaviour. By establishing and thoroughly characterising a new *shakB* mutant line (Figs. 4–8, Supplementary Figs. 6–14), we could link the behavioural deficits observed by silencing LPTCs (Fig. 2) with the loss of hetero-lateral electrical connections between HS and H2 cells (Figs. 6–8). These results expand the functional roles mediated by gap junctions and implicate them with non-linear operations with a decisive role in animal course control. Taken together, our results show that the animal's reflexive behaviour to visual motion is finely tuned to bilateral heterogeneity in global motion that flies may encounter while navigating natural environments.

### A repertoire of course control behaviours
Flies' remarkable course control behaviours have been postulated to reside largely in the algorithms embedded in the connectivity among LPTCs, the so-called "cockpit of the fly"[5,18,20]. Yet, the relevance of the connectivity between the LPTCs is poorly understood, partly because the behavioural repertoire has been restricted mainly to the classical optomotor response – a reflexive response that does not require the complexity seen in these circuits. Here, by employing closed-loop freely-walking behavioural assays (Fig. 1a–d), we have identified strikingly different visuomotor behaviours, opening the path for a dissection of the underlying neuronal mechanisms. Specifically, we showed that the turning direction and behavioural actions (saccadic or smooth) depend on the bilateral optic-flow's properties. These results, at first glance, appear counterintuitive because previous studies have shown that syn-saccades are triggered when smooth turning is insufficient to counteract retinal slip[59]. But anti-saccadic responses, as seen during FtB motion, would increase retinal slip. This shows that the fly reacts differently to diverse optic-flow patterns beyond heading stabilisation. For example, a strong unilateral FtB motion could be elicited, e.g., during an approach to a wall at a steep angle. In such a scenario, an optomotor response would instruct the fly to turn into the wall, whereas a counter-saccadic response reflexively corrects a collision course by forcing a counter-turn (Fig. 8j).

Our observations differ from recent reports of anti-directional, saccade-independent turning in response to long-lasting, high-contrast visual stimuli[32]. While we did observe a few anti-directional responses to full-field rotation in our experiments, they were mainly saccadic. Yet, the average response of the fly was in the direction of motion across the entire trial length, composed of smooth turns and syn-saccades. Strong and immediate anti-saccadic reactions were observed specifically for FtB motion. This difference may arise due to our closed-loop configuration, suggesting that the gain of visual feedback affects the turning response of the fly. Interestingly, a

previous study[60] used similar stimulus paradigms during tethered flight but did not observe a counter-saccadic response to FtB motion. This suggests that flies may interpret optic-flow based on their locomotor state, i.e., either walking or flying. Additionally, this might also reflect adaptations to the location of stimulus presentation. Whereas previous tethered flight paradigms used cylindrical arenas, presenting optic-flow stimuli to the frontal visual field, our study presents visual stimuli from above, targeting the dorsal visual field. Recent work has shown that moths have different flight stabilisation strategies depending on whether the optic-flows are presented from above or below[61], consistent with changes in natural image statistics across elevation. Adaptation of such panoramic natural image statistics has been recently shown to occur in the mammalian retina as well[62], indicating that these adaptations are a general phenomenon of visual systems. Thus, local heterogeneities in global optic-flow play a critical role in controlling behaviours.

## Molecular diversity of gap junction channels

The *shakB* gene in *Drosophila* produces multiple isoforms (Fig. 4b, Supplementary Fig. 6b), resulting in distinct protein variants. These isoforms exhibit specific temporal and spatial expression patterns, suggesting their involvement in the formation of different types of gap junction channels. However, the function of this molecular diversity remains poorly understood. This becomes evident when observing the phenotypic discrepancies between the *Flp-shakB* and *shakB[2]* mutant flies. We observed systematic variations in dye coupling, hyperpolarization patterns, and the strength of responses in different neuron types. Most strikingly, whereas HS cells in *shakB[2]* mutant flies exhibit significant differences in their visual responses and a complete absence of dye coupling[22], HS cells in *Flp-shakB* flies largely preserve their visual response properties (Fig. 5) and show very specific changes in their electrical coupling (Fig. 7). The potential sources of these discrepancies include (i) a reduced penetrance of the *Flp-shakB* allele, which was evident in female flies, probably because *shakB* is located on the X chromosome. We circumvented this problem by performing all our experiments on male flies. (ii) Additional mutations accumulated or generated during EMS-induced mutagenesis in the *shakB[2]* mutant, or (iii), putatively dominant-negative effects of the additional *shakB-RF* isoform that remains intact and is expressed at a relatively high level throughout the development and during adult stage (Fig. 4b, Supplementary Fig. 6b). Given the difficulties in ruling out the underlying differences in the *shakB[2]* mutant, new genetic approaches similar to our FlpStop approach are needed to understand the specific roles of gap junctions.

## Electrical connectivity logic

Why would the disruption of one innexin gene differentially affect electrical synapses formed by HS cells? Given that the shakB gene produces multiple isoforms, the most obvious explanation is that HS cells ummari different protein variants to form gap junctions with distinct synaptic partners. Several lines of evidence corroborate this hypothesis. Different isoforms of the *shakB* form various channels with distinct properties that shape a unique current flow pattern in a given neural circuit[63]. For example, heterotypic channels were demonstrated to form rectifying electrical synapses in the giant fibre system, while homotypic channels exhibit symmetrical voltage responses[63]. Specifically, the giant fibre neuron was shown to express the ShakB(N + 16) to form gap junction channels with postsynaptic motoneurons that express the ShakB(Lethal) variant. In these experiments, only ShakB(N + 16) expression and not ShakB(N) rescued the connectivity phenotype in the giant fibre system in a *shakB[2]* background[63]. This suggests that coupling between LPTCs or to their downstream DNs and motor neurons could be similarly composed of non-rectifying and rectifying electrical synapses. A thorough genetic and physiological analysis would be required to unravel the exact molecular composition and physiological properties of ShakB electrical synapses in tangential cells.

## Disrupting *shakB*

Previous cell-specific knock-down attempts by RNAi[22] did not significantly reduce the electrical coupling formed by LPTCs. These technical challenges are rather puzzling, considering that the predicted lifespan of gap junction proteins in *Drosophila* is several hours to several days[64,65]. However, the analysis of several driver lines for LPTC-specific inversion of the FlpStop cassette shows that this prediction may not be accurate for the ShakB protein in all cells. The induction of gene disruption in LPTCs at around developmental stage P9 results in a stronger reduction of protein signal in adult animals than induction at later stage P12 (Fig. 4e, f). Despite strong reduction in the protein amount induced by early and unspecific DB331 lines[21], we observed only on rare occasions a full disruption of gap junctions In VS cells (Supplementary Fig. 10c). This effect is likely due to an earlier onset of GAL4 in these cells. Importantly, a previous study has shown that a weak pan-neuronal expression of ShakB(N + 16) can rescue electrical connectivity in the giant fibre system[63], or in our case, a severe reduction in the amount of ShakB did not result in a loss of electrical synapses formed by LPTCs. This shows that even a small amount of protein can form functional gap junction channels. Thus, future work aimed at disrupting electrical synapses in a cell-specific manner will also need to consider the exact onset of the driver lines during development.

## Visual response properties in the absence of ShakB

As previously reported for *shakB[2]* flies, we also observed β-oscillations in the resting membrane potential in *Flp-shakB* flies[22] (Fig. 5f, g). Interestingly, we show that these changes are cell-intrinsic by correlating the oscillations of membrane potential with the absence of dye coupling after LPTC-specific disruption of *shakB* (Supplementary Fig. 10c). This indicates that gap junctions formed between HS cells facilitate the dissipation of cell-intrinsic noise that otherwise could interfere with signals arriving from presynaptic cells. Contrary to a previous report on *shakB[2]* mutant flies[22], the main direction-selective properties, as well as the velocity and contrast tuning, are not affected in *Flp-shakB* flies (Fig. 5k–m). The only observed difference was an increase in hyperpolarization amplitude in response to gratings moving in the null direction for ON edges (Fig. 5i–m), indicating changes in the presynaptic circuitry involving Lpis[66]. The most striking effect is observed in the structure of the receptive fields of HS cells. In wild-type animals, HSE cells are depolarized by contralateral motion in the preferred direction[8]. This property was absent in *Flp-shakB* flies (Fig. 6, Supplementary Fig. 11c, e). In line with previous work[19] and intracellular dye fillings described above, this contralateral input arrives from back-to-front-motion-sensitive H2 neurons. This binocular connectivity has been proposed to be required for accurate interpretation of optic-flows dominated by horizontal components, such as yaw rotation and translation[9,19,47]. Therefore, the loss of the HSE-to-H2 connection, as well as perturbations in the larger HS-H2 network[15], would restrict the analysis of the optic-flow to individual hemispheres, limiting the ability of the fly to differentiate global optic-flow patterns and perform adequate steering behaviours. This can be relevant for interpreting local irregularities in optic-flow patterns[4]. For example, when navigating in the natural environment, a global displacement caused by a gust of wind should elicit a compensatory change in the direction of motion, whereas local motion cues caused by approaching objects in the visual periphery would require a corrective turn in the opposite direction to avoid a collision (Fig. 8j).

## Course control in the absence of ShakB

A rigorous characterization of the stimulus–behaviour mapping is required to define the computation being performed by the LPTC

network and new genetic tools are needed to understand their neuronal implementation[67]. Thus, we performed a behavioural analysis comparing two sets of FlpStop flies: one with the cassette in the disruptive orientation and the other in the non-disruptive orientation. These flies share an identical genetic background, and any behavioural discrepancies can be attributed solely to the disruption of the *shakB* gene. Remarkably, we observed a similar behavioural phenotype to that observed in our silencing experiments targeting LPTCs (Figs. 2, 3), suggesting that the mechanism behind bilateral visual control relies on the electrical coupling between these neurons. The extent of behavioural disruption in *Flp-shakB* appeared to be more pronounced compared to our silencing experiments, where we observed an initial counter-saccadic turn in response to FtB stimuli (Fig. 2b). This discrepancy is likely attributable to the silencing approach using Kir2.1, strong inward rectifying channels that hyperpolarize the membrane potential to the potassium reversal potential, an effect that is dependent on expression levels. In HS cells, this silencing approach has been shown to leave a residual direction-selective activity[13]. Interestingly, in both cases, the linear sum of the behavioural response to unilateral stimuli matches the binocular optomotor response (Figs. 2c, 8d, e), an equivalence that is absent in wild-type and *Flp-control* flies due to a striking non-linear interaction between two hemispheres (Figs. 1h, 8c, e). The enlargement of the ipsilateral RF in *Flp-shakB* compared to *Flp-control* is unlikely to be involved. Our behavioural arena activates the dorsal half of HSE in *Flp-control* animals. Increasing the ipsilateral RF size, as in *Flp-shakB* (Fig. 6a–f), should increase the strength of the counter-directional response to unilateral FtB motion. We observed the opposite, suggesting that the contralateral input is necessary for eliciting the counter-directional responses. This is supported by our H2 block experiments in which the anti-saccadic response is strongly diminished (Fig. 3). Consequently, our results provide compelling evidence implicating the HS-H2 network[15] in the proper selection of smooth and saccadic turns during navigation. Further evidence for such network interactions can be taken from our modelling results, which show that disrupting chemical or electrical connectivity in an experimentally informed HS-H2 network reduces the discriminability of bilateral interactions across the circuit (Supplementary Fig. 15a, b). These findings underscore the functional significance of gap junctions within the insect nervous system, connections that have been predominantly overlooked in connectomic analysis due to the challenges associated with their visualisation. Moreover, they suggest that gap junctions can facilitate non-linear operations that play a decisive role in animal course control rather than passively averaging neighbouring signals, as suggested through careful experiments and of the electrical HS-H2 interaction in blow flies[19]. However, a significant question remains unanswered. What are the underlying circuit motifs and biophysical implementations underlying this non-linear, gap junction-mediated binocular interaction that modulates behavioural output? In flies, pre-motor descending neurons (DNs) connect the brain to the ventral nerve cord and are responsible for generating appropriate visuomotor commands. DNs that receive inputs from LPTCs have been shown to be involved in walking and flying. However, the picture is further complicated by the presence of interneurons modulating LPTCs and DNs and recent finding showing that DN act in coordination to control behaviour[68]. Specifically, a recent study showed that inhibitory inputs from an interhemispheric interneuron, bIPS, to a descending neuron, DNp15, are required to maintain path straightness[15]. This particular subnetwork is unlikely to be involved in the counter-saccadic behaviour described in our work, as DNp15 innervates the neck and the halters[69]. It is more likely that other descending pathways connected to HS or H2 cells (Supplementary Fig. 15d)[70] are required, such as those involved in saccadic turns[71]. Nonetheless, it highlights the challenges involved in understanding the full circuit involved in transforming optic-flow patterns into specific motor outputs.

More than 50 years ago, the optomotor response[1] laid the foundation for subsequent research on the neuronal implementations of motion vision[2]. Likewise, we believe that the expansion of the known repertoire of course control behaviours, as done in this study, is a requirement for a comprehensive understanding of the neuronal mechanisms of vision in flies[67,72].

## Method

### Detailed fly genotypes used

Figure 1c–m, Supplementary Figs. 2b–j, 3a–j
*CantonS*
Figure 2a–c, f–j, Supplementary Fig. 4a, c, d
*w+; tsh-GAL80/+;VT058487-GAL4/10XUAS-IVS-eGFPKir2.1*
Supplementary Fig. 4e
*w+; tsh-GAL80/R23C12-p65.AD;R32A11-GAL4.DBD/10XUAS-IVS-eGFPKir2.1*
Figure 2d–j, Supplementary Fig. 4b, c, d
*w+; tsh-GAL80/+;pBDPGAL4Uw/10XUAS-IVS-eGFPKir2.1*
Figure 3a, b, g–k, Supplementary Fig. 5c, d–h
*w+; tsh-GAL80/+;VT058487-GAL4/UAS-shi[ts1]*
Figure 3e, f, g–k, Supplementary Fig. 5b, d–h
*w+; R23C12-p65.AD/+;R32A11-GAL4.DBD/UAS-shi[ts1]*
Figure 3c, d, g–k, Supplementary Fig. 5a, d–h
*w+; +;UAS-shi[ts1]/+*
Figure 4c, e
*shakB[2]; 10XUAS-IVS-mCD8::GFP/+; VT058487-GAL4/+*
*shakB[FlpStopND]; 10XUAS-IVS-mCD8::GFP/+; VT058487-GAL4/+*
*shakB[FlpStopD]; 10XUAS-IVS-mCD8::GFP/+; VT058487-GAL4/+*
Figure 4d, f, Supplementary Fig. 7a, b
*shakB[2]; +;+*
*shakB[FlpStopND], w+; +;+*
*shakB[FlpStopD], w+; +;+*
Figure 4e, f, Supplementary Fig. 5c
*shakB[FlpStopND], DB331-GAL4; 20XUAS-FLPG5.PEST/+;+*
*shakB[FlpStopND], w+; 20XUAS-FLPG5.PEST/+; VT058487-GAL4*
Supplementary Fig. 8
*w1118; 20XUAS-SPARC2-I-mCD8::GFP/VT058487-p65.AD, 20XUAS-IVS-PhiC31; VT000343-GAL4.DBD/+*
*shakB[2]; 20XUAS-SPARC2-I-mCD8::GFP/VT058487-p65.AD, 20XUAS-IVS-PhiC31; VT000343-GAL4.DBD/+*
*shakB[FlpStopD]; 20XUAS-SPARC2-I-mCD8::GFP/VT058487-p65.AD, 20XUAS-IVS-PhiC31; VT000343-GAL4.DBD/+*
Supplementary Fig. 9
*w1118, UAS-myrGFP.QUAS-mtdTomato-3xHA; trans-Tango; R81G07-GAL4/+*
*shakB[2] UAS-myrGFP.QUAS-mtdTomato-3xHA; trans-Tango; R81G07-GAL4/+*
*shakB[FlpStopD] UAS-myrGFP.QUAS-mtdTomato-3xHA; trans-Tango; R81G07-GAL4/+*
Figure 5b–m, Supplementary Fig. 10d–i
*shakB[2]; +;+*
*shakB[FlpStopND], w+; +;+*
*shakB[FlpStopD], w+; +;+*
*shakB[FlpStopD], DB331-GAL4; 20XUAS-FLPG5.PEST/+;+*
Figure 6a–j, Fig. 7a–f, Fig. 8a–i, Supplementary Fig. 11b–I, Supplementary Fig. 12a–e
*shakB[FlpStopND], w+; +;+*
*shakB[FlpStopD], w+; +;+*
Supplementary Fig. 10c, Supplementary Fig. 11j–o
*shakB[FlpStopND], DB331-GAL4; 20XUAS-FLPG5.PEST/+;+*

### Fly husbandry

*Drosophila melanogaster* was reared on a standard cornmeal-molasses agar medium at 18 °C and 60% humidity and kept on a 12 h light/12 h dark cycle. Experiments were performed with 2–6-day-old flies. Due to reduced penetrance and high phenotypic variability in female mutant

## Table 1 | Fly stocks and sources

| Genotype | Source | Stock number |
|---|---|---|
| *Canton S (wild-type)* | BDSC | RRID: BDSC_64349 |
| *shakB[2]; +; +* | Augustin Hrvoje | |
| *shakB[FlpStopND], w+; +; +* | this paper | |
| *shakB[FlpStopD], w+; +; +* | this paper | |
| *w1118; +; VT058487-GAL4* | Vienna Drosophila Resource Center (VDRC) | |
| *w1118; +; 10XUAS-IVS-eGFPKir2.1/TM6B* | Eugenia Chiappe | |
| *DB331-GAL4; +; +* | Alexander Borst | |
| *w+;tsh-GAL80/Cyo;VT058487-GAL4/+* | this paper | |
| *w+; R23C12-p65.AD; R32A11-GAL4.DBD* | Eugenia Chiappe | |
| *w+; +; UAS-shi[ts1]* | BDSC | RRID:BDSC_44222 |
| *y1, w*, UAS-myrGFP.QUAS-mtdTomato-3xHA; trans-Tango; +* | BDSC | RRID:BDSC_77124 |
| *w1118; +; R81G07-GAL4* | BDSC | RRID:BDSC_40122 |
| *w*;10XUAS-IVS-mCD8::GFP; +* | BDSC | RRID:BDSC_32186 |
| *w1118; 20XUAS-FLPG5.PEST; +* | BDSC | RRID:BDSC_55806 |
| *w+; 20XUAS-SPARC2-I-mCD8::GFP/Cyo; +* | BDSC | RRID:BDSC_84147 |
| *y1, w*; 20XUAS-IVS-PhiC31; +* | Thomas Clandinin | |
| *w1118; VT058487-p65.AD; +* | BDSC | RRID:BDSC_74086 |
| *w1118; +; VT000343-GAL4.DBD* | BDSC | RRID:BDSC_75194 |

flies, we used only male flies. A list of all flies used in this work can be found in Table 1.

### Generation of shakB[FlpStop] transgenic flies

*pFlpStop-attB-UAS-2.1-tdTom* (addgene #88910) donor plasmid was injected into shakB[MI15228] flies together with φC31 integrase-expressing transgene (BestGene Inc.). The orientation of the FlpStop cassette was identified using primers MiL-F, FRTspacer_5p_rev, and FRTspacer_3p_for from[39].

### Protein isolation and quantification of ShakB

To extract insoluble protein fraction, approximately 150 brains of 3–5-day-old male flies were homogenised in 300 µl of extraction buffer (20 mM Tris pH 7.6, 50 mM NaCl, 1% Triton X-100, 1Xhalt Protease inhibitor Cocktail (Thermo Fisher, 78429)), and incubated for 30 min on ice. Homogenates were centrifuged for 60 min at $15 \times 1000\,g$ in 4 °C. Supernatant was discarded, and the remaining pellet was used for the isolation of insoluble proteins. For that, the pellet was resuspended in the SDS extraction buffer (50 mM Tris pH 7.6, 5 mM EDTA, 4% SDS), and incubated at 95 °C for 10 min. Supernatants were collected after centrifugation for 10 min at $15 \times 1000\,g$ at room temperature and were used for protein quantification with BCA protein assay (Thermo Fisher, 23225).

For western blot analysis, 10 µg of each protein sample were mixed with Laemmli's buffer, boiled for 5 min, and subjected to SDS-PAGE using 4–20% TGX Stain-Free Precast Gels (Bio-Rad, 4568095). Gels were activated by UV exposure for 2 min using a Bio-Rad Chemidoc MP imager. Proteins were transferred to LF PVDF membranes (Bio-Rad, 1620263) using a Transblot Turbo apparatus (Bio-Rad). The membrane was incubated in EveryBlot Blocking Buffer (Bio-Rad, 12010020) and immunoblotted following standard protocols.

The following antibodies were used for the western blot analysis: anti-ShakB (1:3000, Innovagen AB), IRDye 800 CW goat anti-rabbit (1:15000, LI-COR Biosciences). Total lane signal was detected using the

Stain-Free Blot application and the ShakB signal was detected using IRDye 800 CW application (Bio-Rad Chemidoc MP imager).

The immunoblots were repeated three times. ShakB signal was quantified and normalised to the total lane signal using Imagelab 4.1 (Bio-Rad).

### Proteomic analysis

For protein sample preparation, brains of 2–5 days-old flies were dissected. All samples (3 genotypes, 4 replicates; 10 or 6 pooled dissected fly brains per sample for replicate 1 and replicates 2, 3, and 4, respectively) were processed in 4 replicate-specific batches with the iST-NHS kit from PreOmics GmbH using the standard manufacturer's protocol with the following modifications: samples were lysed in 100 µL iST-LYSE buffer, boiled at 95 °C for 5 min, then sonicated for 10 cycles of 30 s each on/off in a Bioruptor plus (Diagenode) in presence of 50 mg Protein Extraction beads (C20000021, Diagenode); samples were trypsin digested for 2 h 30 min, then labelled with TMT-6plex (ThermoScientific, lot # VI307213) according to the manufacturer's instructions (one bridging channel, consisting of a mix of all 20 samples, was included in each combined TMT sample). Combined TMT samples were dried, re-dissolved in 45 µL 100 mM NH4OH, then loaded onto an ACQUITY UPLC BEH C18 column (130 Å, 1.7 µm, 2.1 mm × 150 mm, Waters) on an Ultimate 3000 UHPLC (Dionex) and fractionated into 24 fractions by High pH Reversed-Phase chromatography (solvent A: deionized water + 10 mM NH4OH; B: 90% LC-grade Acetonitrile + 10 mM NH4OH; flow: 0.15 ml/min; gradient: 0–4 min = 1% B, 115 min = 25%, 140 min = 40%, 148–160 min = 75% followed by re-equilibration at 1% B). Fractions were combined at mid-gradient, re-dissolved in 50 µL iST-LOAD and sent for MS analysis.

All samples were analysed by LC-MS/MS on an Ultimate 3000 nano-HPLC (Dionex) coupled with a Q-Exactive HF (Thermo Fisher Scientific). Spectral data were acquired on data-Dependent Acquisition (Full MS/dd-MS2); chrom. peak width (FWHM) 20 s, MS1: 1 microscan, 120,000 resolving power, 3e6 AGC target, 50 ms maximum IT, 380 to 1500 m/z, profile mode; up to 20 data-dependent MS2 scans per duty cycle, excluding charges 1 or 8 and higher, dynamic exclusion window 10 s, isolation window 0.7 m/z, fixed first mass 100 m/z, resolving power 60,000, AGC target 1e5 (min 1e3), maximum IT 100 ms, (N)CE 32.

Acquired raw files were searched in MaxQuant[73] (1.6.17.0) against a *Drosophila melanogaster* fasta database downloaded from Uni-ProtKB. Fixed cysteine modification was set to C6H11ON (+113.084064). Variable modifications were set to include Acetyl (protein N-term), Oxidation (M), Gln->pyroGlu, Deamidation (NQ) and Phospho (STY). Match-between-runs and second peptide search were set to active. All FDRs were set to 1%. The output "evidence.txt" files were then re-processed in R using in-house scripts. Briefly, MS1 parent intensities were normalised per fraction; evidence reporter intensities were corrected using the relevant TMT lot's purity table, then normalised using the Levenberg-Marquardt procedure and scaled to normalised parent MS1 intensity. Peptidoform intensity values were $\log_{10}$ transformed. The TMT/replicate-specific batch effect was corrected using Internal Reference Scaling, then values were re-normalized (Levenberg-Marquardt procedure). Protein groups were inferred from observed peptidoforms, and, for each group, its expression vector across samples was calculated by averaging the $\log_{10}$ intensity vectors across samples of individual unique and razor peptidoform, scaling the resulting relative profile vector to an absolute value reflecting the intensity level of the most intense peptidoform according to the best flyer hypothesis (phospho-peptides and their unmodified counterpart peptide were excluded). Peptidoform and protein group $\log_2$ ratios were calculated per replicate to the corresponding control (*Flp-control*) sample. Statistical significance was tested with the limma package, performing both a moderated *t*-test and an *F*-test (limma package) for all other genotypes against the

*Flp-control* genotype. The Benjamini–Hochberg procedure was applied to compute significance thresholds at various pre-agreed FDR levels (up to 10, FDR thresholds calculated globally for the *F*-test). Regardless of the test, protein groups with a significant *P*-value were deemed to be regulated if their absolute $\log_2$ ratio was larger than the 90% least extreme individual control to control $\log_2$ ratios.

## Immunohistochemistry and confocal imaging

For quantitative and qualitative analysis of ShakB localization, the brains of 2-day-old male flies were dissected in cold PBS. The brains were fixed for 25 min in 4% PFA/PBS, washed 2 h in PBS and then 4 times for 15 min in 0.3% PBST, blocked in 10% Donkey normal serum in 0.3% PBST (Agrisera, AS10 1564) for 3 h (all at RT), and incubated with primary antibodies diluted in 0.3% PBST containing 5% Donkey normal serum for 48 h at 4 °C. Samples were washed for 5 h in PBS at 4 °C and then 4 times for 15 min in 0.3% PBST at RT, and incubated with secondary antibodies diluted in 0.3% PBST containing 5% Donkey normal serum for 12 h. Samples were washed again 4 times for 15 min in 0.3% PBST and in PBS for 5 min at RT. The samples were mounted on glass microscope glasses with 0.12 mm-deep spacers in VECTASHIELD® mounting medium (Vector laboratories, H-1000).

For the analysis of neuronal morphology, the dissections and immunostainings were done as described above, with the difference that 2–5-day-old flies were used for experiments. The GFP signal was enhanced using anti-GFP primary antibodies in combination with AF594-conjugated secondary antibodies.

For trans-Tango-mediated transsynaptic tracing, the whole CNS of 10–15-day-old flies were dissected in cold PBS, fixed and washed as described above. To detect the postsynaptic partners of HS cells, CF594 anti-RFP antibodies were used. The samples were incubated with antibodies diluted in 0.3% PBST containing 5% Donkey normal serum overnight at 4 °C, washed and mounted as described above.

Antibodies and dilutions used in the IHC experiments: anti-shakB rabbit serum antibody (kind gift of Alexander Borst, Max Planck Institute for Biological Intelligence, Martinsried, Germany; 1:800), goat anti-GFP (Abcam, ab6673; 1:500), goat anti-RFP (Rockland, 200-101-379S; 1:500), rabbit anti-GFP (Thermo Fisher, A11122; 1:500), donkey anti-goat AF488 (Abcam, ab150129; 1:1000), donkey anti-goat AF594 (Thermo Fisher, A32758; 1:1000), donkey anti-rabbit AF594 (Thermo Fisher, A21207; 1:1000), CF594 rabbit anti-RFP (Biotium, 20422; 1:500).

Images of ShakB immunostainings and neurobiotin labelling were acquired on Zeiss LSM800 confocal microscope using 20× air objective (420650-9901-000) or 63× oil immersion objective (420782-9900-799). Images of HS cells for morphological analysis and of transsynaptic labelling were acquired on a Leica SP8 confocal microscope using 25× water-immersion (15506374) or 20× air objective (11506517), respectively. The images were processed using Fiji software.

## Reconstruction of neuronal morphology

To label individual HS cells, we used 2–3 days-old male flies of the following genotype: *+; 20XUAS-SPARC2-I-mCD8::GFP/VT058487-p65.AD; VT000343-GAL4.DBD*. Only flies with single labelled HS cells were used. After IHC and imaging (see above), confocal z-stacks of individual HS cells were used for neuronal reconstruction using Neutube[74]. The generated .swc files were loaded into Imaris software (Imaris9.3.1) as filaments using PylmarisSWC extension, implemented in Python. The diameter of each segment was manually readjusted based on the confocal images. Parameters of Filament dendrite length (sum) and Filament Bounding BoxAA for each neuron were normalised to the size of the optic lobe. For Sholl analysis, the step resolution was adjusted to the total length of the dendrite to obtain an equal amount of Sholl intersections for each cell type. Axes X and Y from Filament Bounding BoxAA were used to compute the dendritic field area.

## Transsynaptic tracing

To identify postsynaptic partners of HS cells, trans-Tango flies carrying a mutant and wild-type shakB alleles

shakB[2],UAS-myrGFP.QUAS-mtdTomato-3xHA/UAS-myrGFP.-QUAS-mtdTomato-3xHA; trans-Tango and

shakB<sup>FlpStopD</sup>,UAS-myrGFP.QUAS-mtdTomato-3xHA/UAS-myrGFP.-QUAS-mtdTomato-3xHA; trans-Tango

were crossed with HS-specific GAL4 driver line R81G07. This way, the genetic diversity and growth conditions between individuals were reduced due to the comparison of wild-type and mutant flies from the same progeny (siblings). The crosses were maintained at 18 °C. Male flies from the progeny were collected after eclosion and kept for another 14–16 days at 18 °C. Fly CNS were dissected and ummar-stained as described above. The variant of *shakB* allele of each dissected fly was identified using primers MiL-F and FRTspacer_3p_for for *Flp-shakB* batch, and primers 5′-CACACCAACGCAACGGTTATATA-3′ and 5′-CGGCCCTGTGAATTGTGAAC-3′ with subsequent sanger sequencing for *shakB[2]* batch.

## Analysis of isoform expression

Quantifying the expression of shakB isoforms was performed with Salmon[75] The transcriptome of *Drosophila melanogaster* was indexed with decoys following the instructions of the Salmon documentation. Briefly, the files dmel-all-chromosome-r6.43.fasta.gz and dmel-all-transcript-r6.43.fasta.gz were downloaded from Flybase on December 16 2021. The chromosome sequences were used as decoy and the index was constructed with default k-mer length 31.

The RNA-seq data from ref. 76 was downloaded from ENA (European Nucleotide Archive) using accession numbers PRJNA658010. Each replicate was quantified independently using Salmon default parameters with options -lA to detect the library type automatically.

## Electrophysiology

1-day-old male flies were briefly anaesthetised on ice and tethered using beeswax to a 3D-printed holder with a hole in the middle fitting the head and the thorax. The head of the tethered fly was passed through the hole and bent down to expose the head's backside. The proboscis was fixed to the thorax with beeswax to avoid head movement. We cut out the cuticle on the backside of the head using a sterile needle (30 G). After the neuropil with LPTCs was exposed, we cut the transverse muscle and removed any fat excess. To digest the neurolemma and gain access to cell bodies, we used a brief 20 s treatment with protease solution (1 mg/ml Protease Type XIV in extracellular solution, Sigma Aldrich, P5147) at RT. The protease was washed off with an extracellular saline solution. The holder with the tethered fly was placed on the setup under 40× water-immersion objective, and cell bodies of LPTCs were additionally cleaned under visual control using a low-resistance patch pipette (tip ~4 μm) filled with extracellular saline solution.

Patch electrodes of 5–7 MΩ resistance (thin wall, filament, 1.5 mm, WPI, Florida, USA) were pulled on DMZ Zeitz-Puller (Zeitz-Instruments Vertriebs GmbH) and filled with intracellular solution. Using a Multiclamp 700B amplifier (Molecular Devices, Sunnyvale, USA), the signals were filtered at 4 kHz, digitised at 10 kHz, and recorded via a digital-to-analog converter (PCI-DAS6025, Measurement Computing, Massachusetts, USA) with Matlab (Vers.9.2.0.556344, MathWorks, Inc., Natick, MA). The recorded membrane potential was corrected for junction potential (12 mV).

The solutions used for in vivo whole-cell patch-clamp recording: extracellular saline solution (in mM): 103 NaCl, 3 KCl, 5 *N*-tris(hydroxymethyl) methyl-2-aminoethane-sulfonic acid, 10 trehalose, 10 glucose, 2 sucrose, 26 NaHCO3, 1 NaH2PO4, 1.5 CaCl2, and 4 MgCl2, adjusted to 275 mOsm, bubbled with 95% O2/5% CO2, and pH equilibrated around 7.3; the intracellular solution (in mM): 140 potassium aspartate, 10 HEPES, 1 KCl, 4 MgATP, 0.5 Na3GTP, and 1 EGTA, pH 7.26,

adjusted to 265 mOsm. In most experiments, 0.5% Neurobiotin was added to the intracellular solution.

### Neurobiotin cell filling and visualisation

After the recordings were accomplished, cells were filled with Neurobiotin using a positive current of 1 nA for 10 min. After filling, the tissue was left in the recording bath for another 10 min for the dye to diffuse.

The heads of flies were fixed in 4% PFA/PBS at RT for 1 h, and washed 3 times for 15 min in PBS. The brains were dissected out of the head capsule in PBS. To visualise neurobiotin we used TSA-mediated streptavidin labelling, as described in ref. 77, with some modifications. For that, fixed brains were incubated in 0.5% PBST; 4% NaCl solution containing 0.5% avidin-biotinylated HRP complex (ABC) solution overnight at 4 °C. After the incubation, samples were washed 3 times for 15 min in PBS and incubated in 0.0001% biotin-tyramide (Sigma Aldrich, SML2135-50MG) and 0.003% $H_2O_2$ in 0.05 M borate buffer, pH 8.5, for 2 h at RT. The samples were washed for 15 min in PBS and 3 times for 15 min in 0.3% PBST. After washing, the samples were incubated Streptavidin–Alexa 546 (Thermo Fisher, S32356) or Streptavidin–Alexa 488 (Thermo Fisher, S32354) diluted 1:500 in 0.5% PBST; 4% NaCl solution at 4 °C overnight. The mounting and imaging of brains were performed as described above.

### Visual stimuli for electrophysiology

Visual stimuli were presented on the screen with the shape of a quarter sphere (diameter 4.6 cm) using an LED projector (Texas Instruments DLP LightCrafter Evaluation Module) at a frame rate of 60 Hz, presented at a 4-bit depth, resulting in a flicker rate of 120 Hz. A reflective filter (ND10A, Thorlabs) was positioned in front of the projector to block and reflect red light, which was then captured by a photodiode and used for synchronising the stimulus and the recorded voltage responses of a cell. The stimulus presented on the screen had both green (peak of LED at ~520 nm) and blue light (peak of LED at ~470 nm). The scripts to present visual stimuli were written in Matlab using Psychtoolbox library[78]. To compensate for the spherical distortions of the screen, we created a custom lookup table that pre-deforms each frame on the GPU. The span of the visual field covered by the stimuli is ca. 140° horizontally and 85° vertically.

### Analysis of patch-clamp recordings

Cell voltage activity was recorded using the custom-designed software in LabView and analysed in Matlab.

**Full-field flashes.** Full-field flashes were interleaved with intervals of dark screen in between. The average response per animal was computed across repetitions. The population average is the average of the responses of all flies with the same genetic background.

**Gratings (contrast, direction and velocity selectivity).** To quantify the tuning of the cells to contrast, direction, and velocity we presented a moving grating stimulus with spatial frequency of 0.04 cycle/°, while changing one of the aforementioned parameters. The moving gratings were interleaved with intervals of the stationary gratings (2 s static before movement onset, 2 s movement, 1 s static after). The baseline computed during the stationary gratings was subtracted from the responses to the moving gratings. Population responses are averages of the responses of individual cells. For the direction tuning analysis, responses were normalised to the maximum response for each cell and then averaged across animals with the same genetic background.

**Power spectrum analysis.** For the power spectrum (PS) analysis, we first extracted the raw responses during the flash OFF periods and subtracted the global mean, such that the contribution of the lowest frequencies does not overshadow the contribution of higher frequencies. We used the Matlab fft function to compute the Fourier transform of the traces. The one-sided spectrum was normalised with respect to the bin size and squared to convert to the power of frequencies. Whenever multiple repetitions of the stimuli were available, we computed the PS for each repetition and averaged them to obtain the PS of the responses of one cell. The PS of each cell was normalised such that the total energy was 1. The PS of the population is the average PS of individual cells with the same genetic background. To quantify the contribution of three different ranges of frequencies, we subdivided the frequency range into two intervals: 0–10 Hz and 11–50 Hz. The sum of the power of the frequencies within these groups is depicted in Fig. 5g as a percentage of the total power over all the frequencies to give the relative contribution of each interval.

**Reconstruction of receptive field using a scanning rectangle stimulus.** A 10° high and 2° wide rectangle scanned the visual field horizontally and vertically. Due to the long duration of the stimulus, a 60 s rolling window was used to subtract the mean voltage from the recording so that only the deviations from the baseline were used for the analysis. The responses were discretized into bins of 16.6 ms (duration of frame update). The position and direction of motion of the rectangle in each frame were weighted by the discretized voltage response to get a response vector at each point in the visual field. We computed the vector sum of the responses to the four cardinal directions and averaged this across multiple repetitions to compute the mean spatial receptive field of a cell which can be represented as a vector field. The location of each arrow in this vector field corresponds to a specific azimuth and elevation in the visual field of the fly; the direction represents the local preferred direction and the length indicates the relative strength of response. After normalising the response vectors of each cell to the maximum, we averaged the responses of multiple cells from animals with the same genetic background to obtain the spatial receptive field of the population. In order to represent the deformations due to the spherical shape of the screen, we deform the receptive field to match the pre-deformed stimulus during the experiment. The horizontal/vertical profiles of the receptive field were computed using the length of the average response vector in the corresponding azimuth/elevation.

### Freely-walking arena

Individual flies walked freely on a 55 mm circular arena made of IR-transparent Perspex acrylic sheet with 3 mm tall walls. The walls were heated with an insulated nichrome wire to prevent the flies from walking on the walls and to encourage them to spend more time close to the centre of the arena. The arena was covered with an IR-transparent acrylic sheet coated with Sigmacote™ to prevent flies from walking on the roof. The entire behavioural setup is placed inside a custom-designed temperature-controlled compartment that maintains the internal temperature at 27 °C. For shi[ts1] experiments, flies were heated to the temperature of 35 °C in a thermal cycler for an hour and then immediately transferred to the arena.

The visual stimulus was projected from the top on the outer face of the roof, which is covered with a projection screen (Gerriets OPERA® Grey Blue Front and Rear Projection Screen). The stimulus was presented by an LED projector (Texas Instruments DLP LightCrafter Evaluation Module) at a frame rate of 60 Hz and pixel size of 12 px/mm. We used only the green LED (peak at ~520 nm) of the projector since the relative sensitivity of the optomotor response of *Drosophila* has been shown to be highest for wavelengths between 350 and 500 nm[79].

The fly was video recorded from the bottom using a monochrome USB3.0 camera (Flea3 FL3-U3-13YM) at a frame rate of 60 Hz and resolution 1024 × 1024 pixels. The arena was illuminated from the top by a custom-made panel of IR LEDs (850 nm). Since the projection screen, the roof and the arena are all made of IR-transparent material; the backlit flies appear as dark silhouettes on a bright background.

## Visual stimuli for behavioural experiments

The stimuli (Fig. 1a) were generated using a custom Python script and the various textures were made using the Psychopy library[80]. The stimulus was updated every frame depending on the position and orientation of the fly. We detected the contour of the fly after performing a pixel intensity thresholding. An ellipse was then fitted to this contour and the position of the centroid and angle of the major axis of the fitted ellipse were used to determine the position and orientation of the fly. The delay between the fly movement and the update of the stimulus was 3 frame updates (<48 ms).

We presented a radial grating pattern with smooth intensity transitions, akin to a sinusoidal intensity transition, that was centred over the fly body. Due to experimental constraints, we could not stabilise the stimuli to the head position. Rotating this pinwheel in either a clockwise or counterclockwise direction evoked a turning response, the optomotor response of the fly. The diameter of the pinwheel was kept 45 mm, since the optomotor response of the fly, which increased with an increase in the size of the pinwheel, saturated at 45 mm and did not change for larger pinwheels. An experiment session consisted of approximately 200 trials of 5 s pinwheel rotation preceded by 5 s without rotation. The direction, contrast and speed of rotation of the sinusoidal pinwheel were changed across trials. Michelson contrast of the grating pattern was computed as:

$$Contrast = \frac{Max\ Intensity - Min\ Intensity}{Max\ Intensity + Min\ Intensity} \quad (1)$$

For experiments with unilateral motion, the pinwheel was divided into 2 segments and the direction of rotation of each segment (counterclockwise, clockwise or no rotation) was changed independently across trials to produce different combinations of rotational optic-flow.

## Pre-processing of behavioural data

The position and orientation of the fly were determined for each frame during the experiment. The contour of the fly was extracted after performing a pixel intensity thresholding and an ellipse was fitted to this contour. The position of the centroid and the angle of the major axis of this fitted ellipse were used to determine the position and orientation of the fly, respectively. While this method is quick (a requirement for performing closed-loop experiments) and works well for determining the orientation of the fly body, it does not provide the heading direction of the fly which could not be extracted at our pixel resolution accurately. The direction was estimated by processing the videos post-experiment. Otsu's binarization was used to obtain two thresholds from the image, one for the body of the fly and one for the translucent wings. The direction of the line joining the centre of mass (COM) of the body and the COM of the wings was used to determine the head direction of the fly, which was then used as the correct orientation for further analysis. All machine vision computations were performed using functions from the OpenCV Python library.

The speed and angular velocity of the fly were calculated from the change in position and orientation, respectively. In order to remove noise, we used a rolling median followed by a rolling average with a window length of 3 frames (~50 ms). Events where the fly jumped were detected using a threshold of 100 mm/s or 1000 degrees/s. Trials in which the fly either jumped, went close to the walls of the arena (within 5 mm of the wall) or was inactive throughout the whole trial were excluded from further analysis.

## Estimation of saccades and path straightness

Fly locomotion is composed of long bouts of relatively straight motion interspersed with sharp turns, called saccades. We detected these saccades using a wavelet transform strategy inspired by Cruz et al. 2021[30]. The stationary wavelet transform (swt) of the angular velocity

was computed using a biorthogonal 2.6 wavelet. The swt signal in the 10–20 Hz band was isolated to reconstruct the angular velocity data with an inverse stationary wavelet transformation. Peaks in this reconstructed angular velocity data that had a maximum angular speed higher than 200 degrees/s and with a width of more than 50 ms and less than 250 ms were deemed to be saccades.

In order to calculate the local curvature of locomotion, we first defined forward walking bouts as periods between two saccades that are longer than 333 ms (20 video frames) where the average speed of the fly is more than 5 mm/s. Then, a window of 333 ms was selected and centred around each point of a walking bout. The length of the straight line connecting the two endpoints of this window (the shortest distance between two points) and the perpendicular distance between this line and the midpoint of the actual trajectory (deviation from the shortest distance) were calculated. A straight walking bout will have a minimal deviation from a straight line trajectory. As such, the straightness of a walking bout was defined as the ratio of the sum of the shortest distances and the sum of deviations from the shortest distance.

## HS-H2 network simulation

The network model used to simulate the response of neurons to different visual motions was adapted from ref. 15 where the authors focused on a disinhibitory circuit involving bIPS and uLPTCrn neurons and adjusted the weights of the connections to match the visual motion responses of the different cell types. We reproduced this model and added a term for the gap junction connection between H2 and HS neurons, the strength of which can be varied by changing the corresponding weight. The weight corresponding to the term in the original model representing contralateral inputs to HS was set to 0. This is consistent with our findings showing that HS neurons receive information about contralateral Back-to-Front motion primarily via the HSE-H2 gap junction. The weight corresponding to the contralateral FrontToBack input to H2 neurons was set to 0.1, as H2 neurons have been shown to receive stronger visual inputs from the frontal visual field[6]. To calculate the effect of gap junctions on the ability of neurons to discriminate between translation and rotation, a measure developed in the original paper called the Discrimination Index (DI) was used, where:

$$DI = \frac{Optic\_Flow\_translation - Optic\_Flow\_rotation}{|Optic\_Flow\_translation| + |Optic\_Flow\_rotation|} \quad (2)$$

$$Optic\_Flow\_rotation = max(clockwise, counterclockwise) \\ - min(clockwise, counterclockwise)$$

$$Optic\_Flow\_translation = max(forward\_translation, backward\_translation) \\ - min(forward\_translation, backward\_translation)$$

We also reproduced our experimental results in silico by setting the weights of either HS or H2 to 0 to compare the effect or discrimination index across the network when modifying different non-overlapping neural substrates (chemical and electrical synapses on the HS-H2 network).

## Statistics

Details about statistical tests are provided in the figure legends and Supplementary Data 1.

## Reporting summary

Further information on research design is available in the Nature Portfolio Reporting Summary linked to this article.

## Data availability

Behavioural data generated in this study have been deposited in the ISTA data repository: https://doi.org/10.15479/AT:ISTA:17488. Source data are provided with this paper.

## Materials availability

Transgenic flies generated in this study are available on request.

## Code availability

Code used for analysis is available at GitHub: https://github.com/joesch-lab/Bilateral-course-control.

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

## Acknowledgements

We thank Georg Ammer and Alexander Borst for sharing anti-ShakB serum antibodies. We thank Nélia Varela and Eugenia Chiappe for the *w1118;+;10XUAS-IVS-eGFPKir2.1/TM6B* fly line, Augustin Hrvoje for the *shakB[2]* line, as well as Jesse Isaacman-Beck and Thomas R Clandinin for the gift of *y¹,w*;20XUAS-IVS-PhiC31;+* fly line. We also thank Armel Nicolas and Tomas Masson for the proteomic analysis, Ece Sönmez for help with fly crosses and dissections for protein analysis, and Lisa Hofer for assistance with the reconstruction experiments. We would also like to thank Laura Burnett for drawing scientific illustrations used in the

figures. We are particularly grateful to members of the Siekhaus, the Kondrashov, and the Chiappe group for providing material support and technical advice. We are grateful to Daria Siekhaus, Eugenia Chiappe, Alexander Borst, Ben deBivort, and all the members of the Joesch laboratory for valuable discussions and comments on the manuscript. Stocks from the Bloomington *Drosophila* Stock Center (NIH P4OOD018537) and the Vienna *Drosophila* Resource Center were used in this study. The Scientific Service Units of ISTA supported the project through resources provided by the Imaging and Optics Facility, MIBA Machine Shop, and the Lab Support Facility, as well as Vienna *Drosophila* Research Centre. This work was funded by the Deutsche Forschungsgemeinschaft (DFG, German Research Foundation) as part of the SPP 2205 – 429960716 (M.J.).

## Author contributions

V.O.P., R.S., and M.J. designed the study. V.O.P. performed the electrophysiological experiments, intracellular dye fillings, neuronal tracing and reconstructions, protein analysis, immunochemistry, acquisition and analysis of microscopy images. R.S. built and designed the behavioural setup, and performed all behavioural experiments and analyses. O.S., R.S., and V.O.P. performed the analysis of electrophysiological data. O.S. and M.J. built the electrophysiology setup. V.O.P., R.S., and M.J. wrote the manuscript.

## Competing interests

The authors declare no competing interests.
