## [Peer Review File · Nature Communications]

Bilateral interactions of optic-flow sensitive neurons coordinate course control in fliesREVIEWER COMMENTS

Reviewer #1 (Remarks to the Author):

Summary

This manuscript dives into a technically vexing problem of how gap junctions operate in visual processing. The challenge has been in experimentally manipulating the right gap junction genes in a cell-specific manner. The authors develop a new robust genetic strategy to use an inducible flp construct to disrupt the transcription of a key gap junction gene under the control of a Gal4 driver, allowing for a cell-type-specific perturbation. They then characterize the impact of gap junction perturbation within specific visual integration neurons on the visual receptive field properties of the neurons and on the visual control of walking behavior. They develop a clever walking paradigm in which a fly freely walking within a small arena is tracked in real time and receives continually updated visual stimulation - a sort of “visual clamp” of optic flow on the eye of the freely walking fly. Using this two part approach, they find that gap junctions do not establish the directional selectivity of the neurons, but rather sculpt the extent of their spatial receptive fields. Additionally, connectivity among visual projection neurons is disrupted, along with visual control behaviors that they serve. Notably, disruption of gap junctions in the HS and VS class of visual projection neurons largely eliminates nondirectional small-field avoidance saccades that impose strong nonlinearities in the optomotor stabilization of gaze; without the gap junctions, the avoidance saccades are largely absent, and the operation of the smooth directional optomotor system is highlighted. I find the work novel and compelling. In particular, the genetic reagents and electrophysiology are strong contributions to the literature. I do take issue with the current interpretation of the behavioral findings, which I detail below.

Major comments

Conceptual framework: the authors’ claim that the pinwheel stimulus that they project onto one visual hemisphere is “monocular”, but claim this is incorrect. In *Drosophila melanogaster*, the region of binocular overlap spans roughly 40-degrees across the visual midline (see Buchner 1976, Heisenberg/Wolf, (Aptekar, Shoemaker, and Frye 2012; Kemppainen et al. 2022). Thus, the stimulus presented here to one half of the visual field, bisected by the longitudinal body axis, is not monocular; rather it is unilateral and activates 50% of the binocular visual field. You are not the first to err in this way - the Duistermars et al. paper cited (Duistermars, Care, and Frye 2012) also conflates monocular and unilateral stimuli. The text shall have to be heavily revised to address this issue because the implications are important (although Figure 7j gets it right!). For example, consider that object tracking saccades in flight are most frequently evoked by stimuli appearing outside of the region of binocular overlap (Aptekar, Shoemaker, and Frye 2012; Mongeau and Frye 2017). Notwithstanding this conceptual criticism, there is a very interesting story here! This

manuscript merely examines unilateral-bilateral interactions, rather than monocular-binocular ones.

Interpretation: some of the visual behavior is misinterpreted. Considering the stimulus geometry, the bilateral pinwheel stimulus projected onto the ceiling of the walking arena subtends less than $\frac{1}{2}$ of the total visual field (the exact subtense should be calculated, see critique item below). In response to the full bilateral pinwheel stimulus, flies evoke mainly smooth optomotor turns in the direction of image rotation in an attempt to minimize slip and stabilize gaze, as the authors correctly interpret. However, in response to unilateral stimulation, impinging upon roughly $\frac{1}{4}$ of the visual field, the smooth optomotor response largely disappears and a new behavior emerges: saccadic turns contralateral to the projected stimulus, regardless of its direction of rotation. The authors frame their analysis (and their conclusions) relative to the direction of image rotation (syn-directional turns for back-to-front stimuli, and anti-directional for front-to-back stimuli). They therefore surmise that optomotor responses to movement subtending one visual hemisphere are reversed for one front-to-back motion, but not back-to-front. They should instead frame their analysis relative to the position of the stimulus, which seems to me to evoke saccades away from the unilateral stimulus regardless of its direction of motion. This is a more parsimonious interpretation of the data, is clear from the movies and raw trajectories that show saccades (bilateral responses appear smooth, unilateral responses are saccadic), and is consistent with published work showing that flying flies evoke aversive saccadic turns away from small objects regardless of their direction of motion e.g. (Mongeau et al. 2019). What you have discovered here is that gap junctions assembled by LPTCs arbitrate the balance between smooth optomotor gaze stabilization of large-field cues and saccadic avoidance of small-field cues.

How might LPTCs arbitrate such a switch? I note that a recent study demonstrated that unilateral optogenetic activation of HS neurons evokes counter-directional steering in flight (Kim et al. 2023). Also, the ratio of HSN/HSE activity in your behavioral experiments (compare dorsal RFs Figure 5a/b) becomes considerably smaller with Flp-shakB. I.e. the transition from smooth slip-stabilization to saccadic avoidance is coincident with increased HSE activity in the dorsal RF, as well as all of the other connectivity changes indicated in Figure 6. Another consideration is that gap junctions operate immediately, whereas chemical transmission requires calcium accumulation in the presynaptic terminal. Perhaps saccades are organized by rapid communication across LPTCs, whereas smooth optomotor responses are coordinated by integrated electrotonic signals. I look forward to your interpretations!

Figure 1: Using the same color scheme for saccades (d) and smooth pursuit (e) is confusing. Perhaps show saccades with narrow vertical black raster marks? Or show a saccade raster? The graphs in panel i are really nice - separating the component of smooth pursuit from saccades. This confusion occurs again in Figure 7 in which turning direction is indicated in green/pink, and then below the smooth response is in pink (or similar hue).

Figure 5. I think it is worth explicitly stating that your stimuli only activate only the uppermost region of the receptive fields. It would also be worth the effort to calculate the angular size of the projection of the behavioral stimuli onto the retina and receptive fields of the relevant LPTCs.

L441 and elsewhere: the terminology is confusing - “turning toward” is generally relative to the direction of motion not the unilateral position of the stimulus. Rectifying the smooth directional responses to full-field and saccadic avoidance responses to hemi-field stimuli will mitigate this issue.

Minor comments

Title: remove ‘binocular’ and perhaps retitle: Gap junctions coordinate transition from course stabilization to avoidance

I do not believe that the figures and panels are called out in order. Is this a requirement for Nature Comms?

L79 cite (Strauss, Schuster, and Götz 1997) who implemented a very similar virtual reality system in freely walking flies, and the biased open loop during free turns approach is also similar to (Rimniceanu, Currea, and Frye 2023) in magnetically tethered flight

L146: “all and only” I think needs to be reworded

L272: “tethered” in this situation is misleading, implying behaving flies. Say “restrained” or “immobilized head-fixed”

L417 change to “Crucially, the reduction of contralateral electrical connectivity enables us to test the involvement of LPTC-mediated behavioral role of gap junctions.”

L440-445 revise with framework of smooth optomotor stabilization of bilateral motion direction vs saccadic orientation away from unilateral stimulus position

L36-41: These statements, thus the premise, about equivocal motion cues are vague and incomplete. Only locally are motion cues equivocal. Revise for specificity and clarity.

L88-89: In walking flies. Be specific. Recentering saccades have been described only in flying flies as resembling the fast-phase of an optokinetic nystagmus (see Mongeau & Frye, 2017)

L92: be more specific adding “major” fly’s body axis

L75: stimuli are not monocular

L110-111: Wouldn’t the number of saccades be a more straightforward measurement of the saccadic behavior? E.g. rasters mentioned above

L116: Extended Data Fig.2b?

L118: Extended Data Fig.2d,e?

L146-147: “all and only”, VT058487-Gal4 targets other neurons in the brain such as FB, AVLP and PLP (to say the least). Remove the word “only”

Fig.2A: Are you sure it is a MIP image of the brain? Looks more like a single plane

L159: There is no Fig.2l

L207-208: for clarity, please use the nomenclature *shakB*[FlpStop-ND] and *shakB*[FlpStop-D] and avoid the abbreviation Flp-control and Flp-*shakB*. Both in the manuscript and Fig.3-4

L211: next time use *shakB*[2] to define the mutation and not the superscript 2 because it is difficult to distinguish it from a reference. Alternatively, write “*shakB*2-mutant”

L414-417: Since there is no effect in the connections with downstream neurons, then why is this a good model for studying gap junctions? This section is confusing. Maybe remove “making Flp-shakB a favorable model for studying the behavioral role of gap junctions”

L417-419: I would be more specific adding “reduction of contralateral electrical connectivity with H2 cells”

L448-449: the conclusion that the binocular interactions mediated by gap junctions are responsible for the behavior is a bit hasty. The flies used for this experiment are not expressing shakB (A and E isoforms are probably not even expressed in the adult brain) not only in the HS neurons (for those we have hints of interhemispheric connections through H2) but also in many other visual neurons that could play a role in this FtB counter-directional response. If it is true that the response in shakB[FlpStop-D] flies looks similar to HS/VS>Kir2.1 flies, that doesn't mean that shakB in those neurons is directly responsible unless the shakB is selectively disrupted in those neurons. In other words, that “same” effect might be the result of two independent pathways/mechanisms

It is not readily apparent how the authors can discriminate between an effect due to the loss of electrical coupling among HS cells and the effect due to the loss of interhemispheric connectivity. In support of the conclusion that H2 cells are the connector, it would be necessary to silence them and recapitulate the lack of FtB counter-directional response

L501-503: This seems to be an overstatement

L506: state-of-the-art

L1109-1110: “The entire behavioral setup is placed inside a custom designed temperature-controlled compartment that maintains the internal temperature at 27°C”. The same exact sentence is repeated at L1129-1130, which is a bit odd. Also, does this mean that the experiments were run at 27°C?

The use of only male flies represents a marked divergence from the vast majority of the literature on fly visual behavior (females are larger, more easily manipulated for physiology and behavior). Did the authors test female flies as well in response to half optic flow patterns? If so, how was the response?

Regarding the flies rearing conditions: Mano et al., 2023 showed that the temperature affects the anti-directional turning behavior. At 25°C flies show less anti-directional behavior than flies reared at 20°C. Since the authors reared their flies at 18°C, did they also try flies reared at different temperatures? If so, how did temperature affect the main findings of the paper?

Change saccade strength in saccade amplitude please (e.g., Extended Data Fig.2d)

Supplementary table 1: please separate the genotype name from the number of flies used:

Figures:

Fig.1: please report that everything related to angular speed and cumulative angular distance is not absolute but relative to the stimulus. In other words, positive and negative values are corrected based on the direction of the rotating pinwheel (i.e., syn- vs counter-directional).

Also, what is the sample size? Please report N of flies recorded (at least) and how many trials per fly in the legend not only in the supplementary table 1

Fig.1c: report the direction of the rotating stimulus please

Fig1d: what does the dashed line mean? I suppose the starting of pinwheel rotation but you need to report it. Also, saccades may be better defined as spontaneous saccades

Fig.1m: the average of trials per fly is removing a lot of useful information. Could the authors just plot all the data points collected?

Fig.1i: I don't understand this plot. How was the plot generated? What is delta angular speed? Do the contours represent the mean angular speed considering exclusively smooth optomotor (red) or saccadic period (blue)? Explain it better in the legend please

Fig.1i: two (i) with two different descriptions, very confusing. Correct as (j)

Fig.1k: summary plot: scatter plot? (same for Fig.1l). Also, add mean \pm SEM of the mean per fly

Fig.1l: What does “turns made” mean? the response amplitude? Please explain it better

Fig.2: using UAS-Kir2.1 flies as control is not the best of the choices for the genetic background. Especially if you use as experimental flies not only bearing a Gal4 but a tsh-Gal80 as well. It would have been better to use flies at least with tsh-Gal80/+;UAS-Kir2.1/+ and ideally with tsh-Gal80/+; UAS-Kir2.1/empty-Gal4 (BDSC # 68384). The early angular velocity of the UAS-Kir2.1 in response to full-field rotation (Fig.2e) is on average twice the angular velocity of Canton-S (Fig.1h)

Please plot the raster plot of the angular velocity as done in Fig.1 and Fig.7.

Extended Data Fig.2c: angular speed (top) and turns (bottom), please correct.

Fig.4i: the moving edges cartoons are confusing, why is there depicted two arrows and a bar in the middle? That’s not an edge, that’s a bright bar moving over a uniform gray background with ON and OFF transitions. Please fix it. Same for Fig.4j

References

Aptekar, Jacob W., Patrick A. Shoemaker, and Mark A. Frye. 2012. “Figure Tracking by Flies Is Supported by Parallel Visual Streams.” *Current Biology: CB* 22 (6): 482–87.

Duistermars, Brian J., Rachel A. Care, and Mark A. Frye. 2012. “Binocular Interactions Underlying the Classic Optomotor Responses of Flying Flies.” *Frontiers in Behavioral Neuroscience* 6 (February): 6.

Kemppainen, Joni, Ben Scales, Keivan Razban Haghghi, Jouni Takalo, Neveen Mansour, James McManus, Gabor Leko, et al. 2022. “Binocular Mirror-Symmetric Microsaccadic Sampling Enables *Drosophila* Hyperacute 3D Vision.” *Proceedings of the National Academy of Sciences of the United States of America* 119 (12): e2109717119.

Kim, Hyosun, Hayun Park, Joowon Lee, and Anmo J. Kim. 2023. “A Visuomotor Circuit for Evasive Flight Turns in *Drosophila*.” *Current Biology: CB* 33 (2): 321–35.e6.

Mongeau, Jean-Michel, Karen Y. Cheng, Jacob Aptekar, and Mark A. Frye. 2019. “Visuomotor Strategies for Object Approach and Aversion in *Drosophila Melanogaster*.” *The Journal of Experimental Biology* 222 (Pt 3). <https://doi.org/10.1242/jeb.193730>.

Mongeau, Jean-Michel, and Mark A. Frye. 2017. "Drosophila Spatiotemporally Integrates Visual Signals to Control Saccades." *Current Biology: CB* 27 (19): 2901–14.e2.

Rimniceanu, Martha, John P. Currea, and Mark A. Frye. 2023. "Proprioception Gates Visual Object Fixation in Flying Flies." *Current Biology: CB* 33 (8): 1459–71.e3.

Strauss, R., S. Schuster, and K. G. Götz. 1997. "Processing of Artificial Visual Feedback in the Walking Fruit Fly *Drosophila Melanogaster*." *The Journal of Experimental Biology* 200 (Pt 9): 1281–96.

Reviewer #2 (Remarks to the Author):

The authors report a study on the heterolateral network connectivity of lobula plate tangential cells (LPTCs) enabling the integration of binocular visual motion information in *Drosophila*. They combined (i) behavioural studies and (ii) electrophysiology with (iii) genetic manipulations of certain pathway elements involved, with a focus on electrical connections (gap junctions) between directional-selective LPTCs, specifically horizontal cells HSN/HSE and the heterolateral H2-cell. They found that disabling gap junction coupling using *Flp-ShakB* mutants has a qualitative impact on the way in which visual motion information from both eyes is combined.

Generally, the topic is certainly of general interest. It was approached using a combination of demanding methodologies and the data are appropriately presented in well-designed figures. There is, however, an imbalance due to a substantial weight on the genetical methods used (including their validation) while a differentiated framework for addressing the scientific question is missing. All the authors provide along the latter is a qualitative account on a generic stabilization reflex (optomotor response), without defining the functional significance for specific behavioural tasks. In my view it is next to impossible to interpret experimental results without knowing what the functional context is (see further comments below).

In the following I list several major issues the authors should address before I would be able to provide a recommendation on whether or not their work may be suitable for publication in *Nat Communications*. In addition to the comments below, I will provide an annotated version of the submitted manuscript including further comments I would like the authors to consider when working on the revision.

My recommendation would be, in any case, to engage on a major revision.

- 1) Clearly define what is currently known about optic flow-based self-motion estimation and control and how the observed behaviour supports specific aspect of “visuo-motor control”
- 2) Focus on the gaps in understanding the functional role of gap junctions in support of specific visuo-motor control tasks. This is where the merits of your work are.
- 3) Motivate the choice of the methods you apply and discuss the potential limitations.
- 4) Revise your terminology keeping a general readership in mind.
- 5) Include a functional interpretation of your experimental data in the context of visuo-motor control. If possible, and in case you decided to talk about computational aspects, include at least a qualitative model of what functional impact the presence/absence of gap junctions has at the network level and how it is related to the observed behaviour.
- 6) Revise the abstract and the discussion section to avoid overstating the conclusions based on the experimental results you obtained.

For further guidance I have added a sentence-by-sentence review of your abstract at the bottom of this report.

Major comments

Editorial:

- The manuscript contains area-specific terminology that is not introduced/explained, which makes it difficult for non-specialist readers to follow the logic of the approach (e.g. innexin; intronic, etc.).
- Also related to terminology: expressions such as “computation” and “algorithm” only make sense when used in connections with operations that are required to control a specific task. Normally, a model helps to formally describe the required operation and to constrain the possible solutions which transform a given set of inputs into the observable systems output. My recommendation for the authors would be to either remove those expressions or to introduce more formal description that would justify the used of such terms.

- Similarly, the term “decision strategy” is hardly justified given (a) the methodology used, (b) the level of the visuo-motor pathway studied, and (c) the lack of an interpretation in the context of a specific optomotor control task (see further comments in amended pdf-file).

Previous work:

How the visual system obtains unambiguous information on self-motion based on the processing of optic flow has been studied extensively in the past (theory/modelling: e.g. Koenderink and van Doorn 1987; Dahmen 2001). In this context, the role of heterolateral LPTCs had been studied in blowflies (e.g. Hausen et al. and Krapp et al.). Substantial contributions to the network of LPTC connectivity in blowflies and *Drosophila* comes from Borst et al., as well as other groups working on fly vision. Even the electrical connectivity between HSN/E and the H2-cell was included in a paper by Borst and Weber (2011).

The introduction includes some relevant citations. But my impression was that it did not bring across the level of understanding that has been established already. Neither the general question Pokusaeva et al. are addressing is novel, nor the finding that the H2-cell contributes to the binocular integration of visual motion information in the context of self-motion estimation. The remit of their manuscript lies in the finding that the failure of gap junctions can alter the properties of neuronal circuits (and thus the behavioural output) in an unexpected, non-linear way. It also shows (ones more) that the phenotype of genetic manipulations depends on genetic background and on specific mutants used (e.g. *ShakB* vs *ShakB2*). This, in turn, makes it particularly difficult to explain the results of the manipulation and come up with a functional interpretation (see further comments in annotated pdf-file).

Here are some questions related to this point:

- *Flp*-control produces stronger behavioural responses than CantonS wild type flies. How do you explain this finding?
- The lack of contralateral input (*Flp-shakB*) introduces smooth optomotor responses upon FtB motion – while *Flp*-control mutants generate anti-saccades. How do you explain this finding?
- Silenced HS and VS-cells: flies still show optomotor responses. How would you interpret these observations? Did you consider:

(i) the possible impact of efference copies?

(ii) the possible impact of other, e.g. position-dependent visual pathways?

Other comments:

- Figure 3c: only HS-cells? Fig3c shows VS-cells.
- 60Hz stimulus presentation during e-phys. What is the flicker fusion frequency in *Drosophila*? It may have an accumulative effect on signal integration - even at slow response dynamics.
- Did you consider head movements? How would they affect your results?
- A clear-cut interpretation of the smooth and saccadic syn- and anti-rotations should be provided. Specifically, what is the functional interpretation of the saccadic anti-rotations in non-genetically modified animals?
- Related to L 460 and the previous comment: I am missing a functional interpretation of your findings. From what I have learned so far, it looks as if the non-modified connectivity enables a higher degree of behavioural variability (smooth syn-turns + saccadic anti-turns). So, I am not sure I understand why you are talking about "constraining the behaviour". To turn these properties into something behaviourally relevant, would the fly have to be able to control the conductivity of the gap junctions?
- Related to L. 515: this sounds all fine. But besides mentioning optomotor responses, you have not provide a functional interpretation of the various behavioural observations (see earlier comment). How can you hope to understand the "underlying neuronal mechanisms" if you don't know what behavioural function they are supposed to control?
- Discussion: The candid discussion of the potential problems using genetic tools is much appreciated. However, for all those issues you should also discuss in which way they may affect the functional interpretation of your results and the conclusions you draw.

Comments on the Abstract:

“Animals rely on visual motion cues to maintain stability and navigate accurately, requiring a range of compensatory actions and sophisticated neuronal computations¹.”

Comment: “ ... sophisticated neuronal computation.” Should be something along the lines “ ... task-specific neuronal processing.”

“Classically, this phenomenon ...”

Comment: “this phenomenon”. Reference not clear. You have included several attributes in your previous sentence.

“... has been studied through the optomotor response², a reflexive visual stabilization behavior.”

Comment: “optomotor response” is a term that includes more behaviours than just the “optomotor course control” originally studied by Hassenstein and Reichardt in the 1950s. It would be useful, in general, to think about the function of the reflex you are talking about. In case of the behaviour observed by Hassenstein and Reichardt, the purpose is to stabilize the entire visual image on the eyes in response to an externally induced perturbation of a given trajectory. This is indeed a stabilization reflex that is control in a negative feedback loop (as far as vision is concerned). However, it has been demonstrated by work on LPTCs in *Drosophila* that efference copies (e.g. Kim et al. 2015) can modify the reflex whenever the animal engages on other vision-driven behaviours, e.g. responses to looming stimuli (Fenk et al. 2021) relevant for escape and collision avoidance.

“However, the simplicity of this behavior is insufficient to explain the complexity of natural motion cues experienced during navigation, ... “

Comment: I don't think that the optomotor course control was ever meant to “ ... explain the complexity of natural motion cues ...” altogether.

“ ... and the control mechanisms involved remain poorly understood.”

Comment: for this statement to be correct, you should say exactly (i) which specific vision-driven behaviour you are talking about and (ii) what its function is. Your wording comes very close to being offensive for many excellent scientists who, quite successfully, devoted their entire career to study specific visual behaviours quantitatively.

“Here, by identifying a rich repertoire of course control behaviors in *Drosophila*, ...”

Comment: I didn't see “a rich repertoire of course control” in your study. You didn't apply any stimuli that would allow you to “identify” a rich repertoire. You used an abstraction of a wide-field stimulus in your closed-loop behavioural studies to observe the impact of monocular and binocular motion on the turning behaviour of the fly. The responses were limited to smooth and saccadic behavioural turns syn- and anti-directional with respect to the induced image motion. But you did not mention/discuss the behavioural relevance of the different response components.

“ ... we establish a direct link between opposing corrective actions and the underlying circuit motifs.”

Comment: along the lines of my previous comment – although you could show behavioural changes upon modifications of the neural circuit, i.e. suppressing binocular signal integration, I would not refer to a “rich repertoire”.

“We determine the algorithmic substrate ... “

Comment: “algorithmic substrate” is very unhelpful wording. How can you determine an algorithmic substrate when you don't provide a clear-cut functional hypothesis?

“ ... in a specific set of motion sensitive interneurons in the fly brain and demonstrate that visual motion information from both eyes plays a crucial role in movement control through bilateral interactions ... “

Comment: up to this part of the sentence there is nothing new. There are several accounts on binocular integration of visual motion, both in blowflies and in fruitflies – which are not appropriately cited.

“... of electrical signals facilitated by gap junctions.”

Comment: the merits of the current manuscript lie in the studies of the electrical connection between HSN/E and a heterolateral LPTC, the H2-cell. Although the connection itself was already published by Borst and Weber (2011), disrupting it using generic methods (e.g. Flp-ShakB), is a novel approach.

“These electrical interactions augment the classic stabilization behavior by inverting response direction and behavioral strategy.”

Comment: “behavioural strategy” is a perfect example of an overstatement. It implies purposeful (cognitive) action, where the behavioural alteration was caused by disabling gap junctions required for binocular signal integration (see also commented manuscript).

“Our findings provide insights into how animals integrate monocular motion cues to generate a diverse range of behaviors, ...”

Comments: “diverse range of behaviour” is overstating the behaviours reported here.

“ ... elucidate the functional significance of the circuit components, and establish a non-canonical role of gap junctions - non-linear operations.”

Comments: yes – gap junctions appear to be important. And yes – the experiments suggest gap junctions to enable the non-linear interaction of binocular motion cues. BTW: based on behavioural work on the landing responses (Borst and Bade 1988) and recordings from the cervical connective (Borst 1991), the non-linear integration of binocular motion signals had been suggested in blowflies before.

“These circuit computations ... “

Comment: I did not find any reference to “computations” in the manuscript, in a sense that the input x requires a “computation” (or formal description) to result in the output y . What the manuscript includes is evidence for a nonlinear transformation based on the difference between the linear sum of the responses to monocular and binocular visual motion and the response to binocular input. Identifying the underlying “computation” would require a more formal description/model that constraints the possible solutions.

“ ... expose neuronal coordination that plays a decisive role in animal behavior beyond the optomotor response, ... “

Comment: this requires further explanation.

“ ... resembling computer vision strategies employed in cutting-edge autonomous artificial navigation^{3,4}.”

Comment: This claim is not related to any aspect of the work reported here. If there was a link – which is difficult to see even for a reader that may be familiar with “cutting-edge autonomous artificial navigation”, it needs further explanation. Otherwise, it would be interpreted as a statement that aims to make the work more relevant for technical applications than it actually is.

Reviewer #3 (Remarks to the Author):

In “Gap junctions coordinate...” Pokusaeva et al investigate the influence of gap junctions on optomotor responses. They use a nice combination of a novel walking closed loop arena, genetics, and well designed visual stimuli to show the importance of bilateral interactions on the HS cells, which are involved in yaw optomotor responses. In particular, they show that gap junctions play a key role in this. It's a nice paper and I mainly have editorial comments.

In the introduction where you talk about the non-uniformity of naturalistic optic flow, you should mention the work of Egelhaaf, especially his papers with Lindemann, as they have thought about this a lot at the theoretical and experimental level.

Line 45, 45-60 LPTCs are often mentioned. The exact number might be ambiguous, but may say at least 30?

Line 47: Kit Longden has shown translation sensitive LPTCs.

Line 55: Several new exciting papers coming out of e.g. the connectome efforts should be mentioned, e.g. <https://www.science.org/doi/10.1126/sciadv.abi7112>

<https://www.biorxiv.org/content/10.1101/2023.08.06.552150v3>

<https://www.biorxiv.org/content/10.1101/2022.12.14.520178v1>

Line 78: Please mention the 50 ms delay for closing the loop here

Line 83 – I was wondering what the benefit of your closed loop system is compared to e.g. the one developed by Andrew Straw?

Line 88: haven't the Chiappe lab or Maimon lab done work on anti-saccades that should be mentioned here?

Line 101, mention the work by Tammero here, as it is highly relevant, e.g.:

<https://journals.biologists.com/jeb/article/207/1/113/14753/Spatial-organization-of-visuomotor-reflexes-in>

That paper also does a good job of citing the classic literature, and as there is a lot of published work that is relevant to this paper, this should be cited.

Line 134, panel is j not i.

The fonts in all the figures are really small (or I am just too old).

The ringing that you show in response to PD motion in figure 4i and j seems to be happening at 60Hz. If so, it could be amplified by the refresh rate of the screen? What did you show in between the stimuli?

When you map the receptive fields, could you use the terms LMS and LPD as they are standard in the field? Could you also cite the original papers for the method, which are cited in the Schnell paper you cite.

Line 338: use the word more dorsal, rather than higher

Line 349: what is total sensitivity?

Figure 5: Your receptive fields are more ventral than published receptive fields. Why is this?

I am not convinced about your comparison in figure 5e and f. If I've understood it correctly, you have normalized the LMS to the max in each neuron. You are then subtracting control from Flp-ShakB. This means, that weaker overall LMS, i.e. weaker directionality, will appear as a receptive field difference? What is the scale in panels e and f?

Line 383, cite relevant papers.

I really like your figure 7ciii. Such a cool finding.

The start of the finding makes it sound like you have completely reinvented motion vision research, but as mentioned above, a lot of good work could be cited here. Many papers have 'expanded the classical optomotor paradigm', it is well known that motion vision is non-linear, you do not need really investigate natural motion statistics, and the sentence on line 500 ('Yet these studies..') seem to suggest that all previous work was subpar. Your work is excellent in itself, and you can simply summarize your findings here, without making it sound like you invented everything.

Line 512, mention all the excellent work done by e.g. Jurgen Haag.

Line 1060-1070: We need information about stimulus time, and interstimulus intervals.

Line 1085, cite relevant literature. What neural delay was assumed when calculating the receptive fields, the LMS and LPD. This is obviously important. Use the terms LMS and LPD.

Lines 108-110 and 128-130 repeat each other.

Line 1140, Mention that this is the Michelson contrast definition.

Red data in Extended data Figure 6j – how can this be such an outlier?

Figure legend to Extended data Figure 7 has not been justified. Line 76 mention oscillations that are not seen in the associated panel. The ringing appears to be at 1 Hz?

I did not understand how you got the receptive field maps in Extended data Figure 8d and e.

What is the saccade strength in Extended data Figure 10d?

Reviewer #4 (Remarks to the Author):

* The paper by Pokusaeva et al. studied the role of gap junctions and well-known visual neurons in one of the most fundamental visual course control behaviors: the optomotor response. This paper is significant for several reasons: (i) the role of gap junctions in this critical model organism is significantly understudied compared to chemical synapses, despite its prevalence. Especially now, as the wiring diagram of the entire brain, based on EM connectomes, is nearly complete, the study of electrical connections becomes timely (refer to a review by Bargmann & Marder 2013 for the importance of gap junction in the post-connectome era). (ii) the binocular integration of optic flow is crucial for mapping vision to action. This study offers an example of dissecting this function from the behavioral to the cellular levels. (iii) The development of a new behavioral assay appears useful for identifying nonlinear binocular integration. The text is written clearly. The experiments employ a range of cutting-edge techniques, including whole-cell patch clamping, new behavioral assays, and the generation of gene knock-down fly lines. The analysis was conducted with an appropriate level of sophistication. While the initial discovery of the nonlinearity in binocular integration and the development of the knock-down agent are praiseworthy, the subsequent exploration and interpretation did not seem entirely coherent. We thus recommend a major revision, which should include additional experiments and analyses.

* Major comments

- In my opinion, the nonlinearity of the optic flow response across both eyes requires additional validation due to the potential drawbacks of the new behavioral assay. First, since the closed-loop control is based on the body axis, the effect of head movement on the visual input is not considered. Second, the closed loop implementation would have significant amount of delay, and it may have effects on the behavior of the fly. Third, the behavioral assay developed and utilized in the paper operates not only in a closed loop but also in an open loop. It functions in a closed-loop mode because the position and orientation of the pinwheel stimulus are controlled by the fly. Concurrently, it's open-loop as the rotation of the pinwheel isn't controlled by the fly's action. Fourth, as noted in the Discussion section, the visual patterns on the roof might have influenced the main findings of the paper. Hence, we request the authors to conduct similar experiments on a tethered walking assay with an air-cushioned ball. We believe that the visual control is more precise in this setup compared to the one used in the paper. If the results from the ball setup align with the existing findings, it would enhance the generality of the main finding of the paper. Any observed differences would likely be attributable to the unique nature of the behavioral assays.

- The only pattern used in the paper is pinwheel. It would be helpful if one can observe the same nonlinear integration in response to different optic flow patterns. One good example is the “starfield” pattern used in Weir, Schnell, Dickinson (2014).

- Two critical behavioral experiments in the paper, one with HS>Kir and the other with Flp-ShakB, resulted in the same phenotype: the abolition of nonlinear interactions between the left and right optic flow inputs. However, I believe it's highly improbable that these effects stemmed from modifications in the same neural substrates, let alone the possibility of one being a subset of the other. That is, the HS>Kir manipulation would have rendered HS cells largely incapable of transmitting neural signals, leaving the majority of other neurons unaffected. In contrast, the Flp-ShakB flies would have influenced a significant portion of innexin8 throughout the brain. Since bilateral interactions via gap junctions might take place at various stages of the visuomotor pathways, including the levels of early visual neurons, VPNs, DNs, and even further down near the motor neurons, the abolition of nonlinear interaction is unlikely to be fully attributed to the gap junctions at the level of VPNs.

- We would like to suggest a simulation of a model in which HS>Kir led to the abolition of nonlinearity. There are many modeling studies on this neuron (e.g., Bahl et al., 2013 Nat Neurosci). Expansion of those models would likely to recapitulate the HS>Kir result.

- While it seems less likely, there still exists a possibility that H2-HS electrical connections are important for this behavioral change. Therefore, we request additional experiments with H2>Kir and H2-selective ShakB knockdown.

* Minor comments

- Figure 1 legend, line 128, consider replacing ‘rotation’ with ‘stimulus’ or ‘rotation stimulus’, to better describe that it is rotation of the pattern, not of the fly.

- Line#172, consider giving explanation of what 'HSE' stands for as this is the first time it is mentioned (also for HSN and HSE on line 333).
- Line#181, consider using a less strong word than "eliminate". GAL80 does not always fully suppress the GAL4 activity.
- Figure 4 legend, perhaps more clarification about features of the figures. For example, in 4c the arrows can be more clearly described to better indicate which is FtB and BtF motion. 4h-j consider clearer titles for each plot as some of the symbols used lack some visual clarity.
- In Figure 5d, authors suggested that the contralateral visual response of HSE cells is likely due to visual inputs from H2 cells from the other side (Line#349-360). We could not find any direct evidence for this statement (especially Line#358-360). I think that the direct evidence can be shown with H2-selective silencing or gap junction knockdown (as suggested in major comment#2).
- In Figure 5, one of the most pronounced changes is observed in the ipsilateral dorsal visual field of HSE cells (red circles in Figure 5f). I think that a description of this phenomenon and the candidate mechanisms would be helpful.
- Line#355, the abolition of the contralateral responses in HSE cells were not "complete". Significant portion of contralateral responses between 0 and -15 degrees in azimuth remained in Flp-ShakB
- Figure 6a, consider some labelling of these micrographs to better aid in explanation of connectivity. Highlighting ipsilateral HS connections, or downstream connections may aid in understanding.
- Line#490: "the statistics of the visual stimulus" the BtF vs. FtB difference in the contra-lateral visual field does not seem to require probabilistic (or statistical) definition. There may exist some statistical match between the optic flow and visual behaviors, but I do not think that any of the experiments performed in this paper explored this question. Please consider to replace "statistics" with a word that better reflect the contents of the paper.
- Line#1128 of methods, temperature of the arena is mentioned twice. Perhaps this is unnecessary.
- Line#529: close —> closed
- Line#589: Shakb —> ShakB

Point by point response to the reviewers' comments

We thank the four reviewers for their insightful comments and suggestions. In our revised version of the manuscript, we have added substantial new data and analyses. First, we have added an additional main figure showing the involvement of the HS-H2 network in the anti-saccadic behavioural response (new Figure 3) via an interplay of electrical and chemical synaptic interactions. Second, we have substantially expanded our analysis of the behaviour by adding four new supplementary figures that further define the properties of the behavioural responses and expanding many others. Thirdly, we have reformulated the text and abstract based on the reviewer's suggestions, citing several recent papers and older literature. As discussed in detail in our response, we have not included a mechanistic model. Based on recently published work (Erginkaya et al. 2023), the complexity of the interactions within the HS-H2 network and their poorly defined parameter space make this endeavour beyond the scope of this paper.

We believe that our current version addresses the reviewers' concerns by improving clarity, referencing all relevant previous work, and substantially strengthening our previous results. We have marked all additions and stylistic changes in red in the manuscript.

Next, we will evaluate and discuss each reviewer's comments point by point. The document is long, so please note that we have included a reference list of our responses at the end of the document.

Reviewer #1 (Remarks to the Author):

Summary

This manuscript dives into a technically vexing problem of how gap junctions operate in visual processing. The challenge has been in experimentally manipulating the right gap junction genes in a cell-specific manner. The authors develop a new robust genetic strategy to use an inducible flp construct to disrupt the transcription of a key gap junction gene under the control of a Gal4 driver, allowing for a cell-type-specific perturbation. They then characterize the impact of gap junction perturbation within specific visual integration neurons on the visual receptive field properties of the neurons and on the visual control of walking behavior. They develop a clever walking paradigm in which a fly freely walking within a small arena is tracked in real time and receives continually updated visual stimulation - a sort of “visual clamp” of optic flow on the eye of the freely walking fly. Using this two part approach, they find that gap junctions do not establish the directional selectivity of the neurons, but rather sculpt the extent of their spatial receptive fields. Additionally, connectivity among visual projection neurons is disrupted, along with visual control behaviors that they serve. Notably, disruption of gap junctions in the HS and VS class of visual projection neurons largely eliminates nondirectional small-field avoidance saccades that impose strong nonlinearities in the optomotor stabilization of gaze; without the gap junctions, the avoidance saccades are largely absent, and the operation of the smooth directional optomotor system is highlighted. I find the work novel and compelling. In particular, the genetic reagents and electrophysiology are strong contributions to the literature. I do take issue with the current interpretation of the behavioral findings, which I detail below.

Major comments

Conceptual framework: the authors' claim that the pinwheel stimulus that they project onto one visual hemisphere is "monocular", but claim this is incorrect. In *Drosophila melanogaster*, the region of binocular overlap spans roughly 40-degrees across the visual midline (see Buchner 1976, Heisenberg/Wolf, (Aptekar, Shoemaker, and Frye 2012; Kemppainen et al. 2022). Thus, the stimulus presented here to one half of the visual field, bisected by the longitudinal body axis, is not monocular; rather it is unilateral and activates 50% of the binocular visual field. You are not the first to err in this way - the Duistermars et al. paper cited (Duistermars, Care, and Frye 2012) also conflates monocular and unilateral stimuli. The text shall have to be heavily revised to address this issue because the implications are important (although Figure 7j gets it right!). For example, consider that object tracking saccades in flight are most frequently evoked by stimuli appearing outside of the region of binocular overlap (Aptekar, Shoemaker, and Frye 2012; Mongeau and Frye 2017). Notwithstanding this conceptual criticism, there is a very interesting story here! This manuscript merely examines unilateral-bilateral interactions, rather than monocular-binocular ones.

We thank the reviewer for this important point. This is correct, and we have amended the text to clarify that accordingly throughout the manuscript.

Given the interesting point raised, we also addressed the question of binocularity directly by removing the binocular overlap of the stimulus. Our results show that although the stimuli are unilateral, the integration is to a large degree binocular, not changing quantitatively. Below is the example for Flp-control and Flp-ShakB flies.

The text has been changed accordingly throughout the manuscript. For example:

Line 37-39: "Here, we use the fruit fly as a model to understand how ambiguous motion cues, i.e. unilateral and bilateral optic flow stimuli, are binocularly integrated across the visual field in order to instruct appropriate motor sequences."

Interpretation: some of the visual behavior is misinterpreted. Considering the stimulus geometry, the bilateral pinwheel stimulus projected onto the ceiling of the walking arena subtends less than $\frac{1}{2}$ of the total visual field (the exact subtense should be calculated, see critique item below).

We have expanded (Extended Fig. 2a) to show the exact extent of the overlap, taking the data from (Buchner 1971) digitized by Andrew Straw. We did the same for the stimulus used in electrophysiology experiments to show the exact overlap for stimuli of the physiological experiments (Extended Fig. 11a)

In response to the full bilateral pinwheel stimulus, flies evoke mainly smooth optomotor turns in the direction of image rotation in an attempt to minimize slip and stabilize gaze, as the authors correctly interpret. However, in response to unilateral stimulation, impinging upon roughly $\frac{1}{4}$ of the visual field, the smooth optomotor response largely disappears and a new behavior emerges: saccadic turns contralateral to the projected stimulus, regardless of its direction of rotation. The authors frame their analysis (and their conclusions) relative to the direction of image rotation (syn-directional turns for back-to-front stimuli, and anti-directional for front-to-back stimuli). They therefore surmise that optomotor responses to movement subtending one visual hemisphere are reversed for one front-to-back motion, but not back-to-front. They should instead frame their analysis relative to the position of the stimulus, which seems to me to evoke saccades away from the unilateral stimulus regardless of its direction of motion. This is a more parsimonious interpretation of the data, is clear from the movies and raw trajectories that show saccades (bilateral responses appear smooth, unilateral responses are saccadic), and is consistent with published work showing that flying flies evoke aversive saccadic turns away from small objects regardless of their direction of motion e.g. (Mongeau et al. 2019). What you have discovered here is that gap junctions assembled by LPTCs arbitrate the balance between smooth optomotor gaze stabilization of large-field cues and saccadic avoidance of small-field cues.

We concur that when considering a unilateral moving stimulus, flies exhibit a tendency to turn away from it. This leads to the initial interpretation that flies are avoiding or escaping from unilateral moving objects, irrespective of their movement statistics. However, we respectfully disagree with the reviewer's perspective and did a series of analysis to offer a detailed clarification of our thought process.

To classified teh behavioural responses to FtB and BtF as a distinct change in behavior, and thus, a course control strategy, the movement of the stimulus would need to consistently and predictably alter the mode of action and trajectory. The presented data supports this notion. Let us elaborate:

1. We observe a specific change in the dynamics of the response between Back-to-Front (BtF) and Front-to-Back (FtB) stimuli (Fig. 1k,l,m)

There is a marked difference in the modes of action, with BtF responses displaying smoother movements, while FtB responses are predominantly characterized by saccadic actions (Fig. 1m). This difference is also present when comparing BtF syn-saccades and FtB anti-saccades, as plotted below, which indicates that the response mode is not the same "type" of escape.

2. The time course of the response is also different. While the FtB response increases slowly, in a low-pass filtered manner, and remains sustained. The BtF response has a 'high-pass' response kinetics and adapts quickly. We have plotted both dynamics together in absolute terms to make our point (Figures 1i & g, 8a-d).

3. We performed a series of additional experiments in which we altered the contrast of the unilateral stimuli, and these supporting data are now included in Extended Data Figure 3. These experiments show that the mode of action is highly dependent on the strength of the stimulus. At low contrast, animals turn in the direction of motion for FtB, whereas at high contrast, they turn against it. Interestingly, this average change occurs because more flies change their mode of action from syn to anti, and not because of a smooth transition. We believe this provides compelling evidence against a "simple" avoidance behavior, as it underscores the following key point: Depending on the motion statistics, animals consistently change their mode of action. These last experiments show that an avoidance behavior becomes a stabilization one, showing the flexibility of these behaviours (Extended Data Figure 3).

4. Finally, we directly compared the changes in orientation after stimulus onset for FtB and BtF. Here the difference in polar coordinates becomes apparent (see below). During the first second after stimulus onset, flies turn on average $\sim 45^\circ$ for BtF motion, whereas FtB motion elicits a compensatory movement of about 90° . This shows that compensatory actions differ between FtB and BtF, resembling different course corrections or modes of "escape" (see extended data Fig. 2b).

Note that the orientations of the fly are aligned to 0° at the onset of the stimulus & time increases radially. Black traces, single trials (for clarity, we removed unresponsive trials where the fly did not respond during the first second after stimulus onset and trials where the animal moved in the opposite direction), red, average.

In summary, we politely disagree with the reviewer's statement "gap junctions assembled by LPTCs determine the balance between smooth optomotor gaze stabilization of large-field cues and saccadic avoidance of small-field cues" as the best description because: (i) we would not classify our stimuli as small-field; they encompass 25% of the fly's entire visual field. Secondly, we demonstrate a shift in the selection of actions, characterized by distinct saccadic and smooth properties as well as clear variations in dynamic properties. Finally, we establish that "escape" can transition to "stabilization" due to modifications in stimulus contrast, underscoring the adaptability of these behaviors rather than an all-or-nothing "escape". In our view, this subtle distinction is important.

We have further clarified this point by addressing it directly and incorporating additional panels mentioned above to enhance the clarity.

Line 120-134: "Although responses to FtB and BtF motion could be interpreted as escape behaviours from local motion as shown during tethered flight (Mongeau et al. 2019), the kinetics and relative contribution of smooth and saccadic responses for FtB and BtF motion differ drastically (Fig. 1i, m), altering the relative number of syn- and anti-saccades and their respective amplitudes and velocities (Extended Data Fig. 2). This indicates that FtB and BtF motion drive different modes of action, as evidenced by the difference in the straightness of the trajectories (Extended Data Fig. 2c, d). The stimulus dependence of the FtB anti-saccadic response becomes apparent as the stimulus strength is reduced. At low contrast, animals begin to reverse their turning direction for FtB motion, from avoidance to stabilisation (Extended Data Fig. 3a-c). We observed a similar dependence when presenting a different type of optic flow stimulus, the 'star field' stimulus (Weir and Dickinson 2015), an overall weaker optomotor-inducing stimulus. Consistent with our low contrast results, we observed that half of the flies showed a robust anti-

response while the other half did not (Extended Data Fig. 3e-g). These results show that the robustness of the anti-saccadic response depends on the strength of the overall optic flow and demonstrate the existence of stereotyped behavioural adaptations to nuanced changes in stimulus properties.”

How might LPTCs arbitrate such a switch? I note that a recent study demonstrated that unilateral optogenetic activation of HS neurons evokes counter-directional steering in flight (Kim et al. 2023).

Based on previous work by Haikala et al. 2012 (Haikala et al. 2013) in flight and Busch et al. 2018 (Busch, Borst, and Mauss 2018) in tethered walking flies, unilateral HS cells result in a turn toward the direction of stimulation, not counter the responses. In Kim et al. 2023 (Suppl. Fig. 1), these experiments were repeated with qualitatively similar results. Perhaps our confusion comes from the nomenclature. Kim et al. study mainly the role of LPLC2 and DNp06, not HS cells, in looming avoidance behavior, cells that elicit counter-turns when optogenetically activated.

It is, unfortunately, unclear what may trigger the transition and how these pathways might interact. The newly added data indicates that the response can shift from avoidance to stabilization based on the stimulus contrast, suggesting a likely non-linear process like spike generation. Given that H2 is a spiking neuron, one possible mechanism is that only strong depolarizations in HS cells can drive spikes in H2 via gap junctions, which will drive the counter-saccadic response. However, it is challenging to understand how this interaction fits into the entire LPTC network with our current knowledge. The Chiappe lab recently uncovered several other neurons mediating intricate interactions between HS and H2 cells through GABAergic interneurons, specifically bIPS (Erginkaya et al. 2023). These interneurons are undoubtedly part of the equation. Still, the lack of clear knowledge regarding the circuitry's electrical connectivity strength presents a challenge in defining circuit mechanisms due to too many unknown parameters.

Also, the ratio of HSN/HSE activity in your behavioral experiments (compare dorsal RFs Figure 5a/b) becomes considerably smaller with Flp-shakB. I.e. the transition from smooth slip-stabilization to saccadic avoidance is coincident with increased HSE activity in the dorsal RF, as well as all of the other connectivity changes indicated in Figure 6.

This is correct and we can't causally separate the role of decreased contralateral activity and the increased HSE activity in the dorsal region. However, we believe that despite not covering the entire RF, we are strongly stimulating HSE cells during behaviour. It has been shown in the past that smaller moving stimuli already saturate the responses in the axonal termini of HS cells (Elyada, Haag, and Borst 2013). Based on these results, the parameters of our behavioural stimulus should drive HSE cells in both genotypes strongly despite the RF differences.

As the reviewer pointed out, there will be numerous changes in the circuitry. This obstacle exists with any circuit manipulations. Any manipulation can, in principle, alter the network's dynamics. Thus, as the reviewer well noted, Fig. 6 and 7 can not be used as a causal argument to directly link these changes with behavior directly. We use these figures to demonstrate that the expected changes from gap-junction manipulation can be observed, such as the blockage of contralateral interactions (Fig.6). Figure 7's data provides us with a qualitative understanding of events that aligns with Fig. 6 and additionally demonstrates the continued coupling to DN.

Another consideration is that gap junctions operate immediately, whereas chemical transmission requires calcium accumulation in the presynaptic terminal. Perhaps saccades are organized by rapid communication across LPTCs, whereas smooth optomotor responses are coordinated by integrated electrotonic signals. I look forward to your interpretations!

Comparing the time courses of chemical and electrical synapses in the context of these behaviors is difficult, although we agree that it is very interesting. Chemical release can occur as rapidly as 2-3 milliseconds, often accompanied by pronounced nonlinearity that can function as a high-pass filter. In contrast, the impact of electrical synapses depends on the strength of the connectivity (resistance) and, without additional mechanisms, will exhibit low-pass filter characteristics. The time constant of this electrical filtering between VS cells has been estimated to be ~1.4 ms (Haag and Borst 2004); although faster than chemical release, not dramatically different. We have now added a new set of experiments testing the role of chemical synapses using *shibire^{ts}* (Fig. 3). Interestingly when blocking chemical synapses, we also observe the same phenotype as when blocking electrical synapses. This indicates that the close interaction between electrical and chemical connectivity is required to generate these computations in concertation. This is consistent with the HS-H2 chemical connectome analyzed recently by the Chiappe lab (Erginkaya et al. 2023). Thus, determining the role of one cell in this behavior is tricky, since many different perturbations will directly influence the dynamics of this particular HS-H2 network. We would like to give a concrete biophysical interpretation of these results, but we are afraid that at its current state, it will be more guesswork than we feel comfortable with.

Figure 1: Using the same color scheme for saccades (d) and smooth pursuit (e) is confusing. Perhaps show saccades with narrow vertical black raster marks? Or show a saccade raster? The graphs in panel i are really nice - separating the component of smooth pursuit from saccades. This confusion occurs again in Figure 7 in which turning direction is indicated in green/pink, and then below the smooth response is in pink (or similar hue).

Plotting saccades on top of the angular speed plot resulted in a visually cluttered plot, so here we have added a raster plot for just saccades. We changed the color scheme we use for distinguishing smooth and saccadic responses (orange/purple) in Fig 1i to avoid confusion with syn and anti-saccades (pink/green).

Figure 5. I think it is worth explicitly stating that your stimuli only activate only the uppermost region of the receptive fields. It would also be worth the effort to calculate the angular size of the projection of the behavioral stimuli onto the retina and receptive fields of the relevant LPTCs.

We have mentioned this now in the discussion directly as part of our interpretations of our results.

Line 723-728: "The enlargement of the ipsilateral RF in Flp-shakB compared to Flp-control is unlikely to be involved. Our behavioural arena activates the dorsal half of HSE in Flp-control animals. Increasing the ipsilateral RF size, as in Flp-shakB (Fig. 6a-f), should increase the strength of the counter-directional response to unilateral FtB motion. We observed the opposite, suggesting that the contralateral input is necessary for instructing the counter-directional responses."

Visual field occupied by behavioural stimulus compared with Receptive Field of HSE for Flp-control flies (at 30% cutoff, same as lighter blue shade in Fig 6)

L441 and elsewhere: the terminology is confusing - “turning toward” is generally relative to the direction of motion not the unilateral position of the stimulus. Rectifying the smooth directional responses to full-field and saccadic avoidance responses to hemi-field stimuli will mitigate this issue.

We changed the sentence to: *“Likewise, Flp-shakB flies turned with the direction of BtF motion, i.e. against the side of stimulation (Fig. 8cii, Supplementary Video 4).”*

Minor comments

Title: remove ‘binocular’ and perhaps retitle: Gap junctions coordinate transition from course stabilization to avoidance

We understand the concerns raised based on the previous version of our manuscript. However, given that we now explicitly show (i) that the trajectory changes for FtB and BtF reflect different behavioural strategies (Figure 1, Extended Figure 2-3) and (ii) that the interaction is binocular, not just bilateral (Extended Figure 14), we believe that the title is appropriate and would prefer to keep it as it is.

I do not believe that the figures and panels are called out in order. Is this a requirement for Nature Comms?

Corrected

L79 cite (Strauss, Schuster, and Götz 1997) who implemented a very similar virtual reality system in freely walking flies, and the biased open loop during free turns approach is also similar to (Rimnieceanu, Currea, and Frye 2023) in magnetically tethered flight

Cited

L146: “all and only” I think needs to be reworded

Reworded to: *“We specifically targeted horizontal system (HS) and vertical system (VS) neurons...”*

L272: “tethered” in this situation is misleading, implying behaving flies. Say “restrained” or “immobilized head-fixed”

Changed.

L417 change to “Crucially, the reduction of contralateral electrical connectivity enables us to test the involvement of LPTC-mediated behavioral role of gap junctions.”

Changed.

L440-445 revise with framework of smooth optomotor stabilization of bilateral motion direction vs saccadic orientation away from unilateral stimulus position

As in WT flies, it is correct that Flp-stop flies turn away from the stimulated hemisphere. However, their actions while doing so differ drastically if presented with FtB or BtF motion. We reformulated the text to make this point clear.

Line 517-522: "Likewise, Flp-shakB flies turned with the direction of BtF motion, i.e. against the side of stimulation (Fig. 8cii, Supplementary Video 4). However, in contrast to Flp-control flies that saccadically turned against the direction of FtB motion, Flp-shakB flies reversed their response direction and mode of action, turning on average smoothly with the direction of the stimulus, i.e. towards the stimulated hemisphere (Fig. 8ciii, Extended Data Fig. 12, Supplementary Video 4)."

L36-41: These statements, thus the premise, about equivocal motion cues are vague and incomplete. Only locally are motion cues equivocal. Revise for specificity and clarity.

Changed to: "However, the accurate interpretation of visual motion cues can be equivocal when only local information is available."

L88-89: In walking flies. Be specific. Recentering saccades have been described only in flying flies as resembling the fast-phase of an optokinetic nystagmus (see Mongeau & Frye, 2017)

Changed accordingly.

L92: be more specific adding "major" fly's body axis

Added.

L75: stimuli are not monocular

Changed to unilateral stimuli across the manuscript.

L110-111: Wouldn't the number of saccades be a more straightforward measurement of the saccadic behavior? E.g. rasters mentioned above

Unfortunately not, since the amplitude of syn and anti-saccades differ. We have stated this in the text:

Line 115-119: "The relative contribution of smooth and saccadic turning is evident across flies for each stimulus (Fig. 1k,m). Interestingly, although the number of anti-saccades per trial is higher (Extended Data Fig. 2d), syn- and anti-saccades contributed equally to the full-field optomotor responses (Fig. 1k) due to larger syn-saccadic turns (Extended Data Fig. 2e, f)."

For completion, we have plotted the saccadic events below.

L146-147: “all and only”, VT058487-Gal4 targets other neurons in the brain such as FB, AVLP and PLP (to say the least). Remove the word “only”

Removed. It is important to note that the expression depends on the reporter used. For Kir2.1, other neurons are very faint or not visible at all compared to LPTCs which have a strong expression. This is why we don't see them in the maximum intensity projection (see answer below).

Fig.2A: Are you sure it is a MIP image of the brain? Looks more like a single plane

No, it is indeed a maximum intensity projection of the entire stack (brain).

L159: There is no Fig.2I

Corrected.

L207-208: for clarity, please use the nomenclature shakB[FlpStop-ND] and shakB[FlpStop-D] and avoid the abbreviation Flp-control and Flp-shakB. Both in the manuscript and Fig.3-4

In the first version of our manuscript we sent out for comments we used that nomenclature, but we were strongly encouraged to change it to Flp-control and Flp-shakb for readability before submitting. This, we noticed, makes the manuscript easier to read. If the reviewer does not mind, we would suggest keeping the simplified nomenclature for clarity.

L211: next time use shakB[2] to define the mutation and not the superscript 2 because it is difficult to distinguish it from a reference. Alternatively, write “shakB2-mutant”

Changed to *Shakb[2]*.

L414-417: Since there is no effect in the connections with downstream neurons, then why is this a good model for studying gap junctions? This section is confusing. Maybe remove “making Flp-shakB a favorable model for studying the behavioral role of gap junctions”

Removed.

L417-419: I would be more specific adding “reduction of contralateral electrical connectivity with H2 cells”

We change the paragraph for clarity.

Line 489-494: “However, this reduction was limited to connections between LPTCs, as shown by the reduction in contralateral dye coupling with H2. Connectivity between HS cells and postsynaptic interneurons, descending neurons and motoneurons remained largely intact. Crucially, the reduction in contralateral electrical connectivity allows us to test the involvement of the LPTC-mediated behavioural role of gap junctions.”

L448-449: the conclusion that the binocular interactions mediated by gap junctions are responsible for the behavior is a bit hasty. The flies used for this experiment are not expressing shakB (A and E isoforms are probably not even expressed in the adult brain) not only in the HS neurons (for those we have hints of interhemispheric connections through H2) but also in many other visual neurons that could play a role in this FtB counter-directional response. If it is true that the response in shakB[FlpStop-D] flies looks similar to HS/VS>Kir2.1 flies, that doesn't mean that shakB in those neurons is directly responsible unless the shakB is selectively disrupted in those neurons. In other words, that "same" effect might be the result of two independent pathways/mechanisms

It is not readily apparent how the authors can discriminate between an effect due to the loss of electrical coupling among HS cells and the effect due to the loss of interhemispheric connectivity. In support of the conclusion that H2 cells are the connector, it would be necessary to silence them and recapitulate the lack of FtB counter-directional response

We agree with the reviewer that our experiments don't allow us to make definitive statements. To do so, we would need to specifically disrupt ShakB between the HS and H2 synapse while leaving all other electrical connections intact. Unfortunately, cell type specific ShakB ablation would require a specific driver line with early enough Gal4 expression to disrupt ShakB in a cell specific manner (see Figure 4), and furthermore, a synapse specific disruption is beyond the current capabilities of the field.

During the revision process we also performed experiments in which we silenced the chemical output. This was important because a recent paper has described this HS-H2 connectome in detail (Erginkaya et al. 2023), and described a strong chemical synaptic connectivity. We therefore replicated the experiments using shibire to inhibit chemical communication in HS/VS or H2 cells. We were surprised to find that inhibiting the chemical output of HS cells greatly impeded the counter-directional behaviour of the FtB. We then performed an identical experiment using the only "specific" driver line we could find, based on a recommendation from the Chiappe lab, which has some off-target expression in the VNC. Interestingly, also blocking chemical synaptic release of H2 inhibits the counter-directional behaviour of the FtB (Fig. 3, Extended Data Fig. 5). Please note that we could not repeat the experiments with Kir2.1 in H2, since the expression level was too low in H2 cells, only visible with antibody staining. For shibire, a dominant negative disruptor, lower expression levels appear to be sufficient to disrupt the system.

Although we can't pinpoint the circuit mechanisms, together with our ShakB manipulations (Figures 4-8), our experiments suggest that the extended chemical and electrical HS-H2 network is required.

We have refined our interpretations accordingly:

Line 728-731: "This is supported by our H2 block experiments in which the anti-saccadic response is strongly diminished (Fig. 3). Consequently, our results provide compelling evidence implicating the HS-H2 network (Erginkaya et al. 2023) in the proper selection of smooth and saccadic turns during navigation."

Line 737-743: "What are the underlying circuit motifs and biophysical implementations underlying this non-linear, gap-junction-mediated binocular interaction? Although our research points to the

relevance of the interaction between HS and H2 cells, it is important to consider that this computation occurs within a circuit context that necessarily involves additional circuit components and their chemical synaptic interactions. This circuit has recently been anatomically characterized (Erginkaya et al. 2023), involving several important cell-types, e.g. the recently described bIPS and descending neurons controlling saccadic turns (Schnell, Ros, and Dickinson 2017). ”

L501-503: This seems to be an overstatement

Modified to (Line 599-602): *By establishing and thoroughly characterising a new shakB mutant line (Figs. 4-8, Extended Data Figs. 6-14), we could link the behavioural deficits observed by silencing LPTCs (Fig. 2) with the loss of heterolateral electrical connections between HS and H2 cells (Figs. 6-8).*”

L506: state-of-the-art

Sentence has been removed according to reviewer’s #2 suggestion.

L1109-1110: “The entire behavioral setup is placed inside a custom designed temperature-controlled compartment that maintains the internal temperature at 27°C”. The same exact sentence is repeated at L1129-1130, which is a bit odd. Also, does this mean that the experiments were run at 27°C?

Yes, the temperature of the experimental chamber is 27C. This slight increase in temperature from our room temperature made flies more active. This is a known fact (see Supl. Fig. 1 from. (Bahl et al. 2013)). We, however, avoided increasing the temperature to 34C, as used in previous studies.

The use of only male flies represents a marked divergence from the vast majority of the literature on fly visual behavior (females are larger, more easily manipulated for physiology and behavior). Did the authors test female flies as well in response to half optic flow patterns? If so, how was the response?

The response for females is consistent to the responses for males across genotypes, although a bit weaker over all. We have added an additional panel to the Extended Fig. 13g and and direct comparison below.

Regarding the flies rearing conditions: Mano et al., 2023 showed that the temperature affects the anti-directional turning behavior. At 25°C flies show less anti-directional behavior than flies reared at 20°C. Since the authors reared their flies at 18°C, did they also try flies reared at different temperatures? If so, how did temperature affect the main findings of the paper?

We repeated the experiments with flies raised at 25°C and found a larger variability in their responses to FtB motion (Extended Figure 13 and see below for a direct comparison), which is

consistent with the larger variability in the connectivity we observed in our dye coupling experiments across flies raised at 25°C. On average (see below), flies do not respond. However, if we reared the flies at 18°C and then transferred the adults to 25°C for at least 24 hrs the experiment, their response was the same as what we observed for flies reared solely at 18°C. This suggest that the effect of temperature on this behavior is due to changes during development but not during adulthood. Similar effects have been shown before (Kiral et al. 2021), raising the question of what the proper rearing temperature should be for studying behavior. Given that we see a more consistent behavioural responses when rearing flies at 18°C, we decided to use this temperature to allows for more consistent circuit dissections.

Change saccade strength in saccade amplitude please (e.g., Extended Data Fig.2d)

Change

Supplementary table 1: please separate the genotype name from the number of flies used:

Change

Figures:

Fig.1: please report that everything related to angular speed and cumulative angular distance is not absolute but relative to the stimulus. In other words, positive and negative values are corrected based on the direction of the rotating pinwheel (i.e., syn- vs counter-directional).

Also, what is the sample size? Please report N of flies recorded (at least) and how many trials per fly in the legend not only in the supplementary table 1

We describe this now directly in the main text.

Line 94-95: “- note that the behavioural responses are not absolute but presented relative to the stimulus direction.”

N for each fly is now reported for all experiments

Fig.1c: report the direction of the rotating stimulus please

Added to the legends -

“c. Example trajectory of a CantonS fly during a typical trial of a clockwise rotating stimulus.”

Fig1d: what does the dashed line mean? I suppose the starting of pinwheel rotation but you need to report it. Also, saccades may be better defined as spontaneous saccades

Added to the legends. *“Dashed line indicates the beginning of the rotation.”*

Fig.1m: the average of trials per fly is removing a lot of useful information. Could the authors just plot all the data points collected?

All datapoints of Fig. 1 were added to the Extended Figure 2 for further scrutiny.

Fig.1i: I don't understand this plot. How was the plot generated? What is delta angular speed? Do the contours represent the mean angular speed considering exclusively smooth optomotor (red) or saccadic period (blue)? Explain it better in the legend please

Add the following sentence to the figure legends to clarify the panel:

"i. Stacked plot showing mean angular speed and the contribution of smooth and saccadic turning. Saccadic response are calculated by taking a 300 ms window centered at the saccade peak, whereas smooth responses are defined as the remaining average response."

Fig.1i: two (i) with two different descriptions, very confusing. Correct as (j)

Corrected

Fig.1k: summary plot: scatter plot? (same for Fig.1l). Also, add mean \pm SEM of the mean per fly

Added

Fig.1l: What does "turns made" mean? the response amplitude? Please explain it better

Changed "turns made" to cumulative angular displacement

Fig.2: using UAS-Kir2.1 flies as control is not the best of the choices for the genetic background. Especially if you use as experimental flies not only bearing a Gal4 but a tsh-Gal80 as well. It would have been better to use flies at least with tsh-Gal80/+;UAS-Kir2.1/+ and ideally with tsh-Gal80/+; UAS-Kir2.1/empty-Gal4 (BDSC # 68384). The early angular velocity of the UAS-Kir2.1 in response to full-field rotation (Fig.2e) is on average twice the angular velocity of Canton-S (Fig.1h)

Please plot the raster plot of the angular velocity as done in Fig.1 and Fig.7.

We have added these plots to the respective Extended Figures 3, 4, 5, 13 & 14.

Extended Data Fig.2c: angular speed (top) and turns (bottom), please correct.

Corrected

Fig.4i: the moving edges cartoons are confusing, why is there depicted two arrows and a bar in the middle? That's not an edge, that's a bright bar moving over a uniform gray background with ON and OFF transitions. Please fix it. Same for Fig.4j

Modified.

References

Aptekar, Jacob W., Patrick A. Shoemaker, and Mark A. Frye. 2012. "Figure Tracking by Flies Is Supported by Parallel Visual Streams." *Current Biology*: CB 22 (6): 482–87.

Duistermars, Brian J., Rachel A. Care, and Mark A. Frye. 2012. "Binocular Interactions Underlying the Classic Optomotor Responses of Flying Flies." *Frontiers in Behavioral Neuroscience* 6 (February): 6.

Kemppainen, Joni, Ben Scales, Keivan Razban Haghighi, Jouni Takalo, Neveen Mansour, James McManus, Gabor Leko, et al. 2022. "Binocular Mirror-Symmetric Microsaccadic Sampling Enables *Drosophila* Hyperacute 3D Vision." *Proceedings of the National Academy of Sciences of the United States of America* 119 (12): e2109717119.

Kim, Hyosun, Hayun Park, Joowon Lee, and Anmo J. Kim. 2023. "A Visuomotor Circuit for Evasive Flight Turns in *Drosophila*." *Current Biology*: CB 33 (2): 321–35.e6.

Mongeau, Jean-Michel, Karen Y. Cheng, Jacob Aptekar, and Mark A. Frye. 2019. "Visuomotor Strategies for Object Approach and Aversion in *Drosophila Melanogaster*." *The Journal of Experimental Biology* 222 (Pt 3). <https://doi.org/10.1242/jeb.193730>.

Mongeau, Jean-Michel, and Mark A. Frye. 2017. "*Drosophila* Spatiotemporally Integrates Visual Signals to Control Saccades." *Current Biology*: CB 27 (19): 2901–14.e2.

Rimniceanu, Martha, John P. Currea, and Mark A. Frye. 2023. "Proprioception Gates Visual Object Fixation in Flying Flies." *Current Biology*: CB 33 (8): 1459–71.e3.

Strauss, R., S. Schuster, and K. G. Götz. 1997. "Processing of Artificial Visual Feedback in the Walking Fruit Fly *Drosophila Melanogaster*." *The Journal of Experimental Biology* 200 (Pt 9): 1281–96.

Reviewer #2 (Remarks to the Author):

The authors report a study on the heterolateral network connectivity of lobula plate tangential cells (LPTCs) enabling the integration of binocular visual motion information in *Drosophila*. They combined (i) behavioural studies and (ii) electrophysiology with (iii) genetic manipulations of certain pathway elements involved, with a focus on electrical connections (gap junctions) between directional-selective LPTCs, specifically horizontal cells HSN/HSE and the heterolateral H2-cell. They found that disabling gap junction coupling using Flp-ShakB mutants has a qualitative impact on the way in which visual motion information from both eyes is combined.

Generally, the topic is certainly of general interest. It was approached using a combination of demanding methodologies and the data are appropriately presented in well-designed figures. There is, however, an imbalance due to a substantial weight on the genetical methods used (including their validation) while a differentiated framework for addressing the scientific question is missing. All the authors provide along the latter is a qualitative account on a generic stabilization reflex (optomotor response), without defining the functional significance for specific behavioural tasks. In my view it is next to impossible to interpret experimental results without knowing what the functional context is (see further comments below).

In the following I list several major issues the authors should address before I would be able to provide a recommendation on whether or not their work may be suitable for publication in *Nat Communications*. In addition to the comments below, I will provide an annotated version of the submitted manuscript including further comments I would like the authors to consider when working on the revision.

My recommendation would be, in any case, to engage on a major revision.

- 1) Clearly define what is currently known about optic flow-based self-motion estimation and control and how the observed behaviour supports specific aspect of “visuo-motor control”
- 2) Focus on the gaps in understanding the functional role of gap junctions in support of specific visuo-motor control tasks. This is where the merits of your work are.
- 3) Motivate the choice of the methods you apply and discuss the potential limitations.
- 4) Revise your terminology keeping a general readership in mind.
- 5) Include a functional interpretation of your experimental data in the context of visuo-motor control. If possible, and in case you decided to talk about computational aspects, include at least a qualitative model of what functional impact the presence/absence of gap junctions has at the network level and how it is related to the observed behaviour.
- 6) Revise the abstract and the discussion section to avoid overstating the conclusions based on the experimental results you obtained.

For further guidance I have added a sentence-by-sentence review of your abstract at the bottom of this report.

Thanks for pointing out better ways to describe our results and all the other comments in the pdf. We have adapted the manuscript accordingly. Changes are marked in red in the revised manuscript.

Major comments

Editorial:

- The manuscript contains area-specific terminology that is not introduced/explained, which makes it difficult for non-specialist readers to follow the logic of the approach (e.g. innexin; intronic, etc.).

Thanks for pointing this out. We have tried to define all terminology as clearly as possible, so that non-specialist readers can follow the logic without the need of previously knowing a specific vocabulary.

- Also related to terminology: expressions such as “computation” and “algorithm” only make sense when used in connections with operations that are required to control a specific task. Normally, a model helps to formally describe the required operation and to constrain the possible solutions which transform a given set of inputs into the observable systems output. My recommendation for the authors would be to either remove those expressions or to introduce more formal description that would justify the used of such terms.

We have chosen not to define a specific mechanistic model because the parameter space is not yet properly defined, making it more guesswork than real science (see our response to reviewer #4 for the limitations we face that make it difficult to define a circuit model at this time). However, we believe that the terms "computation" and "algorithm substrate" are appropriate and commonly used to describe similar neural computations. See (Marr 2010; Krakauer et al. 2017). Based on this definition, computations could be the observable behavioural changes (e.g. optomotor response) and the algorithm (e.g. the Reichard detector) and the algorithmic substrate could be the cell types involved. Unfortunately, we can't define the implementation, i.e. the biophysics and exact processes of the circuit involved, but we can identify the circuit components.

- Similarly, the term “decision strategy” is hardly justified given (a) the methodology used, (b) the level of the visuo-motor pathway studied, and (c) the lack of an interpretation in the context of a specific optomotor control task (see further comments in amended pdf-file).

We appreciate the comment and understand that defining this as a decision will be contested. Thus, we have changed the terminology.

Previous work:

How the visual system obtains unambiguous information on self-motion based on the processing of optic flow has been studied extensively in the past (theory/modelling: e.g. Koenderink and van Doorn 1987; Dahmen 2001). In this context, the role of heterolateral LPTCs had been studied in blowflies (e.g. Hausen et al. and Krapp et al.). Substantial contributions to the network of LPTC connectivity in blowflies and *Drosophila* comes from Borst et al., as well as other groups working on fly vision. Even the electrical connectivity between HSN/E and the H2-cell was included in a paper by Borst and Weber (2011).

The introduction includes some relevant citations. But my impression was that it did not bring across the level of understanding that has been established already. Neither the general question Pokusaeva et al. are addressing is novel, nor the finding that the H2-cell contributes to the binocular integration of visual motion information in the context of self-motion estimation. The

remit of their manuscript lies in the finding that the failure of gap junctions can alter the properties of neuronal circuits (and thus the behavioural output) in an unexpected, non-linear way. It also shows (ones more) that the phenotype of genetic manipulations depends on genetic background and on specific mutants used (e.g. ShakB vs ShakB2). This, in turn, makes it particularly difficult to explain the results of the manipulation and come up with a functional interpretation (see further comments in annotated pdf-file).

We are well-versed in the extensive and detailed literature on this subject and regret that, due to space constraints in our scientific paper, we couldn't incorporate all the interesting aspects that have previously been presented in this field. However, we made an effort to reference the most pertinent papers. For instance, Farrow et al. in 2006 described the nonlinear interaction between H2 and HSE cells prior to Weber et al.'s work in 2011. However, we have added several new references that we should have cited previously, thanks for the suggestions. We hope that we haven't missed any relevant one this time.

It is important to note that much of the previous literature has primarily focused on the physiological computations resulting from LPTC interactions. In many of these cases, the functional relevance at the behavioral level remained speculative, as establishing direct links proved challenging. We believe that the paramount contribution of our work lies in connecting circuit components with behavior. We also acknowledge the reviewer's point that the questions we are addressing are not entirely novel, as they have been explored for over half a century using various approaches. However, we respectfully disagree with the notion that our findings lack novelty for the following reasons:

(i) We performed the experiments with freely walking flies, as opposed to tethered flies in most previous studies, allowing us to get a more naturalistic responses. This approach unveiled a robust and novel behavior. (ii) We establish a connection between behavior and specific circuit components, namely HS and H2 cells. (iii) We identify that this interaction is mediated by gap junctions and chemical synapses, elucidating a behaviourally relevant interaction for the larger H2-Hs network (Erginkaya et al. 2023). (iv) We introduced a novel fly line, which demonstrates that not all gap-junction connections are equal, as we can disrupt connectivity between LPTCs while keeping other electrical connections intact. (v) Finally, we expand the repertoire of course control beyond stabilization and avoidance paradigms to encompass more subtle changes in course that are dependent on stimulus properties.

While it has been demonstrated that HS and H2 cells interact in a nonlinear fashion electrophysiologically in blowflies (Farrow, Haag, and Borst 2006), and that optogenetic activation of HS cells elicits behavioral responses (Haikala et al. 2013; Busch, Borst, and Mauss 2018), we are not aware of any study that illustrates how interactions between LPTCs shape behavior. Although insightful theoretical perspectives can be found in the literature, such as (Cuntz et al. 2007), proposing how electrical communication between the VS cell network can enable accurate estimation of rotational flowfields, many of these ideas have not been experimentally tested due to the lack of specific tools and appropriate behavioral assays. As a field, there have been few instances in which LPTC activity has been directly linked to its behavioral functions (Kim et al. 2017), and our study serves as one of the examples paving the way for further exploration.

Finally, although it is important to recognise behavioural differences due to genetic background, we believe that the comparison between Flp-control and Flp-ShakB is not due to this - which is an important point. In this particular case, Flp-control and Flp-ShakB have identical genetic backgrounds and an identical insertion that is oriented in either the disruptive or non-disruptive direction. The observed differences are therefore not due to any changes that might have been introduced by the old mutant generation procedure, but to a specific disruption of ShakB.

Here are some questions related to this point:

- Flp-control produces stronger behavioural responses than CantonS wild type flies. How do you explain this finding?

It's well-recognized that the genetic background of flies can exert a significant influence on their behavioral traits. Consequently, in many instances, it becomes necessary to conduct multiple generations of backcrossing to minimize this variability. While we do observe variations in the strength of the response across different wild-type (WT) fly strains, it's crucial to emphasize that qualitatively, the response remains consistent.

It's worth noting that the genetic backgrounds of the Flp-control and Flp-ShakB flies are identical, but different from the CantonS wild-type strain. It is therefore expected that quantitative comparisons between these strains and CantonS may not be identical. Qualitatively, however, they match very well.

- The lack of contralateral input (Flp-shakB) introduces smooth optomotor responses upon FtB motion – while Flp-control mutants generate anti-saccades. How do you explain this finding?

This question is indeed quite intriguing, as it pertains to the circuit level, and unfortunately, we lack a comprehensive answer in that regard. We think that smooth optomotor response and anti-saccades are both present in response to a full-field response. The contribution of these two types of behavior is most likely governed by relative strength of H2 and HS output, as supported by our recent findings, which show that low-contrast FtB motion alone is insufficient to trigger the counter-saccadic response, but sufficient to trigger smooth stabilizing turns. In addition, Erginkaya et al. (2023) recently show that these neurons have recurrent connections which could be a possible mechanism by which the responses of these neurons are compared. By disrupting electrical connections while leaving chemical synapses intact, we are altering the relative strength of the output of these neurons, causing a switch in the relative abundance of smooth optomotor response and anti-saccadic response. It is likely that the recently described DNae014 (Ros, Omoto, and Dickinson 2024), a neuron involved in the regulations of saccadic responses, since it appears to be connected to the HS-H2 network via putative DNae02.

https://codex.flywire.ai/app/search?filter_string=%7Bupstream%7D+720575940631759663+%7Band%7D+left+%7Bdownstream_region%7D+720575940629153020&sort_by=&page_size=10&data_version=783#

However, to provide a more detailed and mechanistic response to this question, a subsequent focused study dedicated to exploring the integration of HS and H2 projections will be necessary.

- Silenced HS and VS-cells: flies still show optomotor responses. How would you interpret these observations? Did you consider:

(i) the possible impact of efference copies?

(ii) the possible impact of other, e.g. position-dependent visual pathways?

These are indeed intriguing points to take into consideration. It has been demonstrated that blocking HS/VS cells results in relatively minor changes. For instance, Kim et al. in 2017 observed only a reduction in head optomotor movements and a minimal change in optomotor response. According to their model, efference copies would be necessary for correcting re-afferent stimuli, particularly during saccades. Consistent with their findings and to the best of our knowledge, we believe that an efference copy may not be the most straightforward explanation.

Regarding the second point, we argue that in addition to the position-dependent visual pathways, recent electron microscopy reconstructions have revealed the existence of over 60 LPTCs (Zhao et al. 2023), all receiving inputs from T4/T5 direction-selective cells. This is a number much higher than previously appreciated. Given this multitude, it is not surprising that a variety of mechanisms likely come into play. For completion, we did an experiment with T4T5-Kir2.1 flies (effectively silencing the motion-dependent pathway while keeping the position-dependent pathway intact (Bahl et al. 2013)) and the responses are gone (see below). We haven't included this in the paper.

Other comments:

- Figure 3c: only HS-cells? Fig3c shows VS-cells.

This might be a confusion. We are using either the VT058587 or DB331 lines for our analysis, which always include both cell types (HS and VS). Please note that the VT058587 line shows slightly different off-target expression when using GFP or Kir2.1 as a reporter. With Kir2.1, only HS and VS cells are strongly expressing the reporter.

- 60Hz stimulus presentation during e-phys. What is the flicker fusion frequency in Drosophila? It may have an accumulative effect on signal integration - even at slow response dynamics.

You've made a keen observation. Our stimulus generates mild flicker fusion artifacts, since we use a LED DLP projector. While we refresh each frame at 60 Hz, our stimulus is presented at a 120 Hz flicker rate, as the blue and green LED lights are presented sequentially. Due to technical constraints preventing an increase in the refresh rate in the closed loop system. However, for the points made in this paper, the critical factor lies in the relative differences between the stimulated

sides. These would flicker at the same rate. Finally, we decided to keep the stimulus properties to maintain consistency with our behavioral experiments.

• Did you consider head movements? How would they affect your results?

We repeated the behavioral experiments gluing the head with the abdomen leaving the flies overnight in food to recover and screen for any possible damage. Although we cannot exclude the role of head movements, stabilizing the head does not change BtF and FtB kinematics quantitatively, dramatically.

We have added this data to Extended Fig. 14 for Flp-stop and Flp-shakB.

• A clear-cut interpretation of the smooth and saccadic syn- and anti-rotations should be provided. Specifically, what is the functional interpretation of the saccadic anti-rotations in non-genetically modified animals?

Thanks for that note. We tried to make that interpretation in panel Fig 8j, but realized that we didn't expand on it in the discussion, which we have now done:

Line 696-704: "Therefore, the loss of the HSE-to-H2 connection, as well as perturbations in the larger HS-H2 network (Erginkaya et al. 2023), would restrict the analysis of the optic flow to individual hemispheres, limiting the ability of the fly to differentiate global optic flow patterns and instruct adequate steering behaviours. This can be relevant if the optic flow is global or local (Dahmen, Franz, and Krapp 2001). For example, when navigating in the natural environment, a global displacement caused by a gust of wind should elicit a compensatory change in direction of motion, whereas local motion cues caused by approaching objects in the visual periphery would require a corrective turn in the opposite direction to avoid a collision (Fig. 8j)."

• Related to L 460 and the precious comment: I am missing a functional interpretation of your findings. From what I have learned so far, it looks as if the non-modified connectivity enables a higher degree of behavioural variability (smooth syn-turns + saccadic anti-turns). So, I am not

sure I understand why you are talking about "constraining the behaviour". To turn these properties into something behaviourally relevant, would the fly have to be able to control the conductivity of the gap junctions?

This could be a misunderstanding. We don't claim that the WT or unmodified connectivity gives a greater degree of behavioural variability, but it expands the mode of action depending on the stimulus presented. For example, FTB and BtF evoke both strong and reproducible responses that are opposite to the direction of movement, meaning anti for FTB and syn for BtF. We have removed the last part of the sentence about "constraining behaviour" to avoid any misunderstanding.

- Related to L. 515: this sounds all fine. But besides mentioning optomotor responses, you have not provide a functional interpretation of the various behavioural observations (see earlier comment). How can you hope to understand the "underlying neuronal mechanisms" if you don't know what behavioural function they are supposed to control?

The point raised is indeed important and valid, and we acknowledge the complexity of defining the "goal" of behaviors. As previously mentioned, our interpretation of the behavior may stem from a somewhat naïve perspective. We are operating under the assumption that animals need to control their course and can do so by either turning toward or away from specific visual/optic-flow patterns, resulting in different behavioral actions. Drawing inspiration from J.J. Gibson's perspective, we can formulate the following hypothesis: Optic-flow-driven behaviors must be adaptable because there are diverse patterns that provide information about impending obstacles and the structure of the environment. For instance, a relatively simple scenario involves approaching a wall at a steep angle. If animals were merely stabilizing to optic-flow patterns, they would instantly turn toward the wall. However, the optic flow information should, if the animal intends to avoid a collision, trigger a corrective behavior, such as a saccade in the opposite direction. While we do not yet possess an overarching theory to establish a particular optimization function and define the boundaries of these behaviors, we believe that this marks an essential initial step in dissecting a broader understanding of course control. We have now discussed this in more detail in the discussion section.

- Discussion: The candid discussion of the potential problems using genetic tools is much appreciated. However, for all those issues you should also discuss in which way they may affect the functional interpretation of your results and the conclusions you draw.

To answer this question, we will first compare the state of the art and our methodology. Our study has shown that utilizing the FlpStop-mediated knockout of the *ShakB* gene can be an effective approach to disrupt gap junctions without triggering pleiotropic effects on neural phenotypes. It's important to note that the FlpStop cassette, when inserted in the disrupting orientation between exons 5 and 6, was expected to induce a disruption pattern for *ShakB* similar to that of a nonsense mutation in exon 5 carried by the EMS-mutagenized *shakB2* line, which has been the sole line previously used for effective gap junction disruption. However, we observed substantial phenotypic differences between *shakB2* and FlpStopDshakB mutant flies.

The most prominent disparities include:

I. Dye-coupling in tangential cells: In shakB2 mutant flies, these cells lose all their electrical synaptic partners, while in FlpStopDshakB flies, electrical connections with postsynaptic DNs and motoneurons remain intact.

II. LPTCs in shakB2 mutant flies exhibit ultraslow waves and strong spontaneous membrane hyperpolarizations, phenomena that were absent in FlpStopDshakB flies.

III. HS cells in shakB2 mutant flies display significantly weaker ON/OFF-transient responses and direction-selective responses in comparison to both wild-type and FlpStopDshakB mutant flies.

Thus, important for the functional interpretation is that these observed distinctions show that the behavioral phenotypes are not because of a lack of (i) connection between LPTC and DN, (ii) intrinsic LPTC dynamics observed in shakB mutant, or (iii) strongly reduced direction-selective responses.

Our data, combined with the precise circuit perturbations of HS&VS cells and the newly introduced H2 blocks, allow a dissection of the circuit involvement. With the new set of experiments, it is clear that many perturbations of the HS-H2 network can induce similar behavioural changes. In retrospect, this is not surprising, as animal behaviour is not directly controlled by the LPTC, but as part of a network. Thus, rather than defining the functional role of a particular electrical synaptic interaction, i.e. HSE-H2, we should interpret disruption of gap junction connectivity as a way of modifying network dynamics. We have refined the manuscript accordingly and discussed this directly.

Line 720-731: “Interestingly, in both cases, the linear sum of the behavioural response to unilateral stimuli matches the binocular optomotor response (Figs. 2c, 8d-e), an equivalence that is absent in wild-type and Flp-control flies due to a striking non-linear interaction between two hemispheres (Figs. 1h, 8b,e). The enlargement of the ipsilateral RF in Flp-shakB compared to Flp-control is unlikely to be involved. Our behavioural arena activates the dorsal half of HSE in Flp-control animals. Increasing the ipsilateral RF size, as in Flp-shakB (Fig. 6a-f), should increase the strength of the counter-directional response to unilateral FtB motion. We observed the opposite, suggesting that the contralateral input are necessary for instructing the counter-directional responses. This is supported by our H2 block experiments in which the anti-saccadic response is strongly diminished (Fig. 3). Consequently, our results provide compelling evidence implicating the HS-H2 network (Erginkaya et al. 2023) in the proper selection of smooth and saccadic turns during navigation.”

Comments on the Abstract:

We thank the reviewer for his detailed comments on the commented pdf. We have reformulated the abstract and manuscript for clarity, adding several new citations and pointing out, more specifically, previous contributions.

“Animals rely on visual motion cues to maintain stability and navigate accurately, requiring a range of compensatory actions and sophisticated neuronal computations¹.”

Comment: “ ... sophisticated neuronal computation.” Should be something along the lines “ ... task-specific neuronal processing.”

We have modified the first sentences of the abstract, changing the terminology to “task-specific sensorimotor transformations.”

“Classically, this phenomenon ...”

Comment: “this phenomenon”. Reference not clear. You have included several attributes in your previous sentence.

The sentence has been changed.

Line 9-11: “Animals must perform a variety of compensatory actions in order to maintain stability and navigate through their environment with precision. These actions rely on global visual motion cues known as optic flow. ”

“... has been studied through the optomotor response², a reflexive visual stabilization behavior.”

Comment: “optomotor response” is a term that includes more behaviours than just the “optomotor course control” originally studied by Hassenstein and Reichardt in the 1950s. It would be useful, in general, to think about the function of the reflex you are talking about. In case of the behaviour observed by Hassenstein and Reichardt, the purpose is to stabilize the entire visual image on the eyes in response to an externally induced perturbation of a given trajectory. This is indeed a stabilization reflex that is control in a negative feedback loop (as far as vision is concerned). However, it has been demonstrated by work on LPTCs in *Drosophila* that efference copies (e.g. Kim et al. 2015) can modify the reflex whenever the animal engages on other vision-driven behaviours, e.g. responses to looming stimuli (Fenk et al. 2021) relevant for escape and collision avoidance.

“However, the simplicity of this behavior is insufficient to explain the complexity of natural motion cues experienced during navigation, ... “

Comment: I don’t think that the optomotor course control was ever meant to “ ... explain the complexity of natural motion cues ...” altogether.

We agree. The optomotor response has been a behavior that has been fundamental for the study of motion vision and not intended to study all aspects of vision. There are beautiful examples of looming avoidance and object fixations. In these instances. However, strictly speaking, these behaviors don’t not seem to be mainly controlled by optic-flow processing. We have reformulated the abstract to make it more specific.

Line 11-15: “Traditionally, optic flow compensation in insects has been studied using the optomotor response, a reflexive visual stabilisation behaviour. Despite its robustness, the simplicity of this behaviour has been insufficient for understanding the complex network of optic flow processing neurons and their computations that are thought to be required for visual course control.”

“ ... and the control mechanisms involved remain poorly understood.”

Comment: for this statement to be correct, you should say exactly (i) which specific vision-driven behaviour you are talking about and (ii) what its function is. Your wording comes very close to being offensive for many excellent scientists who, quite successfully, devoted their entire career to study specific visual behaviours quantitatively.

We apologise if our statement has come across as offensive; this was never our intention. Our research has been greatly influenced by the invaluable contributions of many outstanding researchers who have dedicated their lives to the study of visual behaviour. It is because of this work that we have been able to carry out our study.

Despite the considerable efforts made over the past decades, it is clear that the field still needs to fully understand the intricacies of the LPTC network. Our work introduces a fresh perspective that aims to contribute to a more comprehensive understanding of the functional roles of LPTC computations by providing a set of behaviours and a direct link to LPTCs that can be further studied. Given the complexity of this system, it is clear that countless open questions will provide fertile ground for research in the coming decades. We have revised our statement to both acknowledge the outstanding work done over the years and to emphasize that, as a field, we still have a long way to go to grasp the complexity of these interactions.

“Here, by identifying a rich repertoire of course control behaviors in *Drosophila*, ...”

Comment: I didn't see “a rich repertoire of course control” in your study. You didn't apply any stimuli that would allow you to “identify” a rich repertoire. You used an abstraction of a wide-field stimulus in your closed-loop behavioural studies to observe the impact of monocular and binocular motion on the turning behaviour of the fly. The responses were limited to smooth and saccadic behavioural turns syn- and anti-directional with respect to the induced image motion. But you did not mention/discuss the behavioural relevance of the different response components.

We have changed the sentence to:

*Line 15-16: “Here, we define a novel and robust set of course control behaviours in *Drosophila* and link them to the underlying circuit components.”*

“ ... we establish a direct link between opposing corrective actions and the underlying circuit motifs.”

Comment: along the lines of my previous comment – although you could show behavioural changes upon modifications of the neural circuit, i.e. suppressing binocular signal integration, I would not refer to a “rich repertoire”.

We changed “rich repertoire” to “novel and robust set of course control behaviours”

“We determine the algorithmic substrate ... “

Comment: “algorithmic substrate” is very un-lucky wording. How can you determine an algorithmic substrate when you don't provide a clear-cut functional hypothesis?

We respectfully hold differing viewpoints regarding this comment. We introduce a novel set of behaviors that exhibit a comparable level of robustness to the classical optomotor response. While the functional interpretation of the optomotor response centers on stabilization, the counter-saccadic responses serve as a mechanism for course correction, as elaborated upon in the discussion section. Perhaps this may be more a matter of personal preference. We have nevertheless changed the sentence to:

Line 16-18 “We identify a specific set of optic flow-sensitive neurons in the lobula plate of the fly brain and show that bilateral electrical coupling is required to control these behaviours properly.”

“ ... in a specific set of motion sensitive interneurons in the fly brain and demonstrate that visual motion information from both eyes plays a crucial role in movement control through bilateral interactions ... “

Comment: up to this part of the sentence there is nothing new. There are several accounts on binocular integration of visual motion, both in blowflies and in fruitflies – which are not appropriately cited.

“... of electrical signals facilitated by gap junctions.”

Comment: the merits of the current manuscript lie in the studies of the electrical connection between HSN/E and a heterolateral LPTC, the H2-cell. Although the connection itself was already published by Borst and Weber (2011), disrupting it using generic methods (e.g. Flp-ShakB), is a novel approach.

We never claimed that our work was the first one describing heterolateral interactions, since we focused our work and cited several times the work that was the first one specifically addressing the electrical interactions and nonlinear computations between H2 and HS cells (Farrow et al. 2006, not Weber et al., 2011). We have now added several citations on that subject and apologise for not having previously cited this literature and given the wrong impression; this is now corrected.

Important to add is that none of the previous work has established a link between neuronal computations and behaviour via gap-junctions directly. Thus, we think that our sentence is accurate, providing strong evidence for the functional role of previously observed heterolateral LPTC interactions beyond a physiological description.

“These electrical interactions augment the classic stabilization behavior by inverting response direction and behavioral strategy.”

Comment: “behavioural strategy” is a perfect example of an overstatement. It implies purposeful (cognitive) action, where the behavioural alteration was caused by disabling gap junctions required for binocular signal integration (see also commented manuscript).

This might have been a misinterpretation of what we aimed to say with “behavioral strategy”. Our intention was to make a point in changing the actions from smooth turns to saccadic. We have changed this sentence to:

Line 18-20: “The identified electrical interactions work in tandem with chemical synaptic communication within a specific network in the lobula plate neurons, the so-called HS-H2 network, to arbitrate between smooth optomotor and saccadic anti-optomotor turning.”

“Our findings provide insights into how animals integrate monocular motion cues to generate a diverse range of behaviors,”

Comments: “diverse range of behaviour” is overstating the behaviours reported here.

Changed.

Line 20-21: “Our research provides a new understanding of how animals use bilateral motion cues to guide course control,...”

“ ... elucidate the functional significance of the circuit components, and establish a non-canonical role of gap junctions - non-linear operations.”

Comments: yes – gap junctions appear to be important. And yes – the experiments suggest gap junctions to enable the non-linear interaction of binocular motion cues. BTW: based on behavioural work on the landing responses (Borst and Bade 1988) and recordings from the cervical connective (Borst 1991), the non-linear integration of binocular motion signals had been suggested in blowflies before.

Certainly, we agree. In fact, Farrow et al., in their 2006 publication in Nature Neuroscience, have presented direct evidence of nonlinear optic-flow computation occurring between H2 and HSE. We have made reference to their work on multiple occasions. However, there has been a notable absence of studies that directly link this computation to behavior. This gap in research can be attributed to a lack of suitable tools and comprehensive behavioral descriptions. It's worth noting that our work relates the non-linear integration with gap junctions.

“These circuit computations ... “

Comment: I did not find any reference to “computations” in the manuscript, in a sense that the input x requires a “computation” (or formal description) to result in the output y . What the manuscript includes is evidence for a nonlinear transformation based on the difference between the linear sum of the responses to monocular and binocular visual motion and the response to binocular input. Identifying the underlying “computation” would require a more formal description/model that constraints the possible solutions.

“ ... expose neuronal coordination that plays a decisive role in animal behavior beyond the optomotor response, ... “

Comment: this requires further explanation.

“ ... resembling computer vision strategies employed in cutting-edge autonomous artificial navigation^{3,4}.”

Comment: This claim is not related to any aspect of the work reported here. If there was a link – which is difficult to see even for a reader that may be familiar with “cutting-edge autonomous artificial navigation”, it needs further explanation. Otherwise, it would be interpreted as a statement that aims to make the work more relevant for technical applications than it actually is.

Given the space constraint in the abstract, we have simplified and shortened the last sentence to avoid misinterpretations:

Line 21-24: “Our research provides a new understanding of how animals use bilateral motion cues to guide course control, links the HS-H2 network with a novel functional role in course control, and suggests a previously unknown role for gap junctions in non-linear operations.”

Reviewer #3 (Remarks to the Author):

In “Gap junctions coordinate...” Pokusaeva et al investigate the influence of gap junctions on optomotor responses. They use a nice combination of a novel walking closed loop arena, genetics, and well designed visual stimuli to show the importance of bilateral interactions on the HS cells, which are involved in yaw optomotor responses. In particular, they show that gap junctions play a key role in this. It's a nice paper and I mainly have editorial comments.

In the introduction where you talk about the non-uniformity of naturalistic optic flow, you should mention the work of Egelhaaf, especially his papers with Lindemann, as they have thought about this a lot at the theoretical and experimental level.

Line 45, 45-60 LPTCs are often mentioned. The exact number might be ambiguous, but may say at least 30?

We have added a recent new reference from the Reiser lab (Zhao et al. 2023). They have done a beautiful reconstruction of the LPTC, showing the existence of ~60 LPTCS.

Line 47: Kit Longden has shown translation sensitive LPTCs.

We have added the citation together with others; thanks for pointing this out.

Line 55: Several new exciting papers coming out of e.g. the connectome efforts should be mentioned, e.g. <https://www.science.org/doi/10.1126/sciadv.abi7112>

<https://www.biorxiv.org/content/10.1101/2023.08.06.552150v3>

<https://www.biorxiv.org/content/10.1101/2022.12.14.520178v1>

Added and included also some older citations.

Line 78: Please mention the 50 ms delay for closing the loop here

Added.

Line 83 – I was wondering what the benefit of your closed loop system is compared to e.g. the one developed by Andrew Straw?

It seems you're referring to the 2017 paper by Stowers et al. Closed-loop experiments are not groundbreaking innovations, but their specific details can be important, largely depending on the research questions posed. We believe that the advantages of our approach are twofold. First, unlike Stowers et al., we track not only the position but also the orientation of the fly. This allows us to use this for split-screen stimulation. Second, the way we present the stimulus allows for wide-field stimulation above the horizon due to the proximity of the screen to the fly. In addition, our Python-based system provides a level of simplicity that suits our research needs. While it's clear that our system may have a different level of generality than the one developed by Andrew, it has proved effective in addressing locomotion-related questions and in training our students to develop a deep understanding of their work. Another important point to note is that we also developed this system because it gives students the opportunity to learn and develop a deep understanding of their work, from engineering to biology. If the reviewer is looking for an excellent postdoc, I can wholeheartedly recommend him.

Line 88: haven't the Chiappe lab or Maimon lab done work on anti-saccades that should be mentioned here?

Added

Line 101, mention the work by Tammero here, as it is highly relevant, e.g.:

<https://journals.biologists.com/jeb/article/207/1/113/14753/Spatial-organization-of-visuomotor-reflexes-in>

That paper also does a good job of citing the classic literature, and as there is a lot of published work that is relevant to this paper, this should be cited.

We have now added several citations on that subject and apologize for not having previously cited this literature; this is now amended.

Line 134, panel is j not i.

Changed.

The fonts in all the figures are really small (or I am just too old).

The ringing that you show in response to PD motion in figure 4i and j seems to be happening at 60Hz. If so, it could be amplified by the refresh rate of the screen? What did you show in between the stimuli?

It's true that our stimulus is not ideal and there are some flicker fusion artifacts. In particular, we are using an LED DLP projector. While we refresh each frame at 60 Hz, our stimulus is presented with a 120 Hz flicker as the blue and green LED lights are presented sequentially. However, for the points made in this paper, the critical factor is the relative differences between the genotypes. Finally, we decided to retain the stimulus characteristics to maintain consistency with our behavioural experiments. These use a similar refresh rate of 60 Hz (due to technical constraints that prevent increasing the refresh rate in the closed loop system). The moving gratings were interleaved with intervals of stationary gratings (2 s static before motion onset, 2 s motion, 1 s static after).

When you map the receptive fields, could you use the terms LMS and LPD as they are standard in the field? Could you also cite the original papers for the method, which are cited in the Schnell paper you cite.

Added the respective citations and changed the nomenclature.

Line 338: use the word more dorsal, rather than higher

Changed.

Line 349: what is total sensitivity?

Total sensitivity is the sum of all local preferred directions. The contribution of each side to the total sensitivity is defined as the fraction over the total sensitivity. We have clarified it in the text:

Line 413-416 "Both types of HS cells showed sensitivity to local back-to-front motion in the contralateral visual field, comprising 25.3 % of the total sensitivity (the amplitude of the sum of all local motion vectors) of HSN cells and 36.6 % of HSE cells (Fig. 6a,b, Extended Data Fig. 11f,g)."

Figure 5: Your receptive fields are more ventral than published receptive fields. Why is this?

Could this be a confusion? The difference in position between our RF and that of Schnell et al. is mainly due to the field of view stimulated. While our stimulus goes from -10 to 65 degrees in elevation, Schnell et al. characterized the RF using an LED arena that stimulated from -40 to 40 degrees in elevation. The horizon is therefore lower in our case.

I am not convinced about your comparison in figure 5e and f. If I've understood it correctly, you have normalized the LMS to the max in each neuron. You are then subtracting control from Flp-ShakB. This means that weaker overall LMS, i.e. weaker directionality, will appear as a receptive field difference? What is the scale in panels e and f?

This is correct. We have now compared the average, non-normalised responses that we replaced with the normalised versions. These look remarkably similar to the normalised ones. We therefore believe that the RF difference comparison is valid, albeit crude.

Line 383, cite relevant papers.

Done.

I really like your figure 7ciii. Such a cool finding.

Thanks.

The start of the finding makes it sound like you have completely reinvented motion vision research, but as mentioned above, a lot of good work could be cited here. Many papers have 'expanded the classical optomotor paradigm', it is well known that motion vision is non-linear, you do not need really investigate natural motion statistics, and the sentence on line 500 ('Yet these studies..') seem to suggest that all previous work was subpar. Your work is excellent in itself, and you can simply summarize your findings here, without making it sound like you invented everything.

We apologise for giving that impression. This was never our intention, and we don't think that other work is inferior in any way. It may have been our English, or perhaps our excitement came across in the wrong way. We have changed the wording to reflect our intentions and extended the citation list to reference all the work properly.

We have changed the sentence to (Line 598-599): "Yet, these studies could not link the intricate computations experimentally to behaviour."

Line 512, mention all the excellent work done by e.g. Jurgen Haag.

This selection is not easy! I have added two citations that are relevant and one review.

Line 1060-1070: We need information about stimulus time, and interstimulus intervals.

We have added (Line 1206-1207): "The moving gratings were interleaved with intervals of the stationary gratings (2 s static before movement onset, 2 s movement, 1sec static after)."

Line 1085, cite relevant literature. What neural delay was assumed when calculating the receptive fields, the LMS and LPD. This is obviously important. Use the terms LMS and LPD.

The speed of the moving stimulus was slow, so there was no delay to consider. The disadvantage of this slow speed is that the recordings take a long time. We tried several iterations, but these

stimulus parameters gave us the best results. This is similar to the analysis done for the RF reconstructions in (Schnell et al. 2010).

Lines 108-110 and 128-130 repeat each other.

Corrected.

Line 1140, Mention that this is the Michelson contrast definition.

Changed

Red data in Extended data Figure 6j – how can this be such an outlier?

Thank you for noticing this discrepancy. It was a normalisation error. Each parameter of dendritic morphology is normalised to the size of the optic lobe to avoid any variability associated with sample preparation. In the previous version of the analysis, the area was normalised to the squared diameter of the optic lobe. For the updated version, we calculated the true area of the optic lobe and renormalised the dendritic area to this value. We updated Extended Figure 8 according to the new analysis.

We double-checked all data again, below the corrected panel.

Figure legend to Extended data Figure 7 has not been justified. Line 76 mention oscillations that are not seen in the associated panel. The ringing appears to be at 1 Hz?

Corrected (Now extended Fig. 10), it was a reference to panel h.

I did not understand how you got the receptive field maps in Extended data Figure 8d and e.

We apologise for the lack of precision in the description. While the RF in Figure 6 is calculated by taking the response of the moving stimulus per location as a vector sum, in Extended Data Figure 11, we only compare the responses that go in the direction of the PD or ND of the cell. This is now clarified in the legends.

What is the saccade strength in Extended data Figure 10d?

Saccade strength are total turns per saccade. We have changed the label to saccade amplitude. (Now Extended Fig. 12)

Reviewer #4 (Remarks to the Author):

* The paper by Pokusaeva et al. studied the role of gap junctions and well-known visual neurons in one of the most fundamental visual course control behaviors: the optomotor response. This paper is significant for several reasons: (i) the role of gap junctions in this critical model organism is significantly understudied compared to chemical synapses, despite its prevalence. Especially now, as the wiring diagram of the entire brain, based on EM connectomes, is nearly complete, the study of electrical connections becomes timely (refer to a review by Bargmann & Marder 2013 for the importance of gap junction in the post-connectome era). (ii) the binocular integration of optic flow is crucial for mapping vision to action. This study offers an example of dissecting this function from the behavioral to the cellular levels. (iii) The development of a new behavioral assay appears useful for identifying nonlinear binocular integration. The text is written clearly. The experiments employ a range of cutting-edge techniques, including whole-cell patch clamping, new behavioral assays, and the generation of gene knock-down fly lines. The analysis was conducted with an appropriate level of sophistication. While the initial discovery of the nonlinearity in binocular integration and the development of the knock-down agent are praiseworthy, the subsequent exploration and interpretation did not seem entirely coherent. We thus recommend a major revision, which should include additional experiments and analyses.

* Major comments

- In my opinion, the nonlinearity of the optic flow response across both eyes requires additional validation due to the potential drawbacks of the new behavioral assay. First, since the closed-loop control is based on the body axis, the effect of head movement on the visual input is not considered.

This is correct. With the current resolution, we cannot account for head movement in the closed loop. If we were to increase the resolution to accurately account for head movement, we would dramatically reduce the field of view. Therefore, we decided to add a new set of controls instead of head movements, where we glued the head to the chest and performed the split-screen experiment. Although there are some differences compared to the non-head-attached flies, the main results do not change. Flp control flies perform antisaccades to FtB movement, whereas this is reversed in Flp-ShakB flies. See new Extended. Fig. 13 or below for convenience.

Second, the closed-loop implementation would have significant amount of delay, and it may have effects on the behavior of the fly.

We have optimised the closed-loop system as much as possible. The limitations are not in the speed of our calculations, but mainly in the technologies used. The system allows us to control 60 Hz (frame rate). However, due to the limitations of the system, we must first send an image to the projector's buffer and then release the image in the next frame. This gives us a minimum delay of 2 frames, we add the computation and we are between 32-48ms (<48ms). Previous studies that use visual stimulation have faced similar latencies (Stowers et al. 60-75 ms latency, Cruz et al. ~ 40 ms latency). Although not ideal, these latencies have shown to be in the range that allows closed-loop experiments (see supplementary information in Stowers et al. (2017)).

Third, the behavioral assay developed and utilized in the paper operates not only in a closed loop but also in an open loop. It functions in a closed-loop mode because the position and orientation of the pinwheel stimulus are controlled by the fly. Concurrently, it's open-loop as the rotation of the pinwheel isn't controlled by the fly's action.

This is correct and it is done so by design. We want to clamp the "optic flow" perceived by the fly.

Fourth, as noted in the Discussion section, the visual patterns on the roof might have influenced the main findings of the paper. Hence, we request the authors to conduct similar experiments on a tethered walking assay with an air-cushioned ball. We believe that the visual control is more precise in this setup compared to the one used in the paper. If the results from the ball setup align with the existing findings, it would enhance the generality of the main finding of the paper. Any observed differences would likely be attributable to the unique nature of the behavioral assays.

We agree with the reviewer that changing the position of the stimulation will, very likely, drive different responses. Our perspective is based on a preliminary study in our lab that use targeted optogenetics to drive different LPTC in a sparse and cell-specific manner. These experiments show that dependent on the HS cell being stimulated, a diversity of behaviors can be elicited. Thus, changing the stimulation field, as would be done in a tethered setup, would very likely change the response properties. A related experiment was performed recently in a study with hawkmoth. Here different grating patterns were shown at different elevations during flight, which elicited different stabilization maneuvers depending on the elevation (Bigge et al. 2021).

Importantly, however, the aim of our work is not to define the complete repertoire of responses of the fly, but to link a novel and robust behaviour to the underlying circuit components. We have tried in the past to use tethered systems to study such behaviours with little success beyond the classical optomotor response. In short, we set up the system described by Vivek Jayaraman and colleagues. We found that, compared to freely moving systems, the responses of tethered walking flies were drastically more variable and had several caveats that required close scrutiny. Most importantly for our study, we don't see comparable behaviours to freely moving flies. For example, saccades don't occur in the same way as in free walking or tethered flight. This has also been seen by others, e.g. the Jayaramn lab (personal communication). Thus, it will be inconclusive to make a detailed comparison in a tethered system because (i) we expect the responses to change depending on the position of the stimulation in freely walking flies, (ii) saccadic responses are not comparable to freely walking flies and these are a hallmark of our behaviour, and (iii) the responses are extremely variable, allowing only a gross assessment of the responses. A detailed

comparison of the differences in their responses would require establishing a system that has similar stimulation paradigms in a tethered system to be comparable. We believe this comparison is beyond the scope of our work. However, a recent report demonstrates anti-directional responses in tethered flies (Mano et al. 2023) for full-field motion. Although they do not stimulate the dorsal field of view the way we do, their results are consistent with our expectations above. In tethered conditions, Mano et al. do not observe saccadic counterreactions. Furthermore, their tethered system does not reproduce the dynamics observed in our case. Contrary to their results, in our closed-loop paradigm, we do not observe anti-directional responses for full field responses.

- The only pattern used in the paper is pinwheel. It would be helpful if one can observe the same nonlinear integration in response to different optic flow patterns. One good example is the “starfield” pattern used in Weir, Schnell, Dickinson (2014).

We conducted experiments to find a parameter space that could generate a good response for full field rotation. Unexpectedly, when performing the split screen experiments, we observed an inter-fly variability. Half of the flies displayed an anti-saccadic response, as seen when presenting the pinwheel grating stimulus, while the other half did not (Extended Data Fig. 3e-g). This seems similar to our recent experiments using low-contrast stimuli (see Extended Data Fig. 3a-c), where syn-responses are observed instead of anti-saccadic responses. As starfield stimuli typically elicit weaker responses compared to full-contrast gratings, this suggests that the observed behaviour is adaptive and depends on the strength of the optic flow.

We have added the following text to results section:

Line 120-134: “Although responses to FtB and BtF motion could be interpreted as escape behaviours from local motion as shown during tethered flight (Mongeau et al. 2019), the kinetics and relative contribution of smooth and saccadic responses for FtB and BtF motion differ drastically (Fig. 1i, m), altering the relative number of syn- and anti-saccades and their respective amplitudes and velocities (Extended Data Fig. 2). This indicates that FtB and BtF motion drive different modes of action, as evidenced by the difference in straightness of the trajectories (Extended Data Fig. 2c, d). The stimulus dependence of the FtB anti-saccadic response becomes apparent as the stimulus strength is reduced. At low contrast, animals begin to reverse their turning direction for FtB motion, from avoidance to stabilisation (Extended Data Fig. 3a-c). We observed a similar dependence when presenting a different type of optic flow stimulus, the ‘star field’ stimulus (Weir and Dickinson 2015), an overall weaker optomotor inducing stimulus. Consistent with our low contrast results, we observed that half of the flies showed a robust anti-response while the other half did not (Extended Data Fig. 3e-g). These results show that the robustness of the anti-saccadic response depends on the strength of the overall optic flow and demonstrate the existence of stereotyped behavioural adaptations to nuanced changes in stimulus properties.”

- Two critical behavioral experiments in the paper, one with HS>Kir and the other with Flp-ShakB, resulted in the same phenotype: the abolition of nonlinear interactions between the left and right optic flow inputs. However, I believe it's highly improbable that these effects stemmed from modifications in the same neural substrates, let alone the possibility of one being a subset of the other. That is, the HS>Kir manipulation would have rendered HS cells largely incapable of transmitting neural signals, leaving the majority of other neurons unaffected. In contrast, the Flp-

ShakB flies would have influenced a significant portion of innexin8 throughout the brain. Since bilateral interactions via gap junctions might take place at various stages of the visuomotor pathways, including the levels of early visual neurons, VPNs, DNs, and even further down near the motor neurons, the abolition of nonlinear interaction is unlikely to be fully attributed to the gap junctions at the level of VPNs.

We agree with the reviewer that the comparison is correlative. An ideal experiment would have been to block, in a cell-specific manner, gap-junctions. Our study, however, revealed that this is not possible with the tools and driver-lines we have due to the late onset of the Gal4 expression (Fig. 4). However, we believe that our Flp-ShalB approach is still the best strategy currently available and an improvement to the previously used ShakB2 mutant flies.

We politely disagree that the effects of the HS cells will be local. As with the gap-junction mutant, any manipulation performed in a highly interconnected circuitry will also have repercussions. Blocking HS cells, for instance, will influence the circuit by affecting bIPS (Erginkaya et al. 2023), for example. These cells form a bilateral network formed by chemical synapses that influences H2 too. Thus, the only way to make a stronger for these circuit component to be involved is to test whether H2, as predicted for our hypothesis. We performed these experiments either expressing, which are now in the new Fig. 3., and show that H2 is also involved in the computations. Together, the emerging picture is not that the specific electrical synapse between H2 and HSE is relevant, but that the network interaction in the recently described larger HS-H2 network are necessary (Erginkaya et al. 2023). While gap junction connectivity is relevant, so are chemical synapses.

Thus, as recommended by the reviewer, we have changed the empasys across the paper in several instances, for example:

Line 737-763: "What are the underlying circuit motifs and biophysical implementations underlying this non-linear, gap-junction-mediated binocular interaction? Although our research points to the relevance of the interaction between HS and H2 cells, it is important to consider that this computation occurs within a circuit context that necessarily involves additional circuit components and their chemical synaptic interactions. This circuit has recently been anatomically characterized (Erginkaya et al. 2023), involving several important cell-types, e.g. the recently described bIPS and descending neurons controlling saccadic turns (Schnell, Ros, and Dickinson 2017)."

- We would like to suggest a simulation of a model in which HS>Kir led to the abolition of nonlinearity. There are many modeling studies on this neuron (e.g., Bahl et al., 2013 Nat Neurosci). Expansion of those models would likely to recapitulate the HS>Kir result.

We would have liked to provide a comprehensive model of the behaviours, and it is tempting to compare it to those once described by Reichard and Hassenstein and subsequently adapted by others (e.g.,(Bahl et al. 2013)). Although these models have been very good at describing the optomotor and motion responses in lobula plate tangential cells (LPTCs), they are not sufficiently helpful in understanding network interactions. To do this, one needs to model the interactions between LPTCs that arise from interconnected components (e.g., Fig. 8 (Farrow, Haag, and Borst 2006)). In our case, we would need to go one step further, as we want to use such models to define the behaviours instructed by the HS-H2 network. However, we don't think this is possible in our case because we know too little about the properties of the network and how these properties drive different actions. This is illustrated by the inconsistencies we have found with

previous studies. For example, unilateral optogenetic activation of HS mimicking front-to-back (FtB) movements (Haikala et al. 2013; Busch, Borst, and Mauss 2018) elicits stabilising turns, consistent with an optomotor response. However, in our hands, these experiments elicit a strong counter-saccadic response - exactly the opposite of what would be expected from the optogenetic experiments. Why do we see this difference? There are many interpretations, the simplest being that optogenetic activation drives only a subset of LPTCs (e.g. HS cells), while visual stimuli activate HS (and other cells) and inhibit H2. To model such a behaviour we would need an additional understanding of the network dynamics, similar to previous attempts for the LPTC network (Borst and Haag 2002), including important neuronal components recently describe, such as the bIPS (Erginkaya et al. 2023). Therefore, we believe that a simple model as described by Bahl et al. would not be useful and a more realistic model would be beyond the scope of this work, requiring years of dedicated work.

- While it seems less likely, there still exists a possibility that H2-HS electrical connections are important for this behavioral change. Therefore, we request additional experiments with H2>Kir and H2-selective ShalB knockdown.

As mentioned in a previous comment, we would have liked to do cell-specific ablations of ShalB in HS or H2 cells. However, the Gal4 expression started too late, which is a caveat because of the long stability of ShalB. For example, in the DB331-Gal4 line, we don't see any reduction in protein expression or physiological properties experiments (Extended Fig. 10).

Nevertheless, as described in a previous answer, we performed the H2->Kir and H2->shibire[ts] experiments. The specific split Gal4 driver line was unfortunately weak compared to the VS/HS line. Although we couldn't perform the Kir2.1 experiments, which require high expression in dendrites, shibire[ts] showed a clear phenotype comparable to the HS->Kir and ShalB mutant experiments (see new Figure 3). Together, the emerging picture is not that the specific electrical synapse between H2 and HSE is relevant, but that the network interaction in the recently described larger HS-H2 network is necessary (Erginkaya et al. 2023). While gap junction connectivity is relevant, so are chemical synapses. (see below, from new Fig. 3).

* Minor comments

- Figure 1 legend, line 128, consider replacing 'rotation' with 'stimulus' or 'rotation stimulus', to better describe that it is rotation of the pattern, not of the fly.

Changed

- Line#172, consider giving explanation of what 'HSE' stands for as this is the first time it is mentioned (also for HSN and HSE on line 333).

We added a short description to the HS cell nomenclature.

- Line#181, consider using a less strong word than "eliminate". GAL80 does not always fully suppress the GAL4 activity.

Changed

- Figure 4 legend, perhaps more clarification about features of the figures. For example, in 4c the arrows can be more clearly described to better indicate which is FtB and BtF motion. 4h-j consider clearer titles for each plot as some of the symbols used lack some visual clarity.

Modified accordingly

- In Figure 5d, authors suggested that the contralateral visual response of HSE cells is likely due to visual inputs from H2 cells from the other side (Line#349-360). We could not find any direct evidence for this statement (especially Line#358-360). I think that the direct evidence can be shown with H2-selective silencing or gap junction knockdown (as suggested in major comment#2).

The connectivity between HSE and H2 has been previously described in blowflies, where these experiments were carried out using double recordings and sharp electrodes (Farrow, Haag, and Borst 2006). The connectivity between HS and CH cells was shown by (Cuntz, Haag, and Borst 2003). We have added these citations to the text.

Our dye-coupling experiments show that HSE is directly coupled with H2, being H2 the only horizontally sensitive contralateral input. Moreover, H2 coupling is absent in our Flp-ShakB mutants, consistent with (Farrow, Haag, and Borst 2006).

- In Figure 5, one of the most pronounced changes is observed in the ipsilateral dorsal visual field of HSE cells (red circles in Figure 5f). I think that a description of this phenomenon and the candidate mechanisms would be helpful.

It is known that HS cells are connected via gap junctions to CH cells, which are inhibitory neurons that have been described to sharpen the RF of e.g., FD (Cuntz, Haag, and Borst 2003)). It is possible that a similar connectivity pattern is present here. Our dye injections in Figure 7 show a very weak dye-coupled horizontal cell to HSN, presumably dCH. We know from the connectomics data that dCH also synapses to HSE, albeit weakly. Thus, although highly speculative at this point, it is possible that there is cross-inhibition between HS cells that is unmasked in the gap junction mutant.

- Line#355, the abolition of the controlateral responses in HSE cells were not "complete". Significant portion of contralateral responses between 0 and -15 degrees in azimuth remained in Flp-ShakB

Correctly spotted and expected. This is due to the fact that the visual field of the eye extends contralaterally to about 15 degrees. This is a significant observation, which introduces additional complexity into the characterisation of our behavioural experiments. As Reviewer #1 rightly

pointed out, it's important to note that the stimuli are not strictly monocular, but unilateral. We have corrected this discrepancy throughout the text. We have also conducted additional behavioural experiments to firmly establish that the actual interaction is indeed binocular by masking that binocular zone (see Extended Fig. 14 and below for convenience)

- Figure 6a, consider some labeling of these micrographs to better aid in explanation of connectivity. Highlighting ipsilateral HS connections, or downstream connections may aid in understanding.

We have added arrowheads depicting the starter cells, descending neurons, and contralaterally projecting cells.

- Line#490: “the statistics of the visual stimulus” the BtF vs. FtB difference in the contra-lateral visual field does not seem to require probabilistic (or statistical) definition. There may exist some statistical match between the optic flow and visual behaviors, but I do not think that any of the experiments performed in this paper explored this question. Please consider to replace “statistics” with a word that better reflect the contents of the paper.

Correct. We changed the sentence to (Line 587-590): “We show that flies change their locomotion depending on the stimulus properties, i.e. unilateral and bilateral wide-field motion, changing the nature and direction of their turning responses (Fig. 1e,f).”

- Line#1128 of methods, temperature of the arena is mentioned twice. Perhaps this is unnecessary.

Corrected.

- Line#529: close → closed

Corrected.

- Line#589: Shakb → ShakB

Corrected.

References cited:

- Bahl, Armin, Georg Ammer, Tabea Schilling, and Alexander Borst. 2013. "Object Tracking in Motion-Blind Flies." *Nature Neuroscience* 16 (6): 730–38.
- Bigge, Ronja, Maximilian Pfefferle, Keram Pfeiffer, and Anna Stöckl. 2021. "Natural Image Statistics in the Dorsal and Ventral Visual Field Match a Switch in Flight Behaviour of a Hawkmoth." *Current Biology: CB* 31 (6): R280–81.
- Borst, A., and J. Haag. 2002. "Neural Networks in the Cockpit of the Fly." *Journal of Comparative Physiology. A, Neuroethology, Sensory, Neural, and Behavioral Physiology* 188 (6): 419–37.
- Buchner, E. 1971. "Dunkelanregung Des Stationären Flugs Der Fruchtfliege Drosophila." Julius-Maximilians-Universität Würzburg, Germany. https://pure.mpg.de/pubman/faces/ViewItemOverviewPage.jsp?itemId=item_3237821.
- Busch, Christian, Alexander Borst, and Alex S. Mauss. 2018. "Bi-Directional Control of Walking Behavior by Horizontal Optic Flow Sensors." *Current Biology: CB* 28 (24): 4037-4045.e5.
- Cuntz, Hermann, Juergen Haag, Friedrich Forstner, Idan Segev, and Alexander Borst. 2007. "Robust Coding of Flow-Field Parameters by Axo-Axonal Gap Junctions between Fly Visual Interneurons." *Proceedings of the National Academy of Sciences* 104 (24): 10229–33.
- Cuntz, Hermann, Jürgen Haag, and Alexander Borst. 2003. "Neural Image Processing by Dendritic Networks." *Proceedings of the National Academy of Sciences of the United States of America* 100 (19): 11082–85.
- Dahmen, Hans-Jürgen, Matthias O. Franz, and Holger G. Krapp. 2001. "Extracting Egomotion from Optic Flow: Limits of Accuracy and Neural Matched Filters." In *Motion Vision: Computational, Neural, and Ecological Constraints*, edited by Johannes M. Zanker and Jochen Zeil, 143–68. Berlin, Heidelberg: Springer Berlin Heidelberg.
- Elyada, Yishai M., Juergen Haag, and Alexander Borst. 2013. "Dendritic End Inhibition in Large-Field Visual Neurons of the Fly." *The Journal of Neuroscience: The Official Journal of the Society for Neuroscience* 33 (8): 3659–67.
- Erginkaya, Mert, Tomás Cruz, Margarida Brotas, Kathrin Steck, Aljoscha Nern, Filipa Torrão, Nélia Varela, Davi Bock, Michael Reiser, and M. Eugenia Chiappe. 2023. "A Competitive Disinhibitory Network for Robust Optic Flow Processing in Drosophila." *BioRxiv*. <https://doi.org/10.1101/2023.08.06.552150>.
- Farrow, Karl, Juergen Haag, and Alexander Borst. 2006. "Nonlinear, Binocular Interactions Underlying Flow Field Selectivity of a Motion-Sensitive Neuron." *Nature Neuroscience* 9 (10): 1312–20.
- Haag, Juergen, and Alexander Borst. 2004. "Neural Mechanism Underlying Complex Receptive Field Properties of Motion-Sensitive Interneurons." *Nature Neuroscience* 7 (6): 628–34.
- Haikala, Väinö, Maximilian Joesch, Alexander Borst, and Alex S. Mauss. 2013. "Optogenetic Control of Fly Optomotor Responses." *The Journal of Neuroscience: The Official Journal of the Society for Neuroscience* 33 (34): 13927–34.
- Kim, Anmo J., Lisa M. Fenk, Cheng Lyu, and Gaby Maimon. 2017. "Quantitative Predictions Orchestrate Visual Signaling in Drosophila." *Cell* 168 (1–2): 280-294.e12.
- Kiral, Ferdi Ridvan, Suchetana B. Dutta, Gerit Arne Linneweber, Selina Hilgert, Caroline Poppa, Carsten Duch, Max von Kleist, Bassem A. Hassan, and P. Robin Hiesinger. 2021. "Brain Connectivity Inversely Scales with Developmental Temperature in Drosophila." *Cell Reports* 37 (12): 110145.
- Krakauer, John W., Asif A. Ghazanfar, Alex Gomez-Marin, Malcolm A. MacIver, and David Poeppel. 2017. "Neuroscience Needs Behavior: Correcting a Reductionist Bias." *Neuron* 93 (3): 480–90.
- Mano, Omer, Minseung Choi, Ryosuke Tanaka, Matthew S. Creamer, Natalia C. B. Matos,

- Joseph W. Shomar, Bara A. Badwan, Thomas R. Clandinin, and Damon A. Clark. 2023. "Long-Timescale Anti-Directional Rotation in *Drosophila* Optomotor Behavior." *ELife* 12 (September). <https://doi.org/10.7554/eLife.86076>.
- Marr, David. 2010. *Vision*. The MIT Press. London, England: MIT Press.
- Mongeau, Jean-Michel, Karen Y. Cheng, Jacob Aptekar, and Mark A. Frye. 2019. "Visuomotor Strategies for Object Approach and Aversion in *Drosophila Melanogaster*." *The Journal of Experimental Biology* 222 (Pt 3). <https://doi.org/10.1242/jeb.193730>.
- Ros, Ivo G., Jaison J. Omoto, and Michael H. Dickinson. 2024. "Descending Control and Regulation of Spontaneous Flight Turns in *Drosophila*." *Current Biology: CB*, January. <https://doi.org/10.1016/j.cub.2023.12.047>.
- Schnell, Bettina, Ivo G. Ros, and Michael H. Dickinson. 2017. "A Descending Neuron Correlated with the Rapid Steering Maneuvers of Flying *Drosophila*." *Current Biology: CB* 27 (8): 1200–1205.
- Weir, Peter T., and Michael H. Dickinson. 2015. "Functional Divisions for Visual Processing in the Central Brain of Flying *Drosophila*." *Proceedings of the National Academy of Sciences of the United States of America* 112 (40): E5523-32.
- Zhao, Arthur, Aljoscha Nern, Sanna Koskela, Marisa Dreher, Mert Erginkaya, Connor W. Laughland, Henrike Ludwigh, et al. 2023. "A Comprehensive Neuroanatomical Survey of the *Drosophila* Lobula Plate Tangential Neurons with Predictions for Their Optic Flow Sensitivity." *BioRxiv : The Preprint Server for Biology*, October. <https://doi.org/10.1101/2023.10.16.562634>.

REVIEWER COMMENTS

Reviewer #1 (Remarks to the Author):

The authors have done a commendable job addressing extensive comments by the referees. In my view, the manuscript is better for it. The key conceptual findings are (1) a peculiar yet robust optomotor inversion evoked by unilateral or monocular optic flow stimuli during walking, which is not present during flight (see below), and (2) this inversion is dependent on electrical connectivity within optic flow sensing visual projection neurons. The work is complex, comprehensive, and worthy of publication. I have two remaining issues to be addressed, which should be rather straightforward for the authors.

(1) In reading the revised manuscript, I dug into some of the cited papers, and was surprised to find that the optomotor inversion detailed here seemingly occurs only in walking flies, not during tethered flight (reference 62). This point emphasizes the impact of the findings presented here and should be introduced near line 143, and discussed near line 615. I would also appreciate the author's interpretation of the walking-specific optomotor inversion in the context of extra-visual pre-motor excitation shown by HS neurons while walking the dark, without optic flow stimuli (reference 12).

(2) The authors mention in lines 99-103 that by contrast to front-back linear superposition (reference 34), they were surprised to find that left-right superposition breaks down. Yet, reference 62 also shows that during flight, left-right superposition fails, albeit in a manner very different from during walking shown here.

Reviewer #2 (Remarks to the Author):

I have read the rebuttal letter and revised version of the manuscript with great interest and would like to thank the authors for seriously considering and replying to the comments of all for referees and implementing most of the requested changes. The revision has significantly improved the quality of the manuscript, not the least because the results of additional experiments related to some referee queries (new Fig 3.) have been included. With reference to a recently deposited manuscript (Erginkaya et al. 2023) they justify not including a mechanistic model of the HS-H2-cell network I had asked for in my comments – which is ok with me, although I much hope that formalizing a model will be possible at some point soon. In addition to many of the amendments

the authors made, I do appreciate their efforts to remove nearly all overstated claims (but see below) from the manuscript and provide an attempt of a functional interpretation of their observed behaviour.

In my view, the remit of the authors' study is now better motivated and embedded within a framework of earlier work that is well covered by a more inclusive citation practice. There is still some room for improvement, but most of the comments below are easy to address and will not require any major efforts.

If I would ask for anything beyond the points below, it would be a brief discussion on how sensory signals from modalities other than vision may also contribute to unambiguous state (self-motion) estimation. But I will leave this to the discretion of the authors whether they provide such discussion item.

"Gap junctions coordinate binocular course control in flies"

Title: strictly speaking it should be *Drosophila*, not flies. The authors are probably aware of the fact that the number of LPTCs does vary between species. So, experiments on neural mechanisms done in one species do not necessarily apply to others. Also, the differences in flight dynamics within the order of dipteran flies are massive – e.g. between *Calliphora* and *Drosophila* – which may require species-specific adaptations. I strongly recommend replacing “flies” with “*Drosophila*” or “fruitflies”.

L. 10: not everything depends on optic flow. Just change to: “Most of these actions ... “

L. 16: “ We identify ... “ Those cells have been identified (first morphological reconstruction and labelling) a long time ago by Hausen et al. . What you can say is that you worked on some individually identified cells that ... “

L. 21: There is still a problem with the wording in this sentence. It certainly needs qualification regarding what exactly “new understanding” means. “ ... how animals use bilateral motion cues to guide course control ...” has been thoroughly studied and the necessary conditions to do so (bilateral comparison) are well known. What the authors provide here is experimental evidence for

the functional significance of gap junctions in the context of the underlying neuronal mechanism in *Drosophila*.

Also the generalization “... how animals ...” in the abstract is not justified. Your experiments were done in *Drosophila*. Whether gap junctions are as important for similar tasks in primates or animals across phyla, in general, is no more than a bold speculation – the authors present no evidence supporting this statement.

L. 37: this statement “... could be interpreted in multiple ways.” Is only correct if you introduce “local” or “monocular” before optic flow at the beginning of the line. What the authors probably mean is that the correct interpretation of wide-field optic flow over one eye depends on knowledge about the wide-field optic flow on the contralateral eye. Also: “... can be interpreted in multiple ways is a bit too much.” People might be interested in what “multiple” means. “... in different ways ... “ would do.

One way to explain the problem clearly would be the ambiguity between yaw rotation and forward (thrust) translation. During forward translation, optic flow across both eyes would be FtB. When the fly performs a clockwise rotation to the right, optic flow over the left eye would also be FtB, but it would be BtF over the right eye. Only knowing the optic flow over the left eye would provide the animal with a real challenge to distinguish between thrust translation and yaw rotation. But knowing the general motion direction over both the left and right eye would disambiguate between the two cases.

N/B: The authors are probably aware of the fact that, in detail, optic flow during forward translation and during yaw rotation over the left eye is qualitatively the same only along the eye equator. At all other positions they are not the same and only the average motion direction across the eye is horizontal. A very tightly tuned “matched filter” could, in principle, pick up the difference if some assumptions on the visual surroundings could be made and there was no, or little noise in the motion vision system.

So, mathematically, the ambiguity is highest at the local spatial scale.

L. 47: insert “(BtF)” after “... back-to-front ...”

L. 130: should be “... a different optomotor response-inducing stimulus.”

L. 133: “ ... on the strength of the overall optic flow ...” this is not well defined. “Strength of optic flow” would be the magnitude of the of the vector field, which basically depends on the length of the local velocity vectors. You have changed the contrast, not the temporal frequency. So, your local motion stimuli are less effective of driving EMDs, the output of which depends on contrast squared. The correct wording would be “ ... the strength of local motion stimuli ... “

L. 174: the qualification added is important. But pls split the sentence into 2 sentences for clarity.

L. 199: insert ”the” before “equator”

L. 200 ff: this will, I suppose, we picked up in the discussion again. (low expression of Kri2.1 in the H2 cell -> no suppression of function by rectifying potassium channels).

L. 226: “endolytic arrest” this could do with a bit of explanation. The overall targeted effect is blockage of chemical transmission – you mentioned that. But the qualification is too technical for a general audience. Pls modify.

L. 231: replace “thinking that” with “assuming that” or “under the assumption that”

L. 240: drop spare “for” before “H2>shi[ts1]”.

L. 248: last lines of the legend change format in my copy of the pdf—file.

L. 264: “gap junctions” plural, I guess. Again, a long sentence. > 3 lines becomes difficult to follow in English.

L. 273: insert article “the” before “Flp-shakB”

L. 288: as earlier, I would expect this point coming up in the discussion section when you interpret the results of you work.

L. 418: here you could/should add citation [49] to [19], I guess.

L. 429: same comment as above. After all, [49] is addressing binocular optic flow processing, showing detailed (local/global) data on the RF organization in blowflies.

L. 589: if you state “ ... our results show ... “ then the end of the sentence should be modified to “ ... as flies may encounter during natural visual navigation.” Your results were not obtained under “natural visual navigation” conditions.

L. 685: “ ... can interfere ... “ or “ ... could interfere ...” ?

L. 694: shouldn't this be singular “ ... from the BtF sensitive H2 neuron.” There is only one per LP.

L. 706: should be “ ... a rigorous behavioural input-output analysis ... “ instead of “ ... rigorous characterization of the stimulus-behaviour mapping ... “

L. 718: do you mean “ ... , a property ...” or “ ... , an effect ...” ?

Reviewer #3 (Remarks to the Author):

The authors have done an excellent job addressing all my comments.

Reviewer #4 (Remarks to the Author):

Thanks for revising the manuscript and for performing two additional experiments: behavioral experiments with head-fixed control flies as well as with ShibireTS flies. The new version of the manuscript addresses many of our concerns and appears to be substantially improved compared to the original version. In particular, ShibireTS experiments clearly demonstrate that nonlinear bilateral integration can be abolished by silencing chemical neurotransmission in either HS or H2 cells, highlighting the involvement of this pathway in the bilateral integration. Some of our

comments were argued by the authors as non-essential or expected to generate non-interpretable data (e.g., tethered walking experiments and modeling), and we respect the authors' positions on this subject.

One remaining concern relates to one of our previous major comments: how do the two different inactivation experiments applied to two different, non-overlapping neural substrates (chemical synapses in the HS-H2 pathway and electrical synapses in the whole nervous system) lead to the same behavioral phenotype, the abolishment of the nonlinear bilateral integration of optic flow stimuli? This is the most critical question in the paper as the paper basically consists of two parts: 1) the behavioral phenotype upon silencing the HS-H2 network using Kir or ShibireTS, and 2) the development of a new gap junction silencing reagent and behavioral experiments using it. Thus, bridging the gap between these two parts is critical for the paper, in our opinion.

The authors argued in their letter that this may arise from the network dynamics of the extended neural circuits, which include HS, H2, and bIPS neurons. Indeed, answers to many of the reviewers' comments were addressed based on this unidentified mechanism. Here are some examples from the revised manuscript on this subject:

- (To one of reviewer#1's comments) The Chiappe lab recently uncovered several other neurons mediating intricate interactions between HS and H2 cells through GABAergic interneurons, specifically bIPS (Erginkaya et al. 2023). These interneurons are undoubtedly part of the equation. Still, the lack of clear knowledge regarding the circuitry's electrical connectivity strength presents a challenge in defining circuit mechanisms due to too many unknown parameters.

- (Also to one of reviewer#1's comments) This indicates that the close interaction between electrical and chemical connectivity is required to generate these computations in concertation. This is consistent with the HS-H2 chemical connectome analyzed recently by the Chiappe lab (Erginkaya et al. 2023). Thus, determining the role of one cell in this behavior is tricky, since many different perturbations will directly influence the dynamics of this particular HS-H2 network.

- (To one of reviewer#2's comments) We identify that this interaction is mediated by gap junctions and chemical synapses, elucidating a behaviourally relevant interaction for the larger H2-Hs network (Erginkaya et al. 2023).

- (Also in the manuscript) Line 737-763: "What are the underlying circuit motifs and biophysical implementations underlying this non-linear, gap-junction-mediated binocular interaction? Although our research points to the relevance of the interaction between HS and H2 cells, it is important to consider that this computation occurs within a circuit context that necessarily

involves additional circuit components and their chemical synaptic interactions. This circuit has recently been anatomically characterized (Erginkaya et al. 2023), involving several important cell-types, e.g. the recently described bIPS and descending neurons controlling saccadic turns (Schnell, Ros, and Dickinson 2017).

A preprint paper on bIPS (Erginkaya et al., 2023) was mentioned many times by the authors as supporting evidence for this argument. However, a close look into Erginkaya et al. (2023) reveals that their interpretation is fully linear—optic flow inputs from both eyes are added by a weighted sum operation to calculate the difference between the two. Furthermore, their study did not take into account any gap junctions and still was able to successfully explain the visual responses of descending neurons. Thus, we do not see any evidence that bIPS or some other neurons linked to the HS-H2 network are capable of performing the nonlinear bilateral integration, let alone the involvement of gap junctions in this process.

Thus, we ask the authors to provide evidence of how gap junctions in this network could give rise to the nonlinear bilateral integration. In fact, to one of the reviewer#2's comments, the authors stated that “We identify that this interaction is mediated by gap junctions and ...”, which we could not find any evidence for. The authors might consider performing physiological recordings from candidate neurons downstream to the H2-HS circuit in Flp-ShakB flies to show the differences in their visual responses. Alternatively, modeling the whole network based on the model provided by Erginkaya et al. (2023) may also provide some critical evidence about the involvement of gap junctions in the nonlinear integration. Although we very much respect the quality of this study, we think that this is essential because, without bridging this gap between the two parts (Figures 1-3 vs. the rest), the paper, as it stands, appears to be two independent stories that share only their behavioral phenotypes.

Response to reviewers:

We thank the reviewers for their detailed comments and for highlighting issues that require further clarification. We have addressed the remaining concerns by (i) adding modelling work that provides further evidence on how perturbations of different neural substrates on the HS and H2 network, i.e. electrical and chemical synapses, can lead to similar impairments in the discrimination of different bilateral optic flow stimuli and (ii) addressing the remaining points in the discussion in more detail. Our responses are provided on a point-by-point basis below:

Reviewer #1 (Remarks to the Author):

The authors have done a commendable job addressing extensive comments by the referees. In my view, the manuscript is better for it. The key conceptual findings are (1) a peculiar yet robust optomotor inversion evoked by unilateral or monocular optic flow stimuli during walking, which is not present during flight (see below), and (2) this inversion is dependent on electrical connectivity within optic flow sensing visual projection neurons. The work is complex, comprehensive, and worthy of publication. I have two remaining issues to be addressed, which should be rather straightforward for the authors.

(1) In reading the revised manuscript, I dug into some of the cited papers, and was surprised to find that the optomotor inversion detailed here seemingly occurs only in walking flies, not during tethered flight (reference 62). This point emphasizes the impact of the findings presented here and should be introduced near line 143, and discussed near line 615.

We have specified in our result the fact that we are describing a course control mechanism during walking.

Line 145-147: "Furthermore, these results reveal a non-linear binocular interaction that generates the classical optomotor response to full-field rotation and extend the repertoire of course control behaviours in walking Drosophila."

We have also expanded the discussion:

Line 624-629: "Interestingly, a previous study⁶² used similar stimulus paradigms during tethered flight, but did not observe a counter-saccadic response to FtB motion. This suggests that flies may interpret optic flow based on their behavioural state, i.e. either walking or flying. Additionally, this might also reflect adaptations to the location of stimulus presentation. Whereas previous tethered flight paradigms used cylindrical arenas, presenting optic flow stimuli to the frontal visual field, our study presents visual stimuli from above, targeting the dorsal visual field."

I would also appreciate the author's interpretation of the walking-specific optomotor inversion in the context of extra-visual pre-motor excitation shown by HS neurons while walking the dark, without optic flow stimuli (reference 12).

It is interesting to define how extra-visual motion affects the computations we observe. Experimentally, we can't answer this question with our current resolution, as we would need to be able to determine the exact position of each limb during locomotion. However, we could provide two rather vague arguments:

(1) During our visual stimulation, we would argue that the strength of the visual response to our visual stimulation will be much stronger than the extra-visual input, at least at the level of the somata. This could override, during strong visual stimulation, the "stride behavioral context". Alternatively, at the level of the axonal output, this could be different. Stride signals might be stronger relative to the visual signals and therefore, have a stronger influence.

(2) However, the subtleties may be more complex. In our LPTC>Kir2.1 experiments, we silenced all HS and VS cells. This perturbation should silence extra-visual activity too, which is thought to be mediated by gap junctions. However, we don't observe that the straightness of the path is affected during free walking (Extended Data Figure 4c, "no motion"). This could be because multiple LPTC pathways connect to different populations of descending neurons that could compensate for the lack of HS cell activity, enabling rapid steering in the absence of HS cell activity.

A dedicated study would be required to investigate these aspects in detail in freely walking flies. Given the Kir2.1 block experiments mentioned above, the effects may be more subtle than in the tethered walking condition. Although these are all interesting points to think about, we don't feel that our manuscript is the right format to discuss them in detail.

(2) The authors mention in lines 99-103 that by contrast to front-back linear superposition (reference 34), they were surprised to find that left-right superposition breaks down. Yet, reference 62 also shows that during flight, left-right superposition fails, albeit in a manner very different from during walking shown here.

This is correct. Tammero et al. suggested that the hemifield stimulation produces saturated responses. Thus, the sum of the hemifield responses would exceed the responses to half-field motion (dotted red line in 4A in their paper). However, the difference between their results and ours cannot be explained by saturation only, as we would expect an average inversion based on linear superposition. We modify the sentence to be more precise:

Line 101-107: "However, the observed full-field response is drastically different from our linear prediction (Figs. 1g,h), contrary to a previous observation during flight using similar stimuli³⁴. During flight, the behavioural response to a hemifield stimulus appears to produce saturated responses that show a small change in their linear prediction in the direction of motion. During walking, the linear prediction would indicate that the flies would rotate against the classical optomotor response. This suggests that the fly's heading control system integrates visual information from the two halves of the visual field in a non-linear way."

Reviewer #2 (Remarks to the Author):

I have read the rebuttal letter and revised version of the manuscript with great interest and would like to thank the authors for seriously considering and replying to the comments of all for referees and implementing most of the requested changes. The revision has significantly improved the quality of the manuscript, not the least because the results of additional experiments related to some referee queries (new Fig 3.) have been included. With reference to a recently deposited manuscript (Erginkaya et al. 2023) they justify not including a mechanistic model of the HS-H2-cell network I had asked for in my comments – which is ok with me, although I much hope that formalizing a model will be possible at some point soon. In addition to many of the amendments the authors made, I do appreciate their efforts to remove nearly all overstated claims (but see below) from the manuscript and provide an attempt of a functional interpretation of their observed behaviour.

In my view, the merits of the authors' study is now better motivated and embedded within a framework of earlier work that is well covered by a more inclusive citation practice. There is still some room for improvement, but most of the comments below are easy to address and will not require any major efforts.

If I would ask for anything beyond the points below, it would be a brief discussion on how sensory signals from modalities other than vision may also contribute to unambiguous state (self-motion) estimation. But I will leave this to the discretion of the authors whether they provide such discussion item.

"Gap junctions coordinate binocular course control in flies"

Title: strictly speaking it should be *Drosophila*, not flies. The authors are probably aware of the fact that the number of LPTCs does vary between species. So, experiments on neural mechanisms done in one species do not necessarily apply to others. Also, the differences in flight dynamics within the order of dipteran flies are massive – e.g. between *Calliphora* and *Drosophila* – which may require species-specific adaptations. I strongly recommend replacing “flies” with “*Drosophila*” or “fruitflies”.

Although our work has been carried out in *Drosophila melanogaster*, there is clear evidence that such a mechanism may be more general across Diptera. For example, the same link between the same cell types, H2 and HS, has been extensively characterised in blowflies (Farrow et al. 2006). It will always be a generalisation, given the large number of Diptera living on Earth. With our title, we don't want to imply that this will be a general rule for all, but that it goes beyond a single species. We would therefore be grateful if you could leave the title as it is.

L. 10: not everything depends on optic flow. Just change to: “Most of these actions ... “

Changed

L. 16: “ We identify“ Those cells have been identified (first morphological reconstruction and labelling) a long time ago by Hausen et al. . What you can say is that you worked on some individually identified cells that ... “

Changed

L. 21: There is still a problem with the wording in this sentence. It certainly needs qualification regarding what exactly “new understanding” means. “ ... how animals use bilateral motion cues to guide course control ...” has been thoroughly studied and the necessary conditions to do so (bilateral comparison) are well known. What the authors provide here is experimental evidence for the functional significance of gap junctions in the context of the underlying neuronal mechanism in *Drosophila*.

Changed

Also the generalization “ ... how animals ...” in the abstract is not justified. Your experiments were done in *Drosophila*. Whether gap junctions are as important for similar tasks in primates or animals across phyla, in general, is no more than a bold speculation – the authors present no evidence supporting this statement.

Changed to insects

L. 37: this statement “ ... could be interpreted in multiple ways.” Is only correct if you introduce “local” or “monocular” before optic flow at the beginning of the line. What the authors probably mean is that the correct interpretation of wide-field optic flow over one eye depends on knowledge about the wide-field optic flow on the contralateral eye. Also: “ ... can be interpreted in multiple ways is a bit too much.” People might be interested in what “multiple” means. “ ... in different ways ... “ would do.

We have introduced “local” optic flow earlier in the paragraph so, we think the context should be clear. Changed “multiple” to “different”

One way to explain the problem clearly would be the ambiguity between yaw rotation and forward (thrust) translation. During forward translation, optic flow across both eyes would be FtB. When the fly performs a clockwise rotation to the right, optic flow over the left eye would also be FtB, but it would be BtF over the right eye. Only knowing the optic flow over the left eye would provide the animal with a real challenge to distinguish between thrust translation and yaw rotation. But knowing the general motion direction over both the left and right eye would disambiguate between the two case.

We have now added a supplementary figure modelling the discriminability of the network for rotation and translation. Although very simplistic, it makes the same point as suggested by the reviewer. Please see our response to reviewer #4.

N/B: The authors are probably aware of the fact that, in detail, optic flow during forward translation and during yaw rotation over the left eye is qualitatively the same only along the eye equator. At all other positions they are not the same and only the average motion direction across the eye is horizontal. A very tightly tuned “matched filter” could, in principle, pick up the difference if some assumptions on the visual surroundings could be made and there was no, or little noise in the motion vision system.

So, mathematically, the ambiguity is highest at the local spatial scale.

Thank you for pointing that out. That is correct. We would also argue that even without noise, the fact that the contrast and structure of the surround are not uniform in natural environments will also challenge such perfectly tuned “match filters”. Thus, we believe that the complex circuitry observed in the LPTC network, of which we have only shown a small part, is required to make more accurate estimates of the underlying optic flow under more natural conditions.

Depending on the ecological niche and flight behaviour of different insects, the natural statistics will differ. This may also be the reason why we see beautiful morphological adaptations across species in this network. While the connectivity of HS and H2 seems to be conserved, at least between blowflies and fruit flies, this may not be the case for all the other intricate connections. Understanding their function will require a lot of dedicated work, in particular robust behavioural paradigms needed to define their role. We hope that our work will contribute to this effort.

L. 47: insert “(BtF)” after “ ... back-to-front ...”

Changed

L. 130: should be “ ... a different optomotor response-inducing stimulus.”

Clarified

L. 133: “ ... on the strength of the overall optic flow ...” this is not well defined. “Strength of optic flow” would be the magnitude of the of the vector field, which basically depends on the length of the local velocity vectors. You have changed the contrast, not the temporal frequency. So, your local motion stimuli are less effective of driving EMDs, the output of which depends on contrast squared. The correct wording would be “ ... the strength of local motion stimuli ... “

This is correct. We have changed the phrasing.

L. 174: the qualification added is important. But pls split the sentence into 2 sentences for clarity.

Altered to make the sentence clearer.

L. 199: insert “the” before “equator”

Added

L. 200 ff: this will, I suppose, we picked up in the discussion again. (low expression of Kri2.1 in the H2 cell -> no suppression of function by rectifying potassium channels).

Yes, we have discussed this further.

L. 226: “endolytic arrest” this could do with a bit of explanation. The overall targeted effect is blockage of chemical transmission – you mentioned that. But the qualification is too technical for a general audience. Pls modify.

Changed to: “...a dynamin orthologue that blocks chemical synaptic transmission at restrictive temperatures.”

L. 231: replace “thinking that” with “assuming that” or “under the assumption that”

changed

L. 240: drop spare “for” before “H2>shi[ts1]”.

Fixed

L. 248: last lines of the legend change format in my copy of the pdf—file.

Corrected

L. 264: “gap junctions” plural, I guess. Again, a long sentence. > 3 lines becomes difficult to follow in English.

Changed

L. 273: insert article “the” before “Flp-shakB”

Fixed

L. 288: as earlier, I would expect this point coming up in the discussion section when you interpret the results of you work.

We discuss the limitations of the tool that are based on the long stability of ShakB proteins in detail. See discussion section: *Disrupting ShakB*

L. 418: here you could/should add citation [49] to [19], I guess.

Citation added

L. 429: same comment as above. After all, [49] is addressing binocular optic flow processing, showing detailed (local/global) data on the RF organization in blowflies.

Citation added

L. 589: if you state “ ... our results show ... “ then the end of the sentence should be modified to “ ... as flies may encounter during natural visual navigation.” Your results were not obtained under “natural visual navigation” conditions.

This is correct, we have changed the phrasing

L. 685: “ ... can interfere ... “ or “ ... could interfere ... ” ?

Changed

L. 694: shouldn't this be singular " ... from the BtF sensitive H2 neuron." There is only one per LP.

This should be plural, as we are referring to both hemispheres

L. 706: should be " ... a rigorous behavioural input-output analysis ... " instead of " ... rigorous characterization of the stimulus-behaviour mapping ... "

We would prefer to keep the current phrasing

L. 718: do you mean " ... , a property ... " or " ... , an effect ... " ?

Changed

Reviewer #3 (Remarks to the Author):

The authors have done an excellent job addressing all my comments.

Reviewer #4 (Remarks to the Author):

Thanks for revising the manuscript and for performing two additional experiments: behavioral experiments with head-fixed control flies as well as with ShibireTS flies. The new version of the manuscript addresses many of our concerns and appears to be substantially improved compared to the original version. In particular, ShibireTS experiments clearly demonstrate that nonlinear bilateral integration can be abolished by silencing chemical neurotransmission in either HS or H2 cells, highlighting the involvement of this pathway in the bilateral integration. Some of our comments were argued by the authors as non-essential or expected to generate non-interpretable data (e.g., tethered walking experiments and modeling), and we respect the authors' positions on this subject.

One remaining concern relates to one of our previous major comments: how do the two different inactivation experiments applied to two different, non-overlapping neural substrates (chemical synapses in the HS-H2 pathway and electrical synapses in the whole nervous system) lead to the same behavioral phenotype, the abolishment of the nonlinear bilateral integration of optic flow stimuli? This is the most critical question in the paper as the paper basically consists of two parts: 1) the behavioral phenotype upon silencing the HS-H2 network using Kir or ShibireTS, and 2) the development of a new gap junction silencing reagent and behavioral experiments using it. Thus, bridging the gap between these two parts is critical for the paper, in our opinion.

The authors argued in their letter that this may arise from the network dynamics of the extended neural circuits, which include HS, H2, and bIPS neurons. Indeed, answers to many of the reviewers' comments were addressed based on this unidentified mechanism. Here are some examples from the revised manuscript on this subject:

- (To one of reviewer#1's comments) The Chiappe lab recently uncovered several other neurons mediating intricate interactions between HS and H2 cells through GABAergic interneurons, specifically bIPS (Erginkaya et al. 2023). These interneurons are undoubtedly part of the equation. Still, the lack of clear knowledge regarding the circuitry's electrical connectivity strength presents a challenge in defining circuit mechanisms due to too many unknown parameters.

- (Also to one of reviewer#1's comments) This indicates that the close interaction between electrical and chemical connectivity is required to generate these computations in concertation. This is consistent with

the HS-H2 chemical connectome analyzed recently by the Chiappe lab (Erginkaya et al. 2023). Thus, determining the role of one cell in this behavior is tricky, since many different perturbations will directly influence the dynamics of this particular HS-H2 network.

- (To one of reviewer#2's comments) We identify that this interaction is mediated by gap junctions and chemical synapses, elucidating a behaviourally relevant interaction for the larger H2-Hs network (Erginkaya et al. 2023).

- (Also in the manuscript) Line 737-763: "What are the underlying circuit motifs and biophysical implementations underlying this non-linear, gap-junction-mediated binocular interaction? Although our research points to the relevance of the interaction between HS and H2 cells, it is important to consider that this computation occurs within a circuit context that necessarily involves additional circuit components and their chemical synaptic interactions. This circuit has recently been anatomically characterized (Erginkaya et al. 2023), involving several important cell-types, e.g. the recently described bIPS and descending neurons controlling saccadic turns (Schnell, Ros, and Dickinson 2017).

A preprint paper on bIPS (Erginkaya et al., 2023) was mentioned many times by the authors as supporting evidence for this argument. However, a close look into Erginkaya et al. (2023) reveals that their interpretation is fully linear—optic flow inputs from both eyes are added by a weighted sum operation to calculate the difference between the two. Furthermore, their study did not take into account any gap junctions and still was able to successfully explain the visual responses of descending neurons. Thus, we do not see any evidence that bIPS or some other neurons linked to the HS-H2 network are capable of performing the nonlinear bilateral integration, let alone the involvement of gap junctions in this process.

Thus, we ask the authors to provide evidence of how gap junctions in this network could give rise to the nonlinear bilateral integration. In fact, to one of the reviewer#2's comments, the authors stated that "We identify that this interaction is mediated by gap junctions and ...", which we could not find any evidence for. The authors might consider performing physiological recordings from candidate neurons downstream to the H2-HS circuit in Flp-ShakB flies to show the differences in their visual responses. Alternatively, modeling the whole network based on the model provided by Erginkaya et al. (2023) may also provide some critical evidence about the involvement of gap junctions in the nonlinear integration. Although we very much respect the quality of this study, we think that this is essential because, without bridging this gap between the two parts (Figures 1-3 vs. the rest), the paper, as it stands, appears to be two independent stories that share only their behavioral phenotypes.

We thank the reviewer for raising the concern about how different circuit perturbations lead to the same behavioural deficits. This is clearly a point that needs clarification. Importantly, we agree with the reviewer's point that bIPS as modelled by Erginkaya et al 2023 cannot account for the behaviour we describe. We believe that this may have been a misunderstanding of the wording of our response. In our previous response to the reviewers, we attempted to demonstrate the complexity of the circuit by citing recent work (Erginkaya et al., 2023). The data presented by Erginkaya et al. show the complexity of the chemical interaction between the H2-HS circuit and some of its connected neurons. We didn't intend to provide a circuit mechanism that could explain the behaviour we see.

Importantly, we believe that the different perturbations, blocking chemical synaptic output from HS and H2 cells, inhibiting HS via Kir, and removing specific gap junctional connectivity (between LPTCs and not from LPTCs to descending neurons), are closely related. We would therefore like to clarify our views by first providing more context and secondly, supporting modelling work. To do this, we would first need to put our results into context.

What we have shown (as a quick reminder):

(i) We established a new behavioral paradigm.

(ii) We identified a novel, robust and reproducible behavioral response that extends the classical optomotor stabilization behavior by changing not only the course of action but also the choice of action (smooth to saccadic).

(iii) We identified, for the first time, a course control behavior that requires specific LPTCs by blocking components (HS and H2) of cells known from previous work to interact in a nonlinear manner through a gap junction-mediated mechanism (Farrow et al. 2006).

(iv) We validated and generated a novel gap junction disruption tool (Flp-ShakB) to test whether gap junctions, as previously modeled and reported in blowflies, are required.

(v) Although we were unable to block this interaction in a cell-specific manner, we performed a detailed study of possible reasons. This is a valuable contribution to future and necessary attempts to modify the electrical synaptic interaction in a cell-specific manner.

(vi) We show that the connectivity between HS and H2 is disrupted in our pan-neuronal mutant. We also show that while visual response properties of HS neurons and their electrical connectivity to descending neurons remain largely intact, there is a clear and robust behavioral phenotype, reinforcing the results that cross-hemispheric interactions are important. Furthermore, we show that the receptive field and dye coupling are disrupted specifically due to the disruption of gap junctions. This concerns our reply to reviewer #2, mentioned by reviewer #4: *"We identify that this interaction is mediated by gap junctions and..."*; which is also associated with our next point.

(vii) We show that the counter-saccadic bilateral optomotor behaviour is impaired in our gap junction mutants. Furthermore, our new modelling results, a new addition to our revised manuscript, suggest that this may result in an inability to discriminate between rotation and translation.

What we did not show (as also pointed out by the reviewer):

(i) We did not show that the specific changes in electrical connectivity between HS and H2 that generate the nonlinear interaction, as shown experimentally and in a model by Farrow et al. 2006, are necessary for the nonlinear behavioral response.

(ii) We don't show a circuit mechanism that is required to generate the behavior. We only define circuit components that are required in three different ways.

For the purposes of the interpretation of our work, we would like to refer to an existing model that shows that gap junctions, specifically in the HS-H2 network, are necessary to disambiguate between rotation and translation (Farrow et al. 2006, Fig. 8).

Taken from Farrow et al: Simulations of experiments showing the difference in H2 cell spiking for translational and rotational optic flow.

Of course, this model does not relate circuit computation to behaviour, but only shows that components of this circuit can enhance rotation/translation discrimination. We would like to emphasise that, to the best of our knowledge, there is no work that relates any of these LPTC network computations to actual optomotor deficits. This is because experimental evidence has only linked the block of direction selectivity at the level of T4/T5 to deficits in optomotor responses. The only exception are the deficits in head movements resulting from the block of HS cells (Kim et al. 2017). This is also the case for the model proposed by Erginkaya et al. They recapitulate the functional findings based on calcium imaging results, not behaviour. Furthermore, they relate these findings to path straightness during walking and not to optomotor-like turns. These changes in straightness could be due to changes in head movement, as the focus is on the descending neuron DNp15. Published connectivity downstream of DNp15 suggests that its likely function is to control neck muscles rather than locomotion (see below).

(Taken from Namiki et al. 2018, Fig4 and Fig15. DNp15 highlighted)

We have no reason to believe that the output of this model, DNp15, controls locomotion and thus, is directly related to the behaviours described in this work. Several other descending neurons connected to HS and H2 cells are not considered. Some are specific for H2, others for HS. We have analysed the connectivity from the FlyWire connectome⁷¹ and provide an overview below (now also in the supplementary material):

Number of synaptic inputs to Descending Neurons (DNs) from HSN, HSE, HSS and H2 neurons. Only DNs where the number of synapses was greater than 5 were considered.

Furthermore, descending neurons that do not receive direct input from HS or H2 neurons may still be indirectly affected via intermediate neurons such as bIPS and uLPTCrn. Thus, there are too many unknowns to build a mechanistic input-output model of behaviour and its relationship to gap junctions. This is clearly valuable but beyond the scope of this paper. We would first need to determine the role of each of these descending neurons in the context of our anti-saccadic response, determine their presynaptic connectivity beyond the bIPS, and establish the underlying circuit mechanisms. Our work provides a clear framework for such follow-up studies by providing a novel and fine-grained behaviour.

Based on the reviewer's suggestions, we recapitulated the model of Erginkaya et al. 2023 and added a specific gap junctional connection between HS and H2. This led us to an interesting observation that highlights the role of gap junctions in the context of translation/rotation discrimination. We show that the gap junction between HS and the contralateral H2 enhances rotation/translation discrimination throughout the network, not just in descending neurons. This information can then be used by several different downstream elements, not just bIPS or DNp15, to accurately estimate optic flow.

In short, we have reproduced this model and added a term for the gap junction connection between H2 and HS, the strength of which can be varied by changing the weight. Thus, we assume that HS and H2 neurons receive all or most of the information about contralateral motion via this gap junction. It should be noted that although the original model includes terms that allow H2 and HS to receive contralateral motion information (we have set the weights of these terms to zero), the terms we have added better represent the bidirectional nature of a gap junction. First, we considered the effect of gap junctions on the ability of the neuron to discriminate between translation and rotation, measured here by the Discrimination Index (DI), where

$$DI = (Optic_Flow_translation - Optic_Flow_rotation) / (|Optic_Flow_translation| + \dots |Optic_Flow_rotation|);$$

$$Optic_Flow_rotation = \max(\text{clockwise}, \text{counterclockwise}) - \min(\text{clockwise}, \text{counterclockwise});$$

$$Optic_Flow_translation = \max(\text{forward_translation}, \text{backward_translation}) - \dots \min(\text{forward_translation}, \text{backward_translation});$$

A value of -1 indicates that the neuron is more selective for rotational optic flow, while a value of +1 indicates that the neuron is more selective for translational optic flow.

We find that increasing the strength of gap junctions increases the discrimination index of all neurons in the network, albeit to varying degrees. This is in line with the findings of Farrow et al who show that H2 neurons respond more strongly to translation than to rotation and that this differential response is dependent on electrical synapses between HSE and H2.

Right: Modified schematic of the HS-H2 network from Chiappe et al. Left: Discrimination index (DI) between translational and rotational stimulus for each neuron type in the HS-H2 network on the left. A value of +1 indicates that the neuron responds differentially to forward and backward translation, while a value of -1 indicates that the neuron responds differentially to clockwise and counterclockwise rotation. Note the change of DI for stronger gap-junctional coupling.

Furthermore, this model allowed us to investigate how the outputs of HS and H2 neurons affect rotation/translation discriminability. Indeed, all these perturbations alter the ability of the network to discriminate between local and global optic flow. Although we don't know how this information is relayed to the ventral nerve cord via the many possible descending pathways to control behaviour, it shows how different biological substrates, i.e. electrical and chemical synapses, can alter network dynamics that could lead to similar behavioural phenotypes.

Comparison of Discrimination Index for different configurations of the circuit for all neurons. Weights corresponding to specific connections in the model were changed to get the different configurations. Schematic at the bottom shows the state of the circuit after changes were made to the weights.

Finally, we were able to show that flies can reverse their direction under extreme conditions. This may suggest that other circuit interactions could take advantage of the existing circuit components by increasing the activity of certain interneurons through non-linear interactions, i.e. spiking mechanisms.

A. Angular speed generated by the model, given by $V_{angular} = W_{visual} * (DNp15_{right} - DNp15_{left})$, for different values of the strength of the gap junction between HSE and H2 neurons. B. Same as in A, for different values of the weight of the uLPTCrn- to -DNp15 synapse and HS - to -DNp15 synapse C. Same as in B for different strengths of gap junction

We haven't added this to our manuscript, since we don't believe that DNp15 is involved in the behaviour we describe and thus, might rather be misleading than helpful at the current stage.

Finally, we have added these results to a new supplementary figure and expanded our discussion to clarify the points raised above:

Line 538-547: “Notably, in *Flp-shakB* flies subjected to front-to-back (FtB) motion, smooth turning in the direction of the stimulus was enhanced, while saccadic counter turns were diminished (Fig. 8g-i, Extended Data Fig. 12b), suggesting that flies cannot discriminate between local and global changes in optic flow. Inspired by a recent study describing a circuit mechanism that enhances the discriminability of bilateral interactions¹⁵, i.e. translation and rotation, we reproduced the model to ask whether bilateral gap junctions between HS and H2 cells affect their selectivity to translational or rotational stimuli (Extended Data Figure 15a). Our modelling results show that gap junctions increase the selectivity of neurons to rotational optic flow in all cells of the circuit, similar to specific perturbations of their chemical output (Figure 15b, c).”

Line 744-764: “Further evidence for such network interactions can be taken from our modelling results, which show that disrupting chemical or electrical connectivity in a biologically inspired HS-H2 network reduces the discriminability of bilateral interactions across the circuit (Extended Data Fig. 15a,b). These findings underscore the functional significance of gap junctions within the insect nervous system, connections that have been predominantly overlooked in connectomic analysis due to the challenges associated with their visualisation. Moreover, they suggest that gap junctions can facilitate non-linear operations that play a decisive role in animal course control rather than merely averaging neighbouring signals, as suggested through careful experiments and of the electrical HS-H2 interaction in blowflies¹⁹. However, a significant question remains unanswered. What are the underlying circuit motifs and biophysical implementations underlying this non-linear, gap-junction-mediated binocular interaction that give rise to behaviour? Although our research points to the relevance of the interaction between HS and H2 cells, it is important to consider that this computation occurs within a circuit context that necessarily involves additional circuit components and their chemical synaptic interactions. The circuit was characterised in a recent study that showed that inhibitory inputs from an interhemispheric interneuron, bIPS, to a descending

neuron, DNp15, are required to maintain path straightness¹⁵. This HS-H2 subnetwork may not be involved in the counter-saccadic behaviour described in our work, as it innervates the neck and the halteres⁷⁰. It is more likely that other descending pathways connected to HS or H2 cells (Extended Data Fig. 15d)⁷¹ are required, such as those involved in saccadic turns⁷².”

1. Duistermars, B. J., Care, R. A. & Frye, M. A. Binocular interactions underlying the classic optomotor responses of flying flies. *Front. Behav. Neurosci.* **6**, 6 (2012).
2. Tammero, L. F., Frye, M. A. & Dickinson, M. H. Spatial organization of visuomotor reflexes in *Drosophila*. *J. Exp. Biol.* **207**, 113–122 (2004).
3. Erginkaya, M. *et al.* A competitive disinhibitory network for robust optic flow processing in *Drosophila*. *bioRxiv* 2023.08.06.552150 (2023) doi:10.1101/2023.08.06.552150.
4. Farrow, K., Haag, J. & Borst, A. Nonlinear, binocular interactions underlying flow field selectivity of a motion-sensitive neuron. *Nat. Neurosci.* **9**, 1312–1320 (2006).
5. Namiki, S., Dickinson, M. H., Wong, A. M., Korff, W. & Card, G. M. The functional organization of descending sensory-motor pathways in *Drosophila*. *Elife* **7**, (2018).
6. Schnell, B., Ros, I. G. & Dickinson, M. H. A Descending Neuron Correlated with the Rapid Steering Maneuvers of Flying *Drosophila*. *Curr. Biol.* **27**, 1200–1205 (2017).

REVIEWER COMMENTS

Reviewer #1 (Remarks to the Author):

I applaud the extensive effort that the authors have put into revising their work. There are a number of valuable results presented here including a novel binocular optomotor behavior, a first-of-its-kind cell-specific perturbation of optomotor control by silencing lobula plate tangential neurons, and that saccadic behavior can be perturbed by gap junction mutants. There is also a deeply penetrating criticism that without cell-specific manipulation of gap junctions, it is impossible to understand how gap junctions function in "coordinating binocular course control", as suggested by the title. It remains possible that the influence of gap junctions on the electrical tone of a network of interconnected neurons might never reveal itself with single-cell manipulations. This limitation seems to be uncorrectable.

Reviewer #2 (Remarks to the Author):

In their latest revision the authors have dealt with many of the referee comments in a reasonable way and managed to implement further improvements. This includes a simple model based on a previous study (Erginkaya et al. 2023) which provides supporting evidence for the significance of gap-junction-based heterolateral connections to improve the LPTC selectivity index that quantifies the disambiguation between translational and rotational optic flow. The model is now described in the Material & Methods section – although without showing the network structures tested and the results, both of which were included in the last rebuttal letter.

Major issues, e.g. the discrepancy between behavioural results obtained in walking as apposed to flying animals, are addressed/discussed.

If I am missing anything, it is still a more detailed functional interpretation of the anti-directional saccades (see also comment at the end of this report). Why/under which conditions would flies need to perform counter-directional saccades? The interpretation of the classic (syn-directional) optomotor response straight forward. Often referred to as “course control” task, it actually stabilizes the visual input against rotational state changes. But what about anti-directional saccades? The tentative explanation in the manuscript is now given in the context of an evasive behaviour. This is at least a reasonable hypothesis. It would be triggered by local inconsistencies in the global optic flow field and would be a behaviour that needs to be integrated with flight stability control – which is likely to involve efference copies (see comment below). The important point is that a hypothesis on the functional relevance of a behaviour (i) can be tested experimentally and (ii)

provides some ideas about the computational operations required to achieve the behavioural task which guides the search for their neural implementation.

The question of how exhaustive the authors dealt with comments/queries brought up by the other referees, I leave to the other referees' assessment. Below I have some further comments on the latest version of the manuscript and on a few statements the authors included in their rebuttal letter.

Comments on the latest revision:

Staying with the generalization “flies” in the title and not specifying the studied species: Even though I would personally prefer a title that explicitly mentions the species, staying with “flies” rather than changing to “Drosophila” would be ok/acceptable.

L. 86: “These anti-saccades, but their origin and function remain unclear.”

Also: in this sentence you include a tentative interpretation of “responses” as escape behaviour, but only mention the stimulus direction (BtF and FtB) without explicitly saying whether you refer to syn-directional or anti-directional saccades, or to smooth (syn-directional) optomotor turns.

L. 123: sentence too long. Split into 2 sentences

L. 131: “ ... reverse their turning direction for FtB from avoidance to stabilization. ...”. So, there is a functional interpretation of the anti-directional saccadic response?

L. 135: “ ... show that ...” should be “ ... suggest that ...”

L. 138: insert “ ... of view ...” after “ ... binocular ...”

L. 144/5: what is again the difference between “monocular” and “unilateral” stimuli?

L. 228: “ ... required for adequate course control.” “adequate” implies you know what course control should look like. Either you specify exactly what “course control” behaviour you are referring to or you drop “adequate” and stay at the general level.

L. 506 “ ... mediate binocular behavioural instructions.” This is a case of uncommon/unlucky wording. I don’t think that gap junctions “instruct” any behaviour – they allow current to pass. An instruction implies a (conscious) purposeful action. What gap junctions do is they contribute to behavioural modulations. Pls change the wording.

L. 534: “ ... are directly responsible ...” sounds like necessary and sufficient. I recommend wording more cautiously. “ ... are involved in ...”

L. 541: “ ... discriminate between local and global optic flow.” is unclear. This should be related to the ambiguity of horizontal motion due to either translation or rotation, which requires the availability and integration of ipsilateral and contralateral motion signals.

L. 552: “ ... summing binocular cues.” Would be great if you did add the functional purpose here, too, as you did for the “monocular information”

L595: “Taken together ...” Good to have a summarizing sentence. But it would be even better to be more specific on the functional context when stating “ ... finely tuned to specific properties of global motion ...”

L. 613: induce blank “ ... beelicited ...” -

Same sentence: this is a good example. It should be mentioned though, that optomotor course control does not appear to involve the use of efference copies while course changes do (Fenk et al. 2021).

L. 626: “ ... behavioural state ...” should be “ ... locomotor state ...”

L. 634: “ ... the panoramic position of optic flow ... “ Optic flow is always panoramic and covers the entire visual field. What you probably mean is which section of an optic flow field is chosen as a visual stimulus and onto which area of the (compound) eye it is projected. Of course, optic flow can locally be modified due to independently moving/approaching objects/animals within the visual field. Those local optic flow inconsistencies would potentially trigger other visually controlled behaviours, different from course or attitude control.

L. 712: replace “ ... instruct ...” with “... perform ...”

L. 713: “ ... optic flow is global or local.” Pls see my earlier comment. Optic flow is by definition always global, as it is related to self-motion. But as I mentioned above, any retinal image shifts resulting from external object motion would locally produce inconsistencies. This is simply the case because local retinal image shifts (irrespective of its origin) will be linearly combined, and the resulting local motion vector is the input to the motion vision pathway.

L. 719: sentence incomplete – what “computation”? I am still not overly happy with “ ... stimulus-behaviour mapping ...” as it sounds both clumsy and vague.

L. 740: replace “instructing”

L. 745: should be “ ... experimentally informed ...” rather than “ ... biologically inspired ...”

L. 751/2: “... rather than merely averaging neighbouring signals, ...” I know the work by Farrow et al. , but reader who don't would be confused by this statement. Also, the working belittles the study. The coupling of neighbouring LPTCs was interpreted in more functional detail.

L. 755/6: ... that give rise to behaviour.” should be “ ... that modulate the behavioural output.”

L. 757: “ ... this computation ...” what computation? Reference not clear.

L. 759ff: The newly added text is more confusing than explanatory. It would require a bit more explanation that is linked to the current study and motivates why it is included. The last sentence, where you proudly mention the “... expansion of the known repertoire...” [of optomotor responses] finishes “... a comprehensive understanding of the neuronal mechanisms of vision in flies” This sounds vague and it conveys the impression that we don't know much about vision in flies – which is, I think, not quite correct.

It is also worth noting that just observing a “new behaviour” and quantifying is only one first step. Anything that is related to “a comprehensive understanding” also requires a clear understanding of the natural behavioural context including a functional interpretation. Knowing what behavioural

state needs to be controlled immensely simplifies the task of modelling it, which in turn enables us to identify the “computation” in the nervous system required to achieve it.

Getting back to the final sentence: In my view, it also dilutes the merits of your finding that gap junctions play an important role in the integration of binocular motion information.

Additional comment:

In your rebuttal letter, after explaining the value of the added model, you wrote:

“This information can then be used by several different downstream elements, not just bIPS or DNp15, to accurately estimate optic flow.”

This is not directly related to your current manuscript, but more of a general comment: To understand the design of visuo-motor control, it is important to keep in mind the different levels required. Generally speaking, these levels include sensing, actuation (motor systems) and the control architecture connecting the former to the latter. A major part of this control architecture are the descending pathways. Their function is no longer the encoding of sensory information (optic flow estimation) but rather the integration of any sensory information that results in the generation of an appropriate control command (driving the motor neurons/systems), ideally within a coordinate system that is aligned with the coordinate system of the actuators. Most DNs connect to motor neurons which, in turn, connect to specific muscles involved in the control of the behavioural output and are likely to do this in the respective motor system coordinates. So, what needs to be estimated is the state change (self-motion) which is based on processing optic flow at the level of LPTCs in the 1st instance (including ipsilateral and heterolateral connections) and is further refined by integrating binocular signals. The output of the integration already provides a control signal in motor coordinates, rather than encoding/estimating optic flow.

Reviewer #4 (Remarks to the Author):

I appreciate the authors' response to my review comments and the changes made to the manuscript. In particular, the new simulation data (Extended Data Figure 15) using the HS-H2 model addresses a large part of our concerns by providing a theoretical basis for the role of the HS-H2 network in binocular optic flow processing. I would like to mention that our previous comment regarding the circuit element responsible for nonlinear binocular integration has not been addressed. Despite this, I think that all the other comments have been directly addressed. With the modeling work in addition to the extensive amount of all the other experiments, I believe the authors have done a commendable job dissecting the role of commissural visual neurons (H1) and

gap junctions in the binocular vision of *Drosophila*. Thus, we recommend the paper for publication as it is.

I have one last request to the authors and leave it to their discretion whether to reflect it in the manuscript. As you mentioned in the rebuttal letter, the paper does not provide “the specific changes in electrical connectivity between HS and H2 that generate the nonlinear interaction.” Thus, I believe that the new sentence in the abstract is not the most accurate statement for the findings of the paper. Therefore, I recommend reverting the following sentence in the abstract to the previous version.

(Line 16-18, new version): We show that bilateral electrical coupling between a specific set of optic flow-sensitive neurons in the lobula plate of the fly brain is required to control these behaviors.

(Line 16-18, previous version): We identify a specific set of optic flow-sensitive neurons in the lobula plate of the fly brain and show that bilateral electrical coupling is required to control these behaviors properly.

We would like to thank all reviewers for their constant and constructive comments. They have significantly improved our manuscript. In this last version, we have changed all remaining and requested changes to the text. Please find the details below.

Yours sincerely,

Victoria Pokusaeva, Roshan Satopathy, Olga Symonova and Maximilian Jösch

Reviewer #1 (Remarks to the Author):

I applaud the extensive effort that the authors have put into revising their work. There are a number of valuable results presented here including a novel binocular optomotor behavior, a first-of-its-kind cell-specific perturbation of optomotor control by silencing lobula plate tangential neurons, and that saccadic behavior can be perturbed by gap junction mutants. There is also a deeply penetrating criticism that without cell-specific manipulation of gap junctions, it is impossible to understand how gap junctions function in "coordinating binocular course control", as suggested by the title. It remains possible that the influence of gap junctions on the electrical tone of a network of interconnected neurons might never reveal itself with single-cell manipulations. This limitation seems to be uncorrectable.

We agree that the title should be slightly modified. We suggest the following title to reflect better the overall findings, including our work on gap junctions and chemical interactions in the HS-H2 network:

"Bilateral interactions of optic-flow sensitive neurons coordinate course control in flies."

Reviewer #2 (Remarks to the Author):

In their latest revision the authors have dealt with many of the referee comments in a reasonable way and managed to implement further improvements. This includes a simple model based on a previous study (Erginkaya et al. 2023) which provides supporting evidence for the significance of gap-junction-based heterolateral connections to improve the LPTC selectivity index that quantifies the disambiguation between translational and rotational optic flow. The model is now described in the Material & Methods section – although without showing the network structures tested and the results, both of which were included in the last rebuttal letter.

Major issues, e.g. the discrepancy between behavioural results obtained in walking as apposed to flying animals, are addressed/discussed.

If I am missing anything, it is still a more detailed functional interpretation of the anti-directional saccades (see also comment at the end of this report). Why/under which conditions would flies

need to perform counter-directional saccades? The interpretation of the classic (syn-directional) optomotor response straight forward. Often referred to as “course control” task, it actually stabilizes the visual input against rotational state changes. But what about anti-directional saccades? The tentative explanation in the manuscript is now given in the context of an evasive behaviour. This is at least a reasonable hypothesis. It would be triggered by local inconsistencies in the global optic flow field and would be a behaviour that needs to be integrated with flight stability control – which is likely to involve efference copies (see comment below). The important point is that a hypothesis on the functional relevance of a behaviour (i) can be tested experimentally and (ii) provides some ideas about the computational operations required to achieve the behavioural task which guides the search for their neural implementation.

The question of how exhaustive the authors dealt with comments/queries brought up by the other referees, I leave to the other referees’ assessment. Below I have some further comments on the latest version of the manuscript and on a few statements the authors included in their rebuttal letter.

Comments on the latest revision:

Staying with the generalization “flies” in the title and not specifying the studied species: Even though I would personally prefer a title that explicitly mentions the species, staying with “flies” rather than changing to “Drosophila” would be ok/acceptable.

L. 86: “These anti-saccades, but their origin and function remain unclear.”

Also: in this sentence you include a tentative interpretation of “responses” as escape behaviour, but only mention the stimulus direction (BtF and FtB) without explicitly saying whether you refer to syn-directional or anti-directional saccades, or to smooth (syn-directional) optomotor turns.

We assume the reviewer is referring to L123, where we presented one of the interpretations of our behavioral data in response to unilateral motion, as suggested by reviewer #1. This interpretation is that the fly's turning can be framed in terms of the position of the unilateral motion rather than its direction, in which case, the fly turns away from unilateral motion irrespective of its direction. We have changed the sentence to clarify this:

Line 123: *“Although responses to FtB and BtF motion could be interpreted as escaping or turning away from the location of unilateral motion irrespective of its direction as shown during tethered flight, the kinetics and relative contribution of smooth and saccadic responses for FtB and BtF motion differ drastically (Fig. 1i, m)”*

L. 123: sentence too long. Split into 2 sentences

We have split the sentence.

L. 131: “ ... reverse their turning direction for FtB from avoidance to stabilization. ...”. So, there is a functional interpretation of the anti-directional saccadic response?

We hope this is now clear with the amendment in Line 123.

L. 135: “ ... show that ...” should be “ ... suggest that ...”

Changed

L. 138: insert “ ... of view ...” after “ ... binocular ...”

Added “field of view” after binocular

L. 144/5: what is again the difference between “monocular” and “unilateral” stimuli?

We described this previously: “Given that flies have a binocular field that spans 40° of visual angle, our unilateral split-screen stimuli stimulates the contralateral eyes. “

L. 228: “ ... required for adequate course control.” “adequate” implies you know what course control should look like. Either you specify exactly what “course control” behaviour you are referring to or you drop “adequate” and stay at the general level.

done

L. 506 “ ... mediate binocular behavioural instructions.” This is a case of uncommon/unlucky wording. I don’t think that gap junctions “instruct” any behaviour – they allow current to pass. An instruction implies a (conscious) purposeful action. What gap junctions do is they contribute to behavioural modulations. Pls change the wording.

We have changed mediate to coordinate, which emphasizes them as the source of the instruction and as part of the process.

L. 534: “ ... are directly responsible ...” sounds like necessary and sufficient. I recommend wording more cautiously. “ ... are involved in ...”

Changed

L. 541: “ ... discriminate between local and global optic flow.” is unclear. This should be related to the ambiguity of horizontal motion due to either translation or rotation, which requires the availability and integration of ipsilateral and contralateral motion signals.

We have changed the sentence to make it more specific:

“Notably, in Flp-shakB flies subjected to front-to-back (FtB) motion, smooth turning in the direction of the stimulus was enhanced, while saccadic counter turns were diminished (Fig. 8g-i, Extended Data Fig. 12b), suggesting that flies cannot properly discriminate between translation and rotation, which requires the integration of bilateral optic flow motion signals.”

L. 552: “ ... summing binocular cues.” Would be great if you did add the functional purpose here, too, as you did for the “monocular information”

We have added the following summary sentence.

Line 553: “The result is a reduced range of locomotor behaviour, with the fly showing a similar turning response to all three types of movement (Fig. 8j).”

Our detailed functional interpretation can be found in the discussion section.

L595: “Taken together ...” Good to have a summarizing sentence. But it would be even better to be more specific on the functional context when stating “ ... finely tuned to specific properties of global motion ...”

We changed “specific properties to” with “bilateral heterogeneity in”

L. 613: induce blank “ ... be elicited ...” -

Corrected

Same sentence: this is a good example. It should be mentioned though, that optomotor course control does not appear to involve the use of efference copies while course changes do (Fenk et al. 2021).

We agree, but we don't think this is the place to mention it. We are not discussing the fact that we want to suppress visual stimulation via efferent copy.

L. 626: “ ... behavioural state ...” should be “ ... locomotor state ...”

Changed

L. 634: “ ... the panoramic position of optic flow ...” Optic flow is always panoramic and covers the entire visual field. What you probably mean is which section of an optic flow field is chosen as a visual stimulus and onto which area of the (compound) eye it is projected. Of course, optic flow can locally be modified due to independently moving/approaching objects/animals within the visual field. Those local optic flow inconsistencies would potentially trigger other visually controlled behaviours, different from course or attitude control.

Agreed, we removed “panoramic” and adapted the sentence to: “Thus, heterogeneities in global optic flow play a critical role in controlling behaviours”

L. 712: replace “ ... instruct ...” with “... perform ...”

Changed

L. 713: “ ... optic flow is global or local.” Pls see my earlier comment. Optic flow is by definition always global, as it is related to self-motion. But as I mentioned above, any retinal image shifts resulting from external object motion would locally produce inconsistencies. This is simply the case because local retinal image shifts (irrespective of its origin) will be linearly combined, and the resulting local motion vector is the input to the motion vision pathway.

Agreed, changed to: “This can be relevant for interpreting local irregularities in optic flow patterns“

L. 719: sentence incomplete – what “computation”? I am still not overly happy with “ ... stimulus-behaviour mapping ...” as it sounds both clumsy and vague.

We respectfully disagree. From our perspective, the term stimulus-response is intended to make an association with the stimulus-response mapping classically done in neuroscience. This is a reference to David Marr's three levels of analysis. We suggest that a fine-grained mapping of different types of optic flow to different types of turning behaviour is needed to understand exactly what the LPTC network is trying to achieve. We believe that this is still largely unexplored, despite being an essential step in understanding the role of the circuit interactions involved.

We have altered the phrasing slightly to make this clear.

Line 722: “A rigorous characterization of the stimulus–behaviour mapping is required to define the computation being performed by the LPTC network and novel genetic tools are needed to understand their neuronal implementation”

L. 740: replace “instructing”

Changed to “eliciting”

L. 745: should be “ ... experimentally informed ...” rather than “ ... biologically inspired ...”

Changed

L. 751/2: "... rather the merely averaging neighbouring signals, ..." I know the work by Farrow et al. , but reader who don't would be confused by this statement. Also, the working belittles the study. The coupling of neighbouring LPTCs was interpreted in more functional detail.

We changed "merely" with "passively". Which should make the statement more clearly.

Line 751: *"Moreover, they suggest that gap junctions can facilitate non-linear operations that play a decisive role in animal course control rather than passively averaging neighbouring signals, as suggested through careful experiments and of the electrical HS-H2 interaction in blowflies"*

L. 755/6: ... that give rise to behaviour." should be " ... that modulate the behavioural output."

Modified

L. 757: " ... this computation ..." what computation? Reference not clear.

Changed "computation" with "bilateral optic flow integration"

L. 759ff: The newly added text is more confusing than explanatory. It would require a bit more explanation that is linked to the current study and motivates why it is included. The last sentence, where you proudly mention the "... expansion of the known repertoire..." [of optomotor responses] finishes "... a comprehensive understanding of the neuronal mechanisms of vision in flies" This sounds vague and it conveys the impression that we don't know much about vision in flies – which is, I think, not quite correct.

It is also worth noting that just observing a "new behaviour" and quantifying is only one first step. Anything that is related to "a comprehensive understanding" also requires a clear understanding of the natural behavioural context including a functional interpretation. Knowing what behavioural state needs to be controlled immensely simplifies the task of modelling it, which in turn enables us to identify the "computation" in the nervous system required to achieve it.

Getting back to the final sentence: In my view, it also dilutes the merits of your finding that gap junctions play an important role in the integration of binocular motion information.

Thanks for note. We have clarified the paragraph:

Line 763: "In flies, pre-motor descending neurons (DNs) connect the brain to the ventral nerve cord and are responsible for generating appropriate visuomotor commands. DN that receive inputs from LPTCs have been shown to be involved in walking and flying. However, the picture is further complicated by the presence of interneurons modulating LPTCs and DN and recent finding showing that DN act in coordination to control behaviour. Specifically, a recent study showed that inhibitory inputs from an interhemispheric interneuron, bIPS, to a descending neuron, DNp15, are required to maintain path straightness. This particular subnetwork is unlikely to be involved in the counter-saccadic behaviour described in our work, as DNp15 innervates the neck and the halteres. It is more likely that other descending pathways connected

to HS or H2 cells (Extended Data Fig. 15d) are required, such as those involved in saccadic turns. Nonetheless, it highlights the challenges involved in understanding the full circuit involved in transforming optic flow patterns into specific motor outputs.”

Additional comment:

In your rebuttal letter, after explaining the value of the added model, you wrote:

“This information can then be used by several different downstream elements, not just bIPS or DNp15, to accurately estimate optic flow.”

This is not directly related to your current manuscript, but more of a general comment: To understand the design of visuo-motor control, it is important to keep in mind the different levels required. Generally speaking, these levels include sensing, actuation (motor systems) and the control architecture connecting the former to the latter. A major part of this control architecture are the descending pathways. Their function is no longer the encoding of sensory information (optic flow estimation) but rather the integration of any sensory information that results in the generation of an appropriate control command (driving the motor neurons/systems), ideally within a coordinate system that is aligned with the coordinate system of the actuators. Most DNs connect to motor neurons which, in turn, connect to specific muscles involved in the control of the behavioural output and are likely to do this in the respective motor system coordinates. So, what needs to be estimated is the state change (self-motion) which is based on processing optic flow at the level of LPTCs in the 1st instance (including ipsilateral and heterolateral connections) and is further refined by integrating binocular signals. The output of the integration already provides a control signal in motor coordinates, rather than encoding/estimating optic flow.

We appreciate the comment. We recognise the complexity of the system. Although LPTC activation has been shown by us and others to induce certain behaviours, it has not been possible to observe clear behavioural perturbations when LPTCs are specifically ablated. The interpretation has always been a redundancy in the code. However, given the specificity of the connectivity, we find this rather hard to believe. We believe that we haven't been able to evaluate their actual control signal because we don't have a refined understanding of their behaviour. This, we believe, lies the most fundamental contribution of our work.

Reviewer #4 (Remarks to the Author):

I appreciate the authors' response to my review comments and the changes made to the manuscript. In particular, the new simulation data (Extended Data Figure 15) using the HS-H2 model addresses a large part of our concerns by providing a theoretical basis for the role of the HS-H2 network in binocular optic flow processing. I would like to mention that our previous comment regarding the circuit element responsible for nonlinear binocular integration has not been addressed.

We apologise for this. We have referred to the work of Farrow et al, which looks at the involvement of nonlinear computation in the HS-H2 circuit. In their work, the combination of active conduction and gap junction is required for binocular nonlinear computation. However, the description of the circuit elements involved in this behaviour is beyond the scope of this paper. This would require a dedicated study screening all possible descending neurons that could be involved in this process. As shown in our previous answer, there are many, and as recently shown by the lab of Paven Ramdya, the coordination of DN, rather than one specific pathway, will be involved. We have added this citation to our amended discussion. Please see the last reply to reviewer #2.

Despite this, I think that all the other comments have been directly addressed. With the modeling work in addition to the extensive amount of all the other experiments, I believe the authors have done a commendable job dissecting the role of commissural visual neurons (H1) and gap junctions in the binocular vision of *Drosophila*. Thus, we recommend the paper for publication as it is.

I have one last request to the authors and leave it to their discretion whether to reflect it in the manuscript. As you mentioned in the rebuttal letter, the paper does not provide “the specific changes in electrical connectivity between HS and H2 that generate the nonlinear interaction.” Thus, I believe that the new sentence in the abstract is not the most accurate statement for the findings of the paper. Therefore, I recommend reverting the following sentence in the abstract to the previous version.

(Line 16-18, new version): We show that bilateral electrical coupling between a specific set of optic flow-sensitive neurons in the lobula plate of the fly brain is required to control these behaviors.

(Line 16-18, previous version): We identify a specific set of optic flow-sensitive neurons in the lobula plate of the fly brain and show that bilateral electrical coupling is required to control these behaviors properly.

We have changed the sentence in the abstract.

REVIEWERS' COMMENTS

Reviewer #2 (Remarks to the Author):

I have had a look at the rebuttal letter and the latest revision of the MS now. Again, I appreciate the authors' efforts to deal with the reviewers' comments, now presenting a version of their work which addresses nearly all queries at an acceptable level and includes an appropriate and subtle change of the title. In a few cases they did not take on board the suggestions of the reviewer(s) but provided an acceptable reasoning why they didn't. Other issues, e.g. the call for additional experiments would have gone beyond the scope of the current study.

After a truly thorough reviewing process and based on the latest version, I believe the MS is now ready for publication.

The authors may consider my last two minor comments when assembling the finalized MS version.

In the Abstract:

L. 17: "We identify a specific set of optic flow-sensitive neurons in the lobula plate of the fly brain and show that bilateral electrical coupling is required to control these behaviors properly."

This sounds as if (i) you did identify the neurons involved, and (ii) it does not link well the previous to the next sentence. Maybe change to:

"We show that bilateral electrical coupling between identified optic flow-sensitive neurons in the lobula plate of the fly brain is required to control these behaviors."

L. 597: pls change "... heterogeneities ..." to "... local inhomogeneities ..." – which would stronger emphasize the point (i.e. local vs global optic flow).